# Multi-View Causal Discovery without Non-Gaussianity: Identifiability and Algorithms

## Abstract

Causal discovery is a difficult problem that typically relies on strong assumptions on the data-generating model, such as non-Gaussianity. In practice, many modern applications provide multiple related views of the same system, which has rarely been considered for causal discovery. Here, we leverage this multi-view structure to achieve causal discovery with weak assumptions. We propose a multi-view linear Structural Equation Model (SEM) that extends the well-known framework of non-Gaussian disturbances by alternatively leveraging correlation over views. We prove the identifiability of the model for acyclic SEMs. Subsequently, we propose several multi-view causal discovery algorithms, inspired by single-view algorithms (DirectLiNGAM, PairwiseLiNGAM, and ICA-LiNGAM). The new methods are validated through simulations and applications on neuroimaging data, where they enable the estimation of causal graphs between brain regions.

## 1 Introduction

Causal discovery is a fundamental problem in scientific data analysis as well as several technological applications (Peters et al., 2017). The basic problem is that we are given a number of observed variables, and we need to infer which variables cause which, and with what "connection strengths". In the typical case, we only observe the data passively without any possibility of performing interventions, and this purely observational quality of the data makes the problem difficult.

Often, the problem is formalized as a structural equation model (SEM) (Bollen, 1989), also called a functional causal model (Pearl, 2009). Then the question is how to estimate the SEM parameters using statistical theory. In fact, before even considering estimation methods, we need to know if it is at all possible to solve the problem. This is the question of *identifiability* of the model: can the parameters that describe the causal relations and directions be (uniquely) estimated?

A well-known fact is that if all variables are Gaussian and the model is linear, identifiability of the SEM is very problematic. Some of the earliest work showed that the directions can sometimes be recovered by an analysis of the *conditional independencies* between variables (Spirtes et al., 2001); however, this is only possible in some special cases and notably not possible when observing only two variables. Recent literature has, therefore, focused on different departures from the linear-Gaussian framework to achieve identifiability. A major advance was to consider a linear model together with *non-Gaussianity* (Shimizu et al., 2006), which leads to full identifiability of the model under weak conditions such as acyclicity; however, such non-Gaussianity may not be strong enough in many data sets. A related framework was proposed by assuming that the cause undergoes a *non-linear transform*, while the disturbance is still additive (Hoyer et al., 2008); however the nonlinearity may not be strong enough in many applications, and such nonlinearity slightly contradicts the linear additivity of the disturbance. Further related frameworks were proposed by (Peters & Bühlmann, 2014; Zhang & Hyvärinen, 2009; Monti et al., 2019). Each framework has lead to development of a number of algorithms, the discussion of which we defer to the Background Section below.

An alternative recent framework is given by the *multi-view* setting, where data is collected from different sources whose similarities can be leveraged for estimation and identifiability. However, current approaches using multi-view structure are scarce and limited; either they assume the views are statistically independent (Shimizu, 2012) or make strong assumptions on their dependencies (Chen et al., 2024). Here, we consider the linear multi-view SEM, providing a general theory and algorithms that leverage that multi-view structure.

Our main contributions are:

1. We generalize some of the most well-known *algorithms* for single-view causal discovery to the multi-view setting. The generalizations of PairwiseLiNGAM and DirectLiNGAM result in completely new and fast causal discovery using second-order statistics only, while our adaptation of ICA-LiNGAM improves on Chen et al. (2024).

2. We show that our algorithms are *consistent* under weak conditions: we can replace the non-Gaussianity assumption on the disturbances by conditions on correlations between the views and "diversity" of such second-order statistics (SOS). Still, if the disturbances are non-Gaussian, we generalize Shimizu (2012); Chen et al. (2024).

3. The proofs above directly lead to rigorous *identifiability* results on the model, including cases where all the disturbances are Gaussian, or, equivalently, identifiability results based on SOS only. Importantly, only weak conditions on SOS is necessary for identifiability, greatly generalizing previous work.

4. We make our algorithms usable by practitioners by providing them as *open-source*, and demonstrate their usefulness by benchmarking them on real neuroimaging data.

## 2 BACKGROUND

**Causal ordering**    In the following, we consider a random vector $x \in \mathbb{R}^p$ whose entries are nodes of a directed acyclic graph (DAG). The DAG is represented by an adjacency matrix $B \in \mathbb{R}^{p \times p}$, where non-zero values indicate the edges of the graph. The entries of $x$ can be reordered such that each "causes" the next. This is formalized by the adjacency matrix admitting a special decomposition $B = P^\top T P$, where $T$ is a strictly lower triangular matrix and $P$ is a permutation matrix referred to as the *causal ordering*. In general, the causal ordering is not unique.

**LiNGAM: A single-view model for causal discovery**    We consider a structural equation model (SEM), also known as a functional causal model (Bollen, 1989; Pearl, 2009), that is linear and where the data follows

$$x = Bx + e \tag{1}$$

and the entries $e_1, \ldots, e_p$ of the random vector $e \in \mathbb{R}^p$ are independent noise terms, or *disturbances*. Causal discovery consists in inferring the parameters $B$ of the model, from observations of $x$. Yet, the identifiability of the model and therefore the uniqueness of $B$ is not straightforward. In fact, it is well-known that with Gaussian disturbances, the model is unidentifiable in general (Richardson & Spirtes, 2002; Genin, 2021). A major advance by Shimizu et al. (2006) was assuming *non-Gaussian* disturbances, leading to their Linear Non-Gaussian Acyclic Model (LiNGAM).

**Identifiability of LiNGAM**    Shimizu et al. (2006) showed that the LiNGAM model is identifiable in terms of the matrix $B$. A rigorous re-statement of this result is given in Appendix A.2. A further question that has received less attention is under what conditions the model is identifiable in terms of the causal ordering $P$. In fact, it is not in general: specifically, there may exist many permutation matrices $P$ and strictly lower triangular matrices $T$ such that the generated data has the same distribution and the generating permutation cannot be identified. For instance, in the degenerate case $B = T = 0$, any permutation matrix $P$ is equally valid and gives the same data distribution; any very sparse $B$ leads to several equally valid orderings. As is well-known, a DAG in general defines only a *partial* ordering in the sense that for some pairs of variables, we cannot necessarily say which is "earlier" and which is "later". Thus, if we want to make the causal ordering unique — *i.e.* to obtain a *total* ordering — we need further assumptions, as considered in later sections.

**Estimation of LiNGAM**    Since the LiNGAM model was proposed, two important categories of algorithms for estimating this model have been developed. First, algorithms based on *recursively* computing the *residuals* of pairwise regressions include DirectLiNGAM (Shimizu et al., 2011) and PairwiseLiNGAM (Hyvärinen & Smith, 2013). These compare pairs of variables from $x$ and deduce what is the causal direction between the two variables based on regressing the variables in the two directions. By aggregating this information, they finally determine a causal ordering $P$. Once an

ordering is found, the matrix $\boldsymbol{T}$ can easily be obtained with conventional methods such as least-squares (LS) regression, which then gives $\boldsymbol{B} = \boldsymbol{P}^\top \boldsymbol{T} \boldsymbol{P}$.

The second category is the ICA-based algorithm that was proposed in the original LiNGAM paper (Shimizu et al., 2005). It is based on the observation that LiNGAM model can be rewritten as a latent variable model, in particular an ICA model (Hyvärinen et al., 2001) as

$$\boldsymbol{x} = \boldsymbol{A}\boldsymbol{s} \ . \tag{2}$$

Again, the entries in $\boldsymbol{s}$ are independent and non-Gaussian, and the "mixing" matrix $\boldsymbol{A}$ expresses how the data is generated from the latent variables $\boldsymbol{s}$. Many methods developed for ICA can be used to estimate the matrix $\boldsymbol{A}$. However, it is important to consider the special structure of the mixing, since $\boldsymbol{A} = (\boldsymbol{I} - \boldsymbol{B})^{-1}$, where $\boldsymbol{B}$ is a DAG (Shimizu et al., 2006). This method first recovers the adjacency matrix $\boldsymbol{B}$, and then estimates a causal ordering $\boldsymbol{P}$ from $\boldsymbol{B}$.

## 3   A Multi-View Model for Causal Discovery

We now extend the linear acyclic model in Eq. 1 to the multi-view setting, leading to a Linear Multi-View Acyclic Model (LiMVAM). Suppose we observe $m$ different views of a $p$-dimensional vector $\boldsymbol{x}^i$. The most general linear acyclic extension is

$$\boldsymbol{x}^i = \boldsymbol{B}^i \boldsymbol{x}^i + \boldsymbol{e}^i \tag{3}$$

for views $i \in [\![1, m]\!]$. The $\boldsymbol{B}^i$ are adjacency matrices of DAGs, and $\boldsymbol{e}^i$ are view-specific disturbance vectors (with entries of non-zero variance). Similarly to Eq. 2, the LiMVAM model can equivalently be written as a latent-variable model

$$\boldsymbol{x}^i = \boldsymbol{A}^i \boldsymbol{e}^i \tag{4}$$

where the mixing matrix has a special structure, $\boldsymbol{A}^i = (\boldsymbol{I} - \boldsymbol{B}^i)^{-1}$, which we will exploit later in Section 5. Our model generalizes that by Chen et al. (2024), who assumed a specific structure on the influences, as will also be explained in Section 5.

To exploit the multi-view structure, we follow Shimizu (2012); Chen et al. (2024) and assume that all adjacency matrices $\boldsymbol{B}^i$ share the same causal ordering. Equivalently, they admit a decomposition $\boldsymbol{B}^i = \boldsymbol{P}^\top \boldsymbol{T}^i \boldsymbol{P}$ where $\boldsymbol{T}^i$ is strictly lower triangular and $\boldsymbol{P}$ is a permutation matrix that is shared among the views $i$. Moreover, as commonly assumed in causal discovery, we require the components of each $\boldsymbol{e}^i$ to be mutually independent, for any given $i$. (In contrast, correlations between views $i$ and $i'$ will be seen to be essential for identifiability.) Contrary to most existing approaches, we do not impose any restriction on the distribution of the disturbances, such as non-Gaussianity, which greatly broadens the applicability of the model.

The next two sections develop efficient methods for estimating the adjacency matrices $\boldsymbol{B}^i$, together with identifiability theory. We begin in Section 4 with methods that estimate the causal ordering using recursive residuals and using only second-order statistics, based on the single-view algorithms DirectLiNGAM (Shimizu et al., 2011) and PairwiseLiNGAM (Hyvärinen & Smith, 2013). Then, we propose a modified version of the ICA-based multi-view model from Chen et al. (2024) in Section 5.

## 4   Estimation by SOS and recursive residuals

First, we show how to estimate the model using Second-Order Statistics (SOS) only, and how the SOS alone lead to identifiability. To do so, we will rely on a principle that we call *recursive residuals*. While it has been used by Shimizu et al. (2011); Hyvärinen & Smith (2013); Shimizu (2012), we provide here a unified treatment.

### 4.1   Causal ordering using recursive residuals

Methods based on what we call recursive residuals are built on the following principle: 1) we can find a root variable (*i.e.* one with no parents) by analyzing the causal directions between all the pairs of variables, and 2) when all other variables are regressed on a root variable, the resulting residuals preserve the causal ordering of the original variables (Shimizu et al., 2011, Lemma 2 and

Corollary 1). In the multi-view case, the same principles apply since all views are assumed to share a common ordering. A formal proof is provided in Appendix D.3.1 and D.3.2.

Algorithms based on recursive residuals therefore follow the same recursive scheme to recover the causal ordering: (i) perform pairwise regressions to obtain residuals, (ii) identify a root variable according to a specific criterion of causal direction computed on the residuals, and (iii) remove the selected root and repeat the procedure on the residuals until all variables are ordered. The main difference between algorithms lies in step (ii), *i.e.* in how the root variable is selected.

Since a root variable has no parents, we can find one by testing the causal direction for every pair of variables and taking as a root a variable that is never inferred to be an effect. A practical way to find the root variable from such tests with a finite sample is described in Appendix E.4.

Thus, root selection reduces to designing a criterion of causal direction capable of reliably distinguishing the causal direction between two variables. In the following, we introduce two new such criteria in the multi-view setting. For simplicity of notation, we consider two random variables aggregated across views: $\boldsymbol{x}_1 = (x_1^1, \ldots, x_1^m)^\top$ and $\boldsymbol{x}_2 = (x_2^1, \ldots, x_2^m)^\top$; either $\boldsymbol{x}_1$ causes $\boldsymbol{x}_2$, or the converse. Covariance matrices are denoted by $\boldsymbol{\Sigma}_{\boldsymbol{x}_1} = \mathbb{E}[\boldsymbol{x}_1 \boldsymbol{x}_1^\top]$, and likewise for $\boldsymbol{x}_2$, $\boldsymbol{e}_1$ and $\boldsymbol{e}_2$.

## 4.2 LIKELIHOOD-BASED CRITERION: PAIRWISELIMVAM

We first derive a criterion that extends the PairwiseLiNGAM criterion of Hyvärinen & Smith (2013) to the multi-view setting. Assuming Gaussian disturbances, we compute the expected log-likelihoods of two models: $\boldsymbol{x}_1 \to \boldsymbol{x}_2$ and $\boldsymbol{x}_2 \to \boldsymbol{x}_1$ (where an arrow denotes a directed edge between two adjacent variables in the causal graph). This leads to the following criterion, which is the logarithm of the maximum likelihood ratio (LR) between the two models:

$$\mathrm{LR}_{12} = -\log \det \boldsymbol{\Sigma}_{\boldsymbol{x}_1} - \log \det \boldsymbol{\Sigma}_{\boldsymbol{e}_2} + \log \det \boldsymbol{\Sigma}_{\boldsymbol{x}_2} + \log \det \boldsymbol{\Sigma}_{\boldsymbol{e}_1} \ . \tag{5}$$

We prove in Appendix B that, under very weak constraints on the correlations of variables and views, this criterion is positive when $\boldsymbol{x}_1 \to \boldsymbol{x}_2$, and negative in the opposite direction, *no matter whether the disturbances are Gaussian or non-Gaussian*. Importantly, this makes the causal direction — and thus the entire causal ordering — identifiable under very general circumstances. So, although we developed this criterion for Gaussian disturbances, we emphasize that the resulting method is consistent regardless of the Gaussianity of the disturbances.

The necessary and sufficient conditions for the consistency of this criterion are given in Appendix B.6; a sufficient condition, assuming for simplicity that $\boldsymbol{x}_1$ is the cause, is that two views $i, i'$ exist such that $|\mathrm{corr}(e_2^i, e_2^{i'})| \neq |\mathrm{corr}(x_1^i, x_1^{i'})|$, while $\mathrm{corr}(x_1^i, e_2^i) \neq 0$ or $\mathrm{corr}(x_1^i, e_2^{i'}) \neq 0$. This essentially says that the views need to be correlated and the correlations need to be diverse, and will be explained in more detail below. In practice, we replace the covariance matrices $\boldsymbol{\Sigma}_{\boldsymbol{x}_1}$, $\boldsymbol{\Sigma}_{\boldsymbol{e}_1}$, $\boldsymbol{\Sigma}_{\boldsymbol{x}_2}$, and $\boldsymbol{\Sigma}_{\boldsymbol{e}_2}$ with consistent estimators (see Appendix E.1); we use Ordinary LS to compute the residuals throughout this paper. Thus, we obtain a consistent estimator of the true direction.

## 4.3 CROSS-COVARIANCE-BASED CRITERION: DIRECTLIMVAM

Our next criterion is a new multi-view extension of the DirectLiNGAM criterion (Shimizu et al., 2011). In a single-view setting, Shimizu et al. (2011) relied on the fact that $x_1$ and $e_2$ are *independent* if the causal direction is $x_1 \to x_2$, and measured the independence using a kernel-based estimator of mutual information called KGV (Bach & Jordan, 2002). A multi-view extension was actually proposed by Shimizu (2012), where the correct direction was chosen by comparing the *sums over views* of these scores for the two directions. However, such summation over views does not fully take advantage of the multi-view structure. Thus, the disturbances still needed to be non-Gaussian.

Here, we efficiently exploit the multi-view framework while only looking at SOS. We choose to evaluate the *cross-covariance* between the *entire* vectors $\boldsymbol{x}_1 = (x_1^1, \ldots, x_1^m)^\top$ and $\boldsymbol{e}_2 = (e_2^1, \ldots, e_2^m)^\top$ for the direction $\boldsymbol{x}_1 \to \boldsymbol{x}_2$, and between $\boldsymbol{x}_2$ and $\boldsymbol{e}_1$ for the direction $\boldsymbol{x}_2 \to \boldsymbol{x}_1$. Specifically, the correct direction is chosen by comparing the Frobenius norms of the cross-covariances between regressors and residuals. Thus, we obtain the following criterion:

$$\mathrm{FC}_{12} = \left\| \mathbb{E}\left[\boldsymbol{x}_2 \boldsymbol{e}_1^\top\right] \right\|_F - \left\| \mathbb{E}\left[\boldsymbol{x}_1 \boldsymbol{e}_2^\top\right] \right\|_F \ . \tag{6}$$

We prove in Appendix C that this criterion consistently finds the correct direction under the same conditions as the likelihood-based criterion, again *no matter whether the disturbances are Gaussian or non-Gaussian*.

## 4.4 ESTIMATING THE CAUSAL COEFFICIENTS

Applying the recursive scheme explained in Section 4.1, and using the criteria from Eq. 5 or Eq. 6, we obtain a causal ordering encoded by $\boldsymbol{P}$. Importantly, since the ordering is now known, the causal matrices $\boldsymbol{B}^i$ can be uniquely recovered with little effort. In fact, the model becomes

$$\boldsymbol{x}'^i = \boldsymbol{T}^i \boldsymbol{x}'^i + \boldsymbol{e}'^i , \qquad i \in [\![1, m]\!] \tag{7}$$

where $\boldsymbol{x}'^i = \boldsymbol{P}\boldsymbol{x}^i$ are the reordered observations and $\boldsymbol{e}'^i = \boldsymbol{P}\boldsymbol{e}^i$ are the reordered disturbances. By construction, each variable $x_j'^i$ only depends on its predecessors $x_1'^i, \ldots, x_{j-1}'^i$. For each $j \in [\![2, p]\!]$, we estimate the $j$-th row of all matrices $\boldsymbol{T}^i$ jointly using one-step Feasible Generalized Least Squares (FGLS) Zellner (1962), which is more efficient than Ordinary LS. A detailed description of one-step FGLS is provided in Appendix E.5. Under the assumption that disturbances are mutually independent within each view, FGLS yields a unique regression for every variable on its parents. Finally, the adjacency matrices are recovered as $\boldsymbol{B}^i = \boldsymbol{P}^\top \boldsymbol{T}^i \boldsymbol{P}$.

The overall procedure is summarized in Algorithm 3 in the Appendix; it notably requires no hyperparameter tuning. Its worst-case *computational complexity* is in $O(m^3 \cdot p^3 \cdot n)$, which we show in Appendix E.7. In practice, these algorithms are highly parallelizable as they rely on basic algebraic operations (*e.g.* multiplying pairs of matrices), and fast as they do not require numerical optimization of an objective function.

## 4.5 IDENTIFIABILITY USING SECOND-ORDER STATISTICS

We can build an identifiability theory based on the conditions required by the criteria LR and FC (given in Appendices B and C). We start by formulating the following central assumption, which will be seen to be a sufficient condition for identifiability. Even milder sufficient conditions are discussed in Appendix D.1 but they are harder to interpret.

**Assumption 1** (Correlation and diversity across views) *For any two variables $\boldsymbol{x}_j$ and $\boldsymbol{x}_{j'}$ such that $\boldsymbol{x}_j \to \boldsymbol{x}_{j'}$ (in the sense that $\boldsymbol{B}_{j'j}^i \neq 0$ for at least one $i$), there exist two distinct views $i \neq i'$ such that a) those variables are correlated in at least one of those views, $\mathrm{corr}(x_j^i, x_{j'}^i) \neq 0$ or $\mathrm{corr}(x_j^{i'}, x_{j'}^{i'}) \neq 0$, and b) their correlations fulfill the "diversity" condition:*

$$|\mathrm{corr}(x_j^i, x_j^{i'})| \neq |\mathrm{corr}(e_{j'}^i, e_{j'}^{i'})| . \tag{8}$$

To better understand this assumption, note first that it requires some *correlation across views*: it rules out the trivial case where all cross-view correlations are zero. Moreover, it demands some *diversity* in these correlations. Consider the extreme situation where, for standardized variables, $x_j^i = \alpha x_j^{i'}$ and $x_{j'}^i = \alpha x_{j'}^{i'}$ (and analogously for the residuals), for some scalar $\alpha$. In this case, all correlations equal 1, so condition Eq. 8 is violated. Appendix D.2 provides more realistic examples of violations. We next introduce an additional assumption that ensures identifiability of the total causal ordering.

**Assumption 2** (Dense connectivity of the graph when pooled across views) *Let $\mathcal{G}$ denote the union graph of the $m$ views, obtained by taking the union of their edge sets, and assume that the ordering induced by $\mathcal{G}$ is* total*, i.e. there exists a directed path between any two variables.*

With these assumptions, we prove in Appendix D.3 the following identifiability result.

**Theorem 1** (Identifiability of the ordering and adjacency matrices $\boldsymbol{B}^i$) *Assume the data follows the model in Eq. 3, and that Assumption 1 holds. Then, causal (partial) ordering and causal coefficients $\boldsymbol{B}^i$ are identifiable. If we further make Assumption 2, then $\boldsymbol{P}$ is identifiable as well.*

The identifiability of the *partial* ordering does not necessarily mean that the permutation matrix $\boldsymbol{P}$ can be recovered uniquely. If the ordering is only partial (for example, in a three-nodes graph: $\boldsymbol{x}_1 \to \boldsymbol{x}_2$ and $\boldsymbol{x}_1 \to \boldsymbol{x}_3$, but there is no relation between $\boldsymbol{x}_2$ and $\boldsymbol{x}_3$, so both orderings $1 \to 2 \to 3$ and $1 \to 3 \to 2$ are valid), then $\boldsymbol{P}$ is only identifiable up to a class of permutations that induce the same partial ordering. The above theorem shows a case where $\boldsymbol{P}$ can be (uniquely) identified.

## 5 Estimation by Non-Gaussianity assuming shared disturbances

Next, we provide an alternative approach to the LiMVAM model definition and estimation. We utilize the non-Gaussianity of the disturbances to obtain identifiability conditions that are different from the previous section. However, we still do not impose non-Gaussianity as a necessary condition.

**Model definition** The approach here is to propose a particular model for the disturbances by supposing that they can be additively composed into two terms: one term is view-specific, while the other is shared across views and therefore makes the views dependent. Specifically:

**Assumption 3** (Shared disturbances) *The disturbances can be decomposed as*

$$e^i = D^i s + n^i \tag{9}$$

*where the $D^i$ are diagonal matrices with positive entries on the diagonal, the $s$ has mutually independent entries and second-order moment $\mathbb{E}[ss^\top] = I_p$, and the view-specific noises are Gaussian $n^i \sim \mathcal{N}(0, \Sigma^i)$ and depend on the view $i$ via the second-order moments $\Sigma^i$ that are diagonal matrices. Lastly, the vectors $s$ and $n^i, i = 1, \ldots, m$, are assumed to be mutually independent.*

This Assumption and Eq. 9 are related to Chen et al. (2024) who where originally inspired by the multi-view Shared ICA of Richard et al. (2021). Specifically, Chen et al. (2024) considered the special case where the matrices $D^i = I_p$, so that there are the same disturbances $s$ over views. We argue here that their model was too restrictive and unrealistic as it imposes an arbitrary scaling for the data variables. Consider the root variable of the DAG, denoting it by $x_i$. Its disturbance is $s_i + n_i$ which has variance of at least one since the variance of $s_i$ is defined as one. Thus, the root variable has variance of at least one which is absurd: the model should be invariant to the scaling of the variables. Likewise, suppose we create a new data set by dividing just one view, say $x^1$ by, say 10, giving new data $x'^1$. This new data does not follow the model by Chen et al. (2024) anymore since $s$ should be rescaled to $s/10$ for just this view; now the $s/10$ here is different from the $s$ in other views. Such problems motivated us to develop a model where the scaling terms $D^i$ are new parameters, and thus to the definition in Eq. 9.

**Identifiability** Now we proceed to show the identifiability of the model based on Assumption 3. The starting point is that our model can be rewritten as a multi-view ICA model as in Eq. 4, just like in the case of LiNGAM. Thus, the methods developed for multi-view ICA can be used to analyze its identifiability. We first discuss the assumptions used for our theorems.

**Assumption 4** (Diversity of the views with Gaussian disturbances) *If $j$ and $j'$, $j \neq j'$, are the indices of two Gaussian common disturbances in $s$, then the number of views is at least three, and there exists a view $i$ such that $\frac{\Sigma_{jj}^i}{(D_{jj}^i)^2} \neq \frac{\Sigma_{j'j'}^i}{(D_{j'j'}^i)^2}$.*

This assumption, inspired by Richard et al. (2021), is trivially fulfilled if all disturbances are non-Gaussian. For Gaussian disturbances, it essentially states that the views should be diverse in terms of disturbances' SOS. This follows from the theory of Shared ICA which does not require the common disturbances $s$ to be non-Gaussian if there is enough diversity (Richard et al., 2021; Anderson et al., 2013). The next assumption assumes that the union graph of the $B^i$ is dense enough across views.

**Assumption 5** (Dense connectivity of the graph when pooled across views) *There exists a permutation $\bar{P}$ such that the matrices $\bar{T}^i = \bar{P} B^i \bar{P}^\top$ are strictly lower triangular, and for each entry $(j, k)$ with $j > k$, there is at least one view $i$ such that $\bar{T}_{j,k}^i \neq 0$.*

Note that this assumption is quite weak as the $T^i$ can be very sparse. This is especially the case when the number of views is large, since only one non-zero entry is required over all views.

We can now state our main identifiability result for this variant of the model.

**Theorem 2** (Identifiability with shared disturbances) *Assume the LiMVAM model in Eq. 3 together with Assumption 3. Under Assumption 4, the model is identifiable in the sense that the parameters $B^i$, $D^i$, and $\Sigma^i$ are identifiable. If we further make Assumption 5, $P$ is identifiable as well.*

The proof of the theorem is given in Appendix F. We thus see that for non-Gaussian disturbances, identifiability is achieved with weak conditions; in fact, no special conditions on the covariances are

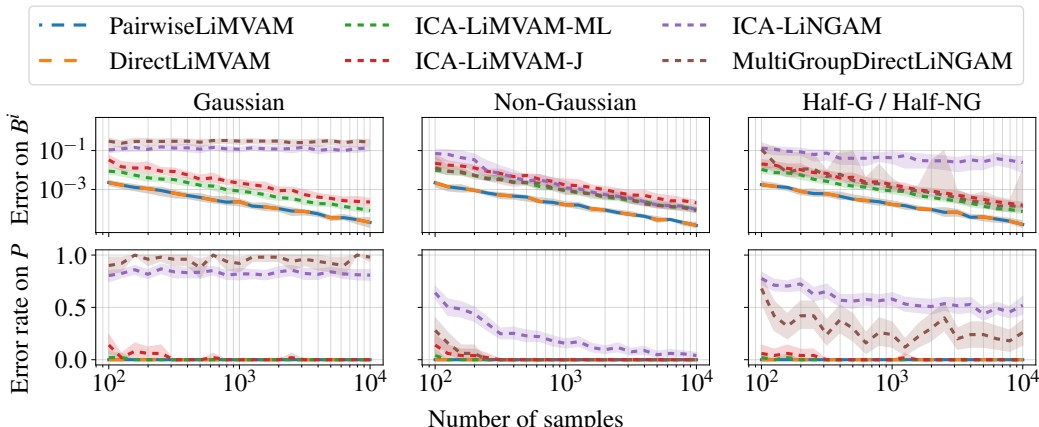

Figure 1: Separation performance of three recursive residuals algorithms and three ICA-based algorithms. We assume shared disturbances as in Eq. 9. We used 50 different seeds. Lower is better.

needed. Still, the theory of Shared ICA leads to identifiability even for Gaussian disturbances, but with stronger conditions than obtained in Section 4. The utility of the Shared ICA theory lies also in the fact that it allows us to construct an algorithm very easily.

**Estimation**  The model can be estimated by a combination of the ICA-based estimation procedure by Shimizu et al. (2006), combined with the Shared ICA methods by Richard et al. (2021). We call it ICA-LiMVAM and explain it in detail in Algorithm 4. Surprisingly, the resulting algorithm gives estimates of $\boldsymbol{B}_i$ that are identical to Chen et al. (2024), even though our model is more general. In contrast, our algorithm estimates different noise variances and scalings $\boldsymbol{D}_i$. The worst-case *computational complexity* is in $O(tnmp^3 + mp^5)$, for $t$ iterations of the Shared ICA (ML) algorithm, $n$ samples, $m$ views and $p$ components. We detail this in Appendix G.2.

## 6 EXPERIMENTS

### 6.1 SIMULATIONS

We benchmark a total of six multi-view causal discovery algorithms on synthetic data. Three of these algorithms are based on recursive residuals: our PairwiseLiMVAM and DirectLiMVAM, as well as MultiGroupDirectLiNGAM from Shimizu (2012). The other three algorithms are ICA-based: ICA-LiMVAM which is essentially the same as Chen et al. (2024), as well as ICA-LiNGAM (Shimizu et al., 2006) which is naively applied to each view, separately. Note that ICA-LiMVAM comes in two variants, each using a different Shared ICA algorithm: either Shared ICA "ML", a likelihood-based method that can handle both Gaussian and non-Gaussian disturbances, or Shared ICA "J", which jointly diagonalizes covariance matrices without using non-Gaussianity.

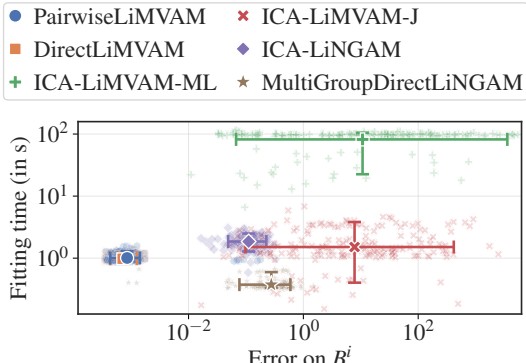

Figure 2: Estimation error of the adjacency matrices $\boldsymbol{B}^i$ versus total fitting time for each method, across 200 random runs with $m = 6$ views, $p = 5$ variables, and $n = 1000$ samples. Disturbances are drawn from a standard Gaussian. Note that the point clouds for PairwiseLiMVAM and DirectLiMVAM largely overlap.

The first synthetic experiment is inspired by Richard et al. (2021). We monitor the performance of causal discovery algorithms across varying sample sizes and disturbance distributions. More specifically, the data are generated according to the LiMVAM model with shared disturbances from Eq. 9. The performance of the causal discovery algorithms is measured by the $\ell_2$ distance between true and

estimated adjacency matrices $\boldsymbol{B}^i$, and the percentage of incorrectly estimated causal orders $\boldsymbol{P}$ over 50 random runs.

Results of the first synthetic experiment are plotted in Figure 1. Our algorithms PairwiseLiMVAM and DirectLiMVAM consistently outperform all others by a significant margin. ICA-LiMVAM yields reliable estimates but with larger errors. In contrast, MultiGroupDirectLiNGAM and ICA-LiNGAM, which are designed for non-Gaussian disturbances, fail entirely in the Gaussian setting, as expected. Note that the high error rate of ICA-LiNGAM in recovering the causal ordering $\boldsymbol{P}$ is likely explained by its inability to exploit that the ordering is shared across views.

For the second synthetic experiment, we report the trade-off between computational complexity and estimation error of the adjacency matrices. We use a different data generation model, described by Eq. 3 where the disturbances are Gaussian with equal variances. This breaks the assumptions for all methods but our PairwiseLiMVAM and DirectLiMVAM. As expected, we see in Figure 2 that our algorithms PairwiseLiMVAM and DirectLiMVAM provide a major reduction in estimation errors, at little or no computational cost. In contrast, the four other algorithms struggle with this data.

In Appendix H, we include further experiments that test Assumptions 4 and 5, as well as scalability with more views and variables. In particular, we find that DirectLiMVAM outperforms Pairwise-LiMVAM in some cases, which is less obvious from Figures 1 and 2 alone.

## 6.2 REAL DATA EXPERIMENTS WITH MEG

Next, we performed experiments on magnetoencephalography (MEG) data measuring human brain activity. MEG data has a high temporal resolution, so typically methods related to Granger causality are used. However, we are here interested in analyzing the *energies* of *oscillatory* signals. The energies change very slowly (on the time-scale of seconds), while the underlying brain activity being measured is likely to change much more quickly. Thus, it is appropriate to use a model of instantaneous causality as in our model.

We used the Cam-CAN dataset, which is the largest publicly available MEG dataset (Shafto et al., 2014; Taylor et al., 2017), and considered the "sensorimotor task" during which each participant had to respond with a right index finger button press to auditory/visual stimuli. Further details are available in Appendix H.2.1. We applied PairwiseLiMVAM to these data, each of the 98 participants being one view. The experiment ran in 2 hours and 20 minutes using 5 CPUs. Although our model does not require or assume any similarity across the matrices $\boldsymbol{T}^i$, one may reasonably hypothesize that the participants' brains share structural patterns. So, to facilitate population-level interpretation, we computed the element-wise median of the $\boldsymbol{B}^i$ matrices.

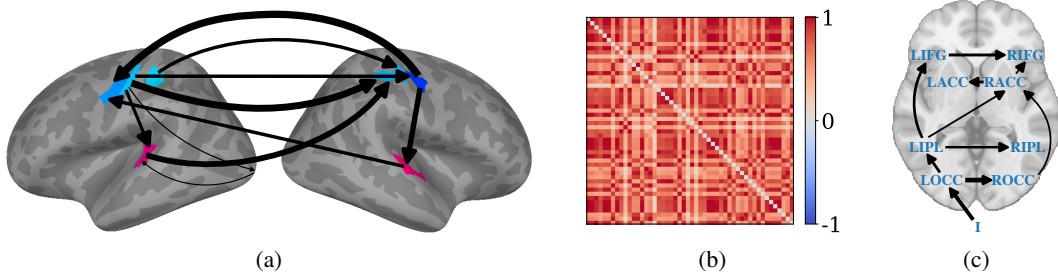

(a)            (b)            (c)

Figure 3: (a) Top ten strongest median causal effects (estimated by PairwiseLiMVAM) across 98 Cam-CAN (MEG) participants. (b) Pearson correlations between median causal matrices across 50 runs in MEG. (c) Top ten strongest median causal effects from PairwiseLiMVAM across 9 fMRI participants. Arrows indicate direction and strength of effects, with width proportional to magnitude.

Fig. 3a displays the ten strongest (in absolute value) median causal effects. Notably, many causal connections are between homologous regions across hemispheres — particularly within the primary motor and somatosensory cortices — consistent with prior findings that symmetric brain regions exhibit strong inter-hemispheric correlations Li et al. (1996). Additionally, numerous arrows involve the postcentral regions, which have been implicated in fine motor control and hand-related tasks Braun et al. (2002). To evaluate the robustness of the results, we repeated the experiment 50 times,

each time randomly selecting 30 participants from the full cohort. We assessed consistency by measuring the Pearson correlation between the resulting median matrices across runs. As shown in Figure 3b, the median effects are highly correlated across different subsets (average correlation: 0.67), demonstrating the stability of our method. Additional experiments using ICA-LiMVAM-ML are reported in Appendix H.2.2.

## 6.3 REAL DATA EXPERIMENTS WITH FMRI

We also evaluated PairwiseLiMVAM on an fMRI rhyming judgment task dataset (Ramsey et al., 2010). The data consist of recordings from nine participants, each represented by nine variables: one task regressor (Input I), obtained by convolving the boxcar design of the rhyming task with a canonical hemodynamic response function, and eight regional time series from bilateral regions of interest. The dataset was acquired on a 3T scanner with a repetition time of 2 s, yielding 160 samples per subject, and is available via the OpenfMRI project (preprocessed version from [1]).

We report the ten strongest effects from the element-wise median of the estimated adjacency matrices in Figure 3c. The recovered structure is consistent with established findings for this task. In particular, we find a strong edge from the task regressor to the left occipital cortex (Input $\rightarrow$ LOCC), reflecting the expected visual drive of the stimuli, and prominent connections from left to right homologous regions (*e.g.* LOCC $\rightarrow$ ROCC, LIFG $\rightarrow$ RIFG, LIPL $\rightarrow$ RIPL), in line with the left-hemisphere dominance and inter-hemispheric influences reported by Ramsey et al. (2010). Edges such as LOCC $\rightarrow$ LACC and connections between LACC and RACC also match relationships highlighted in their analyses.

## 7 DISCUSSION AND RELATED WORK

A multi-view causal model based on LiNGAM was considered by Shimizu (2012), but their model was very different. The fundamental difference is that we assume the disturbances have some shared information across views and can be Gaussian, whereas Shimizu (2012) estimates disturbances that are independent across views and necessarily non-Gaussian. Thus, they only showed how to use the multiple views to improve the estimation while still assuming non-Gaussianity of the disturbances.

Chen et al. (2024) considered multi-view SEM using a special case of our shared disturbances model, which is in its turn a special case of our general LiMVAM model. Algorithmically, there is little difference between their work and our ICA-LiMVAM. However, their identifiability proofs are not available nor did they consider the identifiability of the causal ordering, while we do both. Moreover, as we already argued in Section 5, their model formulation has the serious flaw that it forces an arbitrary scaling of the data variables. Crucially, we propose new algorithms, PairwiseLiMVAM and DirectLiMVAM, that use only SOS; they are shown to be empirically superior to the ICA-LiMVAM in most cases.

A related very general framework, allowing for various kinds of interventions, was proposed by Mooij et al. (2020). On the other hand, a related "multi-context" approach, where the disturbances are not necessarily related at all while the DAGs are, was proposed by Sturma et al. (2023).

## 8 CONCLUSION

We propose a new approach for causal discovery in the multi-view linear-DAG case, improving the earlier work by Shimizu (2012); Chen et al. (2024). We show that the model is fully identifiable by second-order statistics (covariance structure) only, and under much weaker conditions than by Chen et al. (2024). We develop algorithms that either use the SOS only or combine them with non-Gaussian statistics. In particular, we adapt the Pairwise and Direct approaches to the multi-view case, resulting in highly efficient estimation. Results on brain imaging data are promising.

---

[1]`https://github.com/cabal-cmu/Feedback-Discovery`

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

APPENDIX

The appendix is organized as follows.

In Section A, we rigorously restate known identifiability results for LiNGAM.

In Section B, we prove the consistency of the likelihood-based criterion for a pair of variables.

In Section C, we prove the consistency of the cross-covariance-based criterion for a pair of variables.

In Section D , we extend the consistency of these criteria for a pair of variables to *many* pairs, and show that the LiMVAM model is identifiable.

In Section E , we show how to implement algorithms used to identify the causal graph.

In section F, we prove the identifiability of a special case of LiMVAM with shared disturbances.

In section G , we present an algorithm for estimating the causal graph in this special case.

In section H , we detail our experiments on real and synthetic data.

## A LINGAM

**Defining a domain connecting ICA and SEM parameters**   In the following proofs, we define the set

$$\mathcal{W} = \{\boldsymbol{W} \in \mathbb{R}^{p \times p} \mid \text{there exist a diagonal matrix } \boldsymbol{D} \text{ with positive diagonal entries}$$
$$\text{and a DAG matrix } \boldsymbol{B}, \text{ such that } \boldsymbol{W} = \boldsymbol{D}^{-1}(\boldsymbol{I} - \boldsymbol{B})\} \tag{10}$$

as the domain of the "unmixing matrices" $\boldsymbol{W} = \boldsymbol{A}^{-1}$ in the ICA model from Eq. 2, that are compatible with the DAG structure in the structural equation model (SEM) from Eq. 1. In general, unmixing matrices are not constrained in the ICA model, except by invertibility (and possibly by normalizing their rows). However, in the context of SEM estimation, the requirement that $\boldsymbol{W} = \boldsymbol{D}^{-1}(\boldsymbol{I} - \boldsymbol{B})$ belongs to the set $\mathcal{W}$ is a key structural constraint. Furthermore, it should be noted that finding $\boldsymbol{W}$ is equivalent to finding $\boldsymbol{D}$ and $\boldsymbol{B}$, since they are related by $\boldsymbol{B} = \boldsymbol{D}^{-1}(\boldsymbol{I} - \boldsymbol{W})$.

Finally, we will use the fact that any DAG matrix $\boldsymbol{B}$ can be decomposed into $\boldsymbol{B} = \boldsymbol{P}^\top \boldsymbol{T} \boldsymbol{P}$, where $\boldsymbol{P}$ is a permutation matrix and $\boldsymbol{T}$ a strictly lower triangular matrix.

### A.1   TECHNICAL LEMMAS

The following two lemmas are about matrices in the context of DAGs. They are used in the proofs of Theorems 5, 2, and 8, and are proven in Appendix A.3.

The basic identifiability results for the causal matrices $\boldsymbol{B}^i$ use the following:

**Lemma 3** *Let* $\boldsymbol{W} = \boldsymbol{D}^{-1}(\boldsymbol{I} - \boldsymbol{B})$ *and* $\boldsymbol{W}' = \boldsymbol{D}'^{-1}(\boldsymbol{I} - \boldsymbol{B}')$ *be two matrices that belong to* $\mathcal{W}$ *(defined in Eq. 10), and let* $\boldsymbol{Q}$ *be a sign-permutation matrix. Then*

$$\boldsymbol{W}' = \boldsymbol{Q}^\top \boldsymbol{W} \implies \boldsymbol{Q} = \boldsymbol{I}, \; \boldsymbol{D}' = \boldsymbol{D}, \; and \; \boldsymbol{B}' = \boldsymbol{B}. \tag{11}$$

On the other hand, the identifiability results for the causal ordering $\boldsymbol{P}$ (or causal orderings $\boldsymbol{P}^i$) use the following:

**Lemma 4** *Let* $\boldsymbol{P}$ *and* $\boldsymbol{P}'$ *be permutation matrices and* $\boldsymbol{T}$ *and* $\boldsymbol{T}'$ *be strictly lower triangular matrices, such that* $\boldsymbol{P}^\top \boldsymbol{T} \boldsymbol{P} = \boldsymbol{P}'^\top \boldsymbol{T}' \boldsymbol{P}'$. *Assume that* $\boldsymbol{T}$ *contains only non-zero elements in its strictly lower triangular part. Then,* $\boldsymbol{P} = \boldsymbol{P}'$ *and* $\boldsymbol{T} = \boldsymbol{T}'$.

### A.2   IDENTIFIABILITY OF LINGAM

As a special case, we restate here the identifiability of the single-view LiNGAM model (in terms of the matrix $\boldsymbol{B}$).

**Theorem 5** (Identifiability of LiNGAM) *In the statistical model defined by Eq. 1, the parameter* $\boldsymbol{B}$ *is identifiable, provided that the entries in* $\boldsymbol{s}$ *are mutually independent, that at most one of them is Gaussian, and that* $\boldsymbol{B}$ *is a DAG.*

This proof is based on the one from Shimizu et al. (2006), which we attempt to make a bit more rigorous. Identifiability was also shown in Shimizu (2022, Section 2.3) in the case with 2 components.

Let us transform the SEM

$$\boldsymbol{x} = \boldsymbol{B}\boldsymbol{x} + \boldsymbol{s} \tag{12}$$

into an ICA model

$$\boldsymbol{x} = \boldsymbol{A}\boldsymbol{s} \tag{13}$$

where the mixing matrix $\boldsymbol{A}$ is constrained as $\boldsymbol{A}^{-1} = \boldsymbol{I} - \boldsymbol{B}$, and where $\boldsymbol{B}$ is a DAG. The assumptions on $\boldsymbol{s}$ are identical in SEM and ICA. Let us parameterize by $\boldsymbol{W} = \boldsymbol{A}^{-1} = \boldsymbol{I} - \boldsymbol{B}$, instead of $\boldsymbol{B}$.

Consider two matrices $\boldsymbol{W} = \boldsymbol{I} - \boldsymbol{B}$ and $\boldsymbol{W}' = \boldsymbol{I} - \boldsymbol{B}'$, with $\boldsymbol{B}$ and $\boldsymbol{B}'$ being DAG matrices, that parameterize the same statistical model in Eq. 12. We have $\det(\boldsymbol{W}) = \det(\boldsymbol{W}') = 1$, so $\boldsymbol{W}$ and $\boldsymbol{W}'$ are invertible, which makes them valid unmixing matrices for the same ICA model in

Eq. 13. So, from the identifiability theory of ICA Comon (1994), we know that there exist a sign-permutation matrix $\boldsymbol{Q}$ and a scaling matrix $\boldsymbol{D}$ (*i.e.* a diagonal matrix with positive entries on the diagonal) such that

$$\boldsymbol{W}' = \boldsymbol{D}\boldsymbol{Q}^{\top}\boldsymbol{W} \tag{14}$$

hence

$$\boldsymbol{W}'' = \boldsymbol{Q}^{\top}\boldsymbol{W} \tag{15}$$

where we defined $\boldsymbol{W}'' = \boldsymbol{D}^{-1}\boldsymbol{W}'$. We thus have that $\boldsymbol{W}'' = \boldsymbol{D}^{-1}(\boldsymbol{I} - \boldsymbol{B}')$ and $\boldsymbol{W} = \boldsymbol{I} - \boldsymbol{B}$ both belong to the set $\mathcal{W}$, so applying Lemma 3 to $\boldsymbol{W}$ and $\boldsymbol{W}''$ imposes $\boldsymbol{Q} = \boldsymbol{I}$, $\boldsymbol{D} = \boldsymbol{I}$, and $\boldsymbol{B}' = \boldsymbol{B}$. This makes $\boldsymbol{B}$ fully identifiable ("fully" identifiable is in contrast to conventional ICA theory where some indeterminacies always remain), which concludes the proof.

### A.3 PROOFS OF TECHNICAL LEMMAS

#### A.3.1 PROOF OF LEMMA 3

Consider $\boldsymbol{W}, \boldsymbol{W}' \in \mathcal{W}$ and $\boldsymbol{Q}$ a sign-permutation matrix. Suppose that

$$\boldsymbol{W}' = \boldsymbol{Q}^{\top}\boldsymbol{W} \ . \tag{16}$$

**A permutation inequality**  Using the definition of $\mathcal{W}$ in Eq. 10 and the decompositions of DAG matrices, we have

$$\boldsymbol{W} = \boldsymbol{D}^{-1}\boldsymbol{P}^{\top}(\boldsymbol{I} - \boldsymbol{T})\boldsymbol{P} \ , \quad \boldsymbol{W}' = \boldsymbol{D}'^{-1}\boldsymbol{P}'^{\top}(\boldsymbol{I} - \boldsymbol{T}')\boldsymbol{P}' \tag{17}$$

where $\boldsymbol{D}$, $\boldsymbol{D}'$ are diagonal matrices with positive entries on the diagonal, $\boldsymbol{P}, \boldsymbol{P}'$ are permutation matrices, and $\boldsymbol{T}, \boldsymbol{T}'$ are strictly lower triangular matrices. Denote $\boldsymbol{L} = \boldsymbol{I} - \boldsymbol{T}$ and $\boldsymbol{L}' = \boldsymbol{I} - \boldsymbol{T}'$; these are lower triangular matrices that have unit diagonals. We plug the decompositions into Eq. 16, and get

$$\boldsymbol{D}'^{-1}\boldsymbol{P}'^{\top}\boldsymbol{L}'\boldsymbol{P}' = \boldsymbol{Q}^{\top}\boldsymbol{D}^{-1}\boldsymbol{P}^{\top}\boldsymbol{L}\boldsymbol{P} \ . \tag{18}$$

To exploit the structure of the lower triangular matrices $\boldsymbol{L}$ and $\boldsymbol{L}'$ and show how they constrain $\boldsymbol{Q}$, we now switch notations from permutation — or sign-permutation — matrices ($\boldsymbol{P}, \boldsymbol{P}', \boldsymbol{Q}$), to their corresponding permutation functions ($\phi, \phi', \psi$). Eq. 18 thus yields

$$\frac{\boldsymbol{L}'_{\phi'(i),\phi'(j)}}{\boldsymbol{D}'_{ii}} = \pm \frac{\boldsymbol{L}_{\phi(\psi(i)),\phi(j)}}{\boldsymbol{D}_{\psi(i),\psi(i)}} \ , \quad \forall i,j \in [\![1,p]\!] \tag{19}$$

where the $\pm$ symbol comes from $\boldsymbol{Q}$. In particular, $\boldsymbol{L}'$ has unit diagonal, so

$$\boldsymbol{L}_{\phi(\psi(i)),\phi(i)} = \pm \frac{\boldsymbol{D}_{\psi(i),\psi(i)}}{\boldsymbol{D}'_{ii}} \neq 0 \ , \quad \forall i \in [\![1,p]\!] \tag{20}$$

and $\boldsymbol{L}$ is lower triangular, so that its non-zero entries must satisfy

$$\phi(i) \leq \phi(\psi(i)) \ , \quad \forall i \in [\![1,p]\!] \ . \tag{21}$$

More generally, we can replace $i$ with $\psi(i)$, and so on, and obtain

$$\phi(i) \leq \phi(\psi(i)) \leq \phi(\psi^{(2)}(i)) \leq \dots \quad \forall i \in [\![1,p]\!] \tag{22}$$

where the superscript denotes composition and where we can apply the permutation $\psi$ an arbitrary number of times.

**$\boldsymbol{Q}$ must be a sign matrix**  Suppose that $\psi$ is not the identity. We can pick an index $k \in [\![1,p]\!]$ where $k \neq \psi(k)$. Because $\phi$ and $\psi$ are injective, we can apply them to the inequality any number of times $n \in \mathbb{N}$ and get

$$\phi(\psi^{(n)}(k)) \neq \phi(\psi^{(n+1)}(k)) \tag{23}$$

which together with Eq. 22 implies

$$\phi(k) < \phi(\psi(k)) < \phi(\psi^{(2)}(k)) < \dots \tag{24}$$

which can be applied any number of times. However, each inequality increases the index by at least one, while the index cannot go above $p$. Thus, applying the chain at least $p$ times, we have a contradiction. So, the permutation $\psi$ in $\boldsymbol{Q}$ must be the identity, and $\boldsymbol{Q}$ boils down to a sign matrix, *i.e.* a diagonal matrix with $1$ and $-1$ on the diagonal.

**$Q$ must be the identity**  Since $Q^\top = Q$, Eq. 18 can be reformulated as

$$P'^\top L' P' = D'QD^{-1}P^\top LP \tag{25}$$

where we now know that $D'QD^{-1}$ is a diagonal matrix. The matrix $P'^\top L' P'$ has a diagonal of ones and so too does $P^\top LP$. It follows that $D'QD^{-1} = I$. Since $D$ and $D'$ have positive diagonal entries, it follows that the entries of $Q$ cannot equal $-1$. So, $Q = I$, and thus $D' = D$. This concludes the proof of the Lemma.

### A.3.2  PROOF OF LEMMA 4

From the equality $P^\top TP = P'^\top T'P'$, we deduce that

$$T = \tilde{P}^\top T' \tilde{P} \tag{26}$$

where $\tilde{P} = P'P^\top$ is the permutation matrix represented by function $\phi$, which is defined such that: $\forall k \in [\![1, p]\!]$, $\tilde{P}_{k,\phi(k)} = 1$ and $\forall l \neq \phi(k)$, $\tilde{P}_{k,l} = 0$. Using the notation $\phi$ instead of $\tilde{P}$, Eq. 26 can be rewritten as, $\forall k, l \in [\![1, p]\!]$,

$$T_{k,l} = T'_{\phi(k),\phi(l)} \ . \tag{27}$$

We proceed by a proof by contradiction: assume that $\tilde{P} \neq I$. As a consequence, there exists an index $k$ such that $\phi(k) \neq k$. Let us fix such an index as $k$. We can assume without loss of generality that

$$\phi(k) > k \tag{28}$$

otherwise we just have to invert signs and switch rows and columns in the following. By assumption, the strictly lower triangular part of $T$ only has non-zero elements, so we have

$$T_{\phi(k),k} \neq 0 \ . \tag{29}$$

Yet, from Eq. 27, we know that $T_{\phi(k),k} = T'_{\phi(\phi(k)),\phi(k)}$ and all the non-zero elements of $T'$ are in its strictly lower triangular part, which implies

$$\phi(\phi(k)) > \phi(k) \ . \tag{30}$$

This logic can be applied repeatedly as in the preceding lemma's proof, and we see that

$$\phi(k) < \phi^{(2)}(k) < \phi^{(3)}(k) < \dots \ . \tag{31}$$

Now, this leads to an infinite strictly increasing sequence, which is contradictory since the index cannot grow greater than $p$. So, $\tilde{P} = I$, which means that $P'P^\top P = P$, and thus, using the orthogonality of $P$,

$$P' = P \ . \tag{32}$$

It also follows that $T' = T$, which concludes the proof of Lemma 4.

## B LIKELIHOOD-BASED CRITERION

### B.1 BASIC SETTING

The fundamental question here is to choose the causal direction for two variables, having many views $i$. We denote the two variables by $x$ and $y$ to avoid too many indices. It could be that the causal direction is $x \to y$:

$$y^i = b_i x^i + e^i \qquad \text{for all } i \tag{33}$$

or $y \to x$:

$$x^i = c_i y^i + d^i \qquad \text{for all } i \ . \tag{34}$$

Recall that superscripts are for view in the case of the random variables; however, for the regression coefficients we use subscripts because we need to take squares of such quantities. For the direction $x \to y$, collect the regressor and residuals in the vectors $\boldsymbol{x} = (x^1, \ldots, x^m)^\top$ and $\boldsymbol{e} = (e^1, \ldots, e^m)^\top$, respectively; and likewise for $\boldsymbol{y}$ and $\boldsymbol{d}$ in the opposite direction. Second-order moment matrices are denoted by $\boldsymbol{\Sigma}_x = \mathbb{E}[\boldsymbol{x}\boldsymbol{x}^\top]$, $\boldsymbol{\Sigma}_y = \mathbb{E}[\boldsymbol{y}\boldsymbol{y}^\top]$, and likewise for $\boldsymbol{d}$ and $\boldsymbol{e}$. Note that the independence between $\boldsymbol{x}$ and $\boldsymbol{e}$ and between $\boldsymbol{y}$ and $\boldsymbol{d}$, and the fact that $\boldsymbol{x}$ and $\boldsymbol{y}$ are standardized, imply that

$$b_i = c_i = \text{cov}(x_i, y_i) \qquad \text{for all } i \ . \tag{35}$$

### B.2 DERIVATION OF THE EXPECTED LOG-LIKELIHOOD IN ONE DIRECTION

We start by formulating the likelihood of the model in the case where disturbances are assumed to be Gaussian.

Consider the log-likelihood of $(\boldsymbol{x}, \boldsymbol{y})$ for the direction $x \to y$. Under the model, the prior (marginal) distribution of $\boldsymbol{x}$ is $\mathcal{N}(\mathbf{0}, \boldsymbol{\Sigma}_x)$, and the prior distribution of $\boldsymbol{e} = \boldsymbol{y} - \boldsymbol{b} \odot \boldsymbol{x}$ is $\mathcal{N}(\mathbf{0}, \boldsymbol{\Sigma}_e)$ — where $\odot$ denotes the elementwise product. Thus, we have

$$\begin{aligned}
\log p(\boldsymbol{x}, \boldsymbol{y}; \boldsymbol{b}, \boldsymbol{\Sigma}_x, \boldsymbol{\Sigma}_e, x \to y) &= \log p(\boldsymbol{y}|\boldsymbol{x}; \boldsymbol{b}, \boldsymbol{\Sigma}_x, \boldsymbol{\Sigma}_e, x \to y) + \log p(\boldsymbol{x}; \boldsymbol{b}, x \to y) \\
&= -(\boldsymbol{y} - \boldsymbol{b} \odot \boldsymbol{x})^T \boldsymbol{\Sigma}_e^{-1} (\boldsymbol{y} - \boldsymbol{b} \odot \boldsymbol{x}) - \log|\boldsymbol{\Sigma}_e| - \boldsymbol{x}^T \boldsymbol{\Sigma}_x^{-1} \boldsymbol{x} - \log|\boldsymbol{\Sigma}_x| \\
&= -\text{tr}\left(\boldsymbol{\Sigma}_e^{-1}(\boldsymbol{y} - \boldsymbol{b} \odot \boldsymbol{x})(\boldsymbol{y} - \boldsymbol{b} \odot \boldsymbol{x})^\top\right) - \log|\boldsymbol{\Sigma}_e| - \text{tr}\left(\boldsymbol{\Sigma}_x^{-1} \boldsymbol{x}\boldsymbol{x}^\top\right) - \log|\boldsymbol{\Sigma}_x|
\end{aligned} \tag{36}$$

where $\text{tr}$ is the trace operator, and where we neglect the part of the normalization constant that does not depend on any parameters.

The expected log-likelihood is thus

$$\mathbb{E}[\log p(\boldsymbol{x}, \boldsymbol{y}; \boldsymbol{b}, \boldsymbol{\Sigma}_x, \boldsymbol{\Sigma}_e, x \to y)] = -\text{tr}\left(\boldsymbol{\Sigma}_e^{-1} \mathbb{E}[(\boldsymbol{y} - \boldsymbol{b} \odot \boldsymbol{x})(\boldsymbol{y} - \boldsymbol{b} \odot \boldsymbol{x})^\top]\right) - \log|\boldsymbol{\Sigma}_e| \tag{37}$$

$$- \text{tr}\left(\boldsymbol{\Sigma}_x^{-1} \mathbb{E}[\boldsymbol{x}\boldsymbol{x}^\top]\right) - \log|\boldsymbol{\Sigma}_x| \tag{38}$$

$$= -\log|\boldsymbol{\Sigma}_x| - \log|\boldsymbol{\Sigma}_e| - 2m \ . \tag{39}$$

Denote by $\mathcal{L}$ this expected log-likelihood. We have (up to constants):

$$\mathcal{L}(\boldsymbol{\Sigma}_x, \boldsymbol{\Sigma}_e; x \to y) = -\log|\boldsymbol{\Sigma}_x| - \log|\boldsymbol{\Sigma}_e| \ . \tag{40}$$

### B.3 DERIVATION OF THE LIKELIHOOD-BASED CRITERION

We now derive a criterion that can be used to find the causal direction.

The expected log-likelihood of the model for the direction $y \to x$ can be obtained by switching the roles of $x$ and $y$ in Eq. 40. Thus, we have

$$\mathcal{L}(\boldsymbol{\Sigma}_y, \boldsymbol{\Sigma}_d; y \to x) = -\log|\boldsymbol{\Sigma}_y| - \log|\boldsymbol{\Sigma}_d| \tag{41}$$

where we recall that $\boldsymbol{\Sigma}_y = \mathbb{E}[\boldsymbol{y}\boldsymbol{y}^\top]$ and $\boldsymbol{\Sigma}_d = \mathbb{E}[\boldsymbol{d}\boldsymbol{d}^\top]$.

The correct direction can then be found by comparing the expected log-likelihoods of both directions. In particular, we calculate the difference between the two log-likelihoods, which can be reformulated as a likelihood ratio (LR):

$$\text{LR} := \mathcal{L}(\boldsymbol{\Sigma}_x, \boldsymbol{\Sigma}_e; x \to y) - \mathcal{L}(\boldsymbol{\Sigma}_y, \boldsymbol{\Sigma}_d; y \to x) = \mathbb{E}\left[\log \frac{p(\boldsymbol{x}, \boldsymbol{y}; \boldsymbol{b}, \boldsymbol{\Sigma}_x, \boldsymbol{\Sigma}_e, x \to y)}{p(\boldsymbol{x}, \boldsymbol{y}; \boldsymbol{c}, \boldsymbol{\Sigma}_y, \boldsymbol{\Sigma}_d, y \to x)}\right] \ . \tag{42}$$

Using Eq. 40 and Eq. 41, the criterion LR becomes

$$\text{LR} = -\log|\boldsymbol{\Sigma}_x| - \log|\boldsymbol{\Sigma}_e| + \log|\boldsymbol{\Sigma}_y| + \log|\boldsymbol{\Sigma}_d| \ . \tag{43}$$

Intuitively, the criterion in Eq. 43 will be large if the $x^i$ are highly correlated (more than the $y^i$) over the views, since then the first determinant will be small. This should make intuitive sense since it means the cause is highly structured. Likewise, if the residuals for $x \to y$ are highly structured, that direction is more likely. In other words, the direction that exhibits more structure wins.

Surprisingly, in this formulation, there is actually no particular model for the dependencies of the disturbances: the likelihood should simply prefer dependent disturbances. It is important to note that the criterion makes sense even if the distribution is non-Gaussian, as will be proven below. We thus obtain a method that works for data of any distribution, but using only second-order statistics.

### B.4 NON-NEGATIVITY OF THE LIKELIHOOD-BASED CRITERION

In this section, we prove that the criterion in Eq. 43 is non-negative for the direction $x \to y$. Importantly, no assumption of Gaussianity needs to be made here.

Consider the direction $x \to y$, and let us write $\boldsymbol{B} := \text{diag}(b_1, \ldots, b_m)$. Using the matrix $\boldsymbol{B}$ rather than the vector $\boldsymbol{b}$ makes notations clearer in the proofs. In the following, we use interchangeably $\boldsymbol{b}$ and $\boldsymbol{B}$. The covariance matrix of $\boldsymbol{y} = \boldsymbol{B}\boldsymbol{x} + \boldsymbol{e}$ is

$$\boldsymbol{\Sigma}_y = \boldsymbol{B}\boldsymbol{\Sigma}_x\boldsymbol{B} + \boldsymbol{\Sigma}_e \ . \tag{44}$$

Given that $b_i = c_i$ for all $i$, we have $\boldsymbol{d} = \boldsymbol{x} - \boldsymbol{B}\boldsymbol{y} = \boldsymbol{x} - \boldsymbol{B}(\boldsymbol{B}\boldsymbol{x} + \boldsymbol{e}) = (\boldsymbol{I} - \boldsymbol{B})^2\boldsymbol{x} - \boldsymbol{B}\boldsymbol{e}$. Thus, the covariance matrix of $\boldsymbol{d}$ is

$$\boldsymbol{\Sigma}_d = (\boldsymbol{I} - \boldsymbol{B})^2\boldsymbol{\Sigma}_x(\boldsymbol{I} - \boldsymbol{B})^2 + \boldsymbol{B}\boldsymbol{\Sigma}_e\boldsymbol{B} \ . \tag{45}$$

Using these expressions, the criterion in Eq. 43 becomes

$$\text{LR} = -\log|\boldsymbol{\Sigma}_x| - \log|\boldsymbol{\Sigma}_e| + \log|\boldsymbol{B}\boldsymbol{\Sigma}_x\boldsymbol{B} + \boldsymbol{\Sigma}_e| + \log\left|(\boldsymbol{I} - \boldsymbol{B})^2\boldsymbol{\Sigma}_x(\boldsymbol{I} - \boldsymbol{B})^2 + \boldsymbol{B}\boldsymbol{\Sigma}_e\boldsymbol{B}\right| \ . \tag{46}$$

Collect the variances of the residuals $\boldsymbol{e}$ in the diagonal matrix

$$\boldsymbol{L} := \text{diag}\left(\sqrt{\text{var}(e^1)}, \ldots, \sqrt{\text{var}(e^m)}\right) \tag{47}$$

and define the correlation matrix of the residuals as $\widetilde{\boldsymbol{\Sigma}}_e$, so that

$$\boldsymbol{\Sigma}_e = \boldsymbol{L}\widetilde{\boldsymbol{\Sigma}}_e\boldsymbol{L} \ . \tag{48}$$

Now, the objective becomes

$$\text{LR} = -\log|\boldsymbol{\Sigma}_x| - \log|\widetilde{\boldsymbol{\Sigma}}_e| - \log|\boldsymbol{L}^2| + \log\left|\boldsymbol{B}\boldsymbol{\Sigma}_x\boldsymbol{B} + \boldsymbol{L}\widetilde{\boldsymbol{\Sigma}}_e\boldsymbol{L}\right|$$
$$+ \log\left|(\boldsymbol{I} - \boldsymbol{B})^2\boldsymbol{\Sigma}_x(\boldsymbol{I} - \boldsymbol{B})^2 + \boldsymbol{B}\boldsymbol{L}\widetilde{\boldsymbol{\Sigma}}_e\boldsymbol{L}\boldsymbol{B}\right| \ . \tag{49}$$

Importantly, the fact that $x$ and $y$ are standardized means that $\boldsymbol{\Sigma}_x$ and $\boldsymbol{\Sigma}_y$ have unit diagonal, so Eq. 44 implies

$$\boldsymbol{L}^2 + \boldsymbol{B}^2 = \boldsymbol{I} \tag{50}$$

and it follows that the diagonal entries of $\boldsymbol{L}$ and $\boldsymbol{B}$ are in $[-1, 1]$. Thus, we have

$$\text{LR} = -\log|\boldsymbol{\Sigma}_x| - \log|\widetilde{\boldsymbol{\Sigma}}_e| - \log|\boldsymbol{L}^2| + \log\left|\boldsymbol{B}\boldsymbol{\Sigma}_x\boldsymbol{B} + \boldsymbol{L}\widetilde{\boldsymbol{\Sigma}}_e\boldsymbol{L}\right| + \log\left|\boldsymbol{L}^2\boldsymbol{\Sigma}_x\boldsymbol{L}^2 + \boldsymbol{L}\boldsymbol{B}\widetilde{\boldsymbol{\Sigma}}_e\boldsymbol{B}\boldsymbol{L}\right| \tag{51}$$

$$= -\log|\boldsymbol{\Sigma}_x| - \log|\widetilde{\boldsymbol{\Sigma}}_e| + \log\left|\boldsymbol{B}\boldsymbol{\Sigma}_x\boldsymbol{B} + \boldsymbol{L}\widetilde{\boldsymbol{\Sigma}}_e\boldsymbol{L}\right| + \log\left|\boldsymbol{L}\boldsymbol{\Sigma}_x\boldsymbol{L} + \boldsymbol{B}\widetilde{\boldsymbol{\Sigma}}_e\boldsymbol{B}\right| \tag{52}$$

where $\boldsymbol{L}$ and $\boldsymbol{B}$ commuted in the first equality because they are diagonal.

The following lemma (proven in Appendix B.5) shows that this criterion is non-negative, and even positive under some assumptions.

**Lemma 6** *Let $\Sigma_x$ and $\widetilde{\Sigma}_e$ be $m \times m$ real symmetric positive definite matrices with unit diagonal entries. Let $\boldsymbol{B} = \mathrm{diag}(b_1, \ldots, b_m)$ and $\boldsymbol{L} = \mathrm{diag}(l_1, \ldots, l_m)$ be diagonal matrices with entries in $[-1, 1]$, satisfying $\boldsymbol{B}^2 + \boldsymbol{L}^2 = \boldsymbol{I}$ (i.e. $b_i^2 + l_i^2 = 1$ for all i). Define the function*

$$J(\boldsymbol{B}, \boldsymbol{L}) := -\log \det \Sigma_x - \log \det \widetilde{\Sigma}_e + \log \det \left( \boldsymbol{B}\Sigma_x\boldsymbol{B} + \boldsymbol{L}\widetilde{\Sigma}_e\boldsymbol{L} \right) + \log \det \left( \boldsymbol{L}\Sigma_x\boldsymbol{L} + \boldsymbol{B}\widetilde{\Sigma}_e\boldsymbol{B} \right) . \tag{53}$$

*Under the above assumptions, we have $J(\boldsymbol{B}, \boldsymbol{L}) \geq 0$. Moreover, $J(\boldsymbol{B}, \boldsymbol{L}) = 0$ if and only if*

$$\boldsymbol{L}\,\widetilde{\Sigma}_e\,\boldsymbol{B} \;=\; \boldsymbol{B}\,\Sigma_x\,\boldsymbol{L} \; . \tag{54}$$

B.5  PROOF OF A USEFUL LEMMA

Let us prove Lemma 6.

**Step 1: define $A$ and $C$ —** Define

$$\boldsymbol{A} := \boldsymbol{B}\Sigma_x\boldsymbol{B} + \boldsymbol{L}\widetilde{\Sigma}_e\boldsymbol{L} \qquad \text{and} \qquad \boldsymbol{C} := \boldsymbol{L}\Sigma_x\boldsymbol{L} + \boldsymbol{B}\widetilde{\Sigma}_e\boldsymbol{B} \; . \tag{55}$$

We have

$$J(\boldsymbol{B}, \boldsymbol{L}) := -\log \det \Sigma_x - \log \det \widetilde{\Sigma}_e + \log \det \boldsymbol{A} + \log \det \boldsymbol{C} \; . \tag{56}$$

**Step 2: define $M$ —** Define the $2m \times 2m$ block matrix

$$\boldsymbol{M} := \begin{pmatrix} \boldsymbol{B} & \boldsymbol{L} \\ -\boldsymbol{L} & \boldsymbol{B} \end{pmatrix} \; . \tag{57}$$

Since $\boldsymbol{B}$ and $\boldsymbol{L}$ are diagonal, they commute, hence $\boldsymbol{B}\boldsymbol{L} = \boldsymbol{L}\boldsymbol{B}$. Using $\boldsymbol{B}^2 + \boldsymbol{L}^2 = \boldsymbol{I}$, we compute

$$\boldsymbol{M}\boldsymbol{M}^\top = \begin{pmatrix} \boldsymbol{B}\boldsymbol{B} + \boldsymbol{L}\boldsymbol{L} & -\boldsymbol{B}\boldsymbol{L} + \boldsymbol{L}\boldsymbol{B} \\ -\boldsymbol{L}\boldsymbol{B} + \boldsymbol{B}\boldsymbol{L} & \boldsymbol{L}\boldsymbol{L} + \boldsymbol{B}\boldsymbol{B} \end{pmatrix} = \begin{pmatrix} \boldsymbol{I} & \boldsymbol{0} \\ \boldsymbol{0} & \boldsymbol{I} \end{pmatrix} = \boldsymbol{I}_{2m} \; . \tag{58}$$

Thus $\boldsymbol{M}$ is orthonormal, so $\det \boldsymbol{M} = \pm 1$.

**Step 3: define $\mathcal{B}$ —** Consider the block-diagonal matrix $\mathrm{diag}(\Sigma_x, \widetilde{\Sigma}_e)$ and conjugate by $\boldsymbol{M}$:

$$\mathcal{B} := \boldsymbol{M} \begin{pmatrix} \Sigma_x & \boldsymbol{0} \\ \boldsymbol{0} & \widetilde{\Sigma}_e \end{pmatrix} \boldsymbol{M}^\top = \begin{pmatrix} \boldsymbol{A} & \boldsymbol{S} \\ \boldsymbol{S}^\top & \boldsymbol{C} \end{pmatrix}, \qquad \boldsymbol{S} := \boldsymbol{L}\widetilde{\Sigma}_e\boldsymbol{B} - \boldsymbol{B}\Sigma_x\boldsymbol{L} \; . \tag{59}$$

Because $\boldsymbol{M}$ is orthonormal and $\Sigma_x, \widetilde{\Sigma}_e \succ \boldsymbol{0}$, the matrix $\mathcal{B}$ is symmetric positive definite (SPD). In particular, $\boldsymbol{A}$ and $\boldsymbol{C}$ are also SPD (as they are principal blocks of a SPD matrix) and thus $\boldsymbol{A}^{-1}$ exists.

**Step 4: Schur complement of $A$ in $\mathcal{B}$ —** For a symmetric block matrix $\begin{pmatrix} \boldsymbol{A} & \boldsymbol{S} \\ \boldsymbol{S}^\top & \boldsymbol{C} \end{pmatrix}$ with $\boldsymbol{A} \succ \boldsymbol{0}$, the Schur complement of $\boldsymbol{A}$ is $\boldsymbol{C} - \boldsymbol{S}^\top \boldsymbol{A}^{-1} \boldsymbol{S}$, and the Schur's formula yields

$$\det \begin{pmatrix} \boldsymbol{A} & \boldsymbol{S} \\ \boldsymbol{S}^\top & \boldsymbol{C} \end{pmatrix} = \det(\boldsymbol{A}) \det \left( \boldsymbol{C} - \boldsymbol{S}^\top \boldsymbol{A}^{-1} \boldsymbol{S} \right) \; . \tag{60}$$

Applying this to $\mathcal{B}$ and using multiplicativity of determinant and the fact that $\det \boldsymbol{M} = \pm 1$ gives

$$\det(\Sigma_x) \det\left( \widetilde{\Sigma}_e \right) = \det(\mathcal{B}) = \det(\boldsymbol{A}) \det \left( \boldsymbol{C} - \boldsymbol{S}^\top \boldsymbol{A}^{-1} \boldsymbol{S} \right) \; . \tag{61}$$

**Step 5: derive a matrix inequality —** Because $\mathcal{B} \succ \boldsymbol{0}$ and $\boldsymbol{A} \succ \boldsymbol{0}$, it follows from the Schur complement characterization of positive definiteness that $\boldsymbol{C} - \boldsymbol{S}^\top \boldsymbol{A}^{-1} \boldsymbol{S} \succ \boldsymbol{0}$. In particular, we have $\det\left( \boldsymbol{C} - \boldsymbol{S}^\top \boldsymbol{A}^{-1} \boldsymbol{S} \right) > 0$. Moreover, since $\boldsymbol{A} \succ \boldsymbol{0}$, we have $\boldsymbol{S}^\top \boldsymbol{A}^{-1} \boldsymbol{S} \succeq \boldsymbol{0}$, and thus $\boldsymbol{C} - \boldsymbol{S}^\top \boldsymbol{A}^{-1} \boldsymbol{S} \preceq \boldsymbol{C}$.

**Step 6: derive a determinant inequality —** If $\mathbf{0} \preceq \mathbf{X} \preceq \mathbf{Y}$ and $\mathbf{Y} \succ \mathbf{0}$, then conjugating by $\mathbf{Y}^{-1/2}$ shows $\mathbf{0} \preceq \mathbf{Y}^{-1/2}\mathbf{X}\mathbf{Y}^{-1/2} \preceq \mathbf{I}$. Thus, all eigenvalues of $\mathbf{Y}^{-1/2}\mathbf{X}\mathbf{Y}^{-1/2}$ lie in $[0,1]$, so their product satisfies $\det\big(\mathbf{Y}^{-1/2}\mathbf{X}\mathbf{Y}^{-1/2}\big) \leq 1$. Hence $\det(\mathbf{X}) \leq \det(\mathbf{Y})$. Applying this to $\mathbf{X} := \mathbf{C} - \mathbf{S}^\top \mathbf{A}^{-1}\mathbf{S}$ and $\mathbf{Y} := \mathbf{C}$ yields

$$\det\big(\mathbf{C} - \mathbf{S}^\top \mathbf{A}^{-1}\mathbf{S}\big) \leq \det(\mathbf{C}) \ . \tag{62}$$

**Step 7: $J(\mathbf{B},\mathbf{L}) \geq 0$ —** From Steps 4 and 6, we get

$$\det(\mathbf{\Sigma}_x)\det\big(\widetilde{\mathbf{\Sigma}}_e\big) = \det(\mathbf{A})\det\big(\mathbf{C} - \mathbf{S}^\top \mathbf{A}^{-1}\mathbf{S}\big) \leq \det(\mathbf{A})\det(\mathbf{C}) \ . \tag{63}$$

Taking natural logarithms and rearranging gives

$$\log\det(\mathbf{A}) + \log\det(\mathbf{C}) \ \geq \ \log\det(\mathbf{\Sigma}_x) + \log\det\big(\widetilde{\mathbf{\Sigma}}_e\big) \ , \tag{64}$$

which is precisely $J(\mathbf{B},\mathbf{L}) \geq 0$.

**Step 8: equality condition —** The equality $J(\mathbf{B},\mathbf{L}) = 0$ holds iff

$$\det\big(\mathbf{C} - \mathbf{S}^\top \mathbf{A}^{-1}\mathbf{S}\big) = \det(\mathbf{C}) \ . \tag{65}$$

Write $\mathbf{K} := \mathbf{C}^{-1/2}\big(\mathbf{S}^\top \mathbf{A}^{-1}\mathbf{S}\big)\mathbf{C}^{-1/2} \succeq \mathbf{0}$. We have

$$\det\big(\mathbf{C} - \mathbf{S}^\top \mathbf{A}^{-1}\mathbf{S}\big) = \det(\mathbf{C})\det\big(\mathbf{I} - \mathbf{C}^{-1}\mathbf{S}^\top \mathbf{A}^{-1}\mathbf{S}\big) \tag{66}$$

$$= \det(\mathbf{C})\det\big(\mathbf{C}^{-\frac{1}{2}}\mathbf{C}^{\frac{1}{2}} - \mathbf{C}^{-\frac{1}{2}}\mathbf{C}^{-\frac{1}{2}}\mathbf{S}^\top \mathbf{A}^{-1}\mathbf{S}\mathbf{C}^{-\frac{1}{2}}\mathbf{C}^{\frac{1}{2}}\big) \tag{67}$$

$$= \det(\mathbf{C})\det\big(\mathbf{C}^{-\frac{1}{2}}\big)\det(\mathbf{I} - \mathbf{K})\det\big(\mathbf{C}^{\frac{1}{2}}\big) \tag{68}$$

$$= \det(\mathbf{C})\det(\mathbf{I} - \mathbf{K}) \ . \tag{69}$$

Thus, the equality condition is $\det(\mathbf{I} - \mathbf{K}) = 1$. From Step 5, we know that $\mathbf{0} \preceq \mathbf{C} - \mathbf{S}^\top \mathbf{A}^{-1}\mathbf{S} \preceq \mathbf{C}$. Conjugating by $\mathbf{C}^{-\frac{1}{2}}$ gives $\mathbf{0} \preceq \mathbf{I} - \mathbf{K} \preceq \mathbf{I}$, so each eigenvalue $\lambda_i$ of $\mathbf{K}$ satisfies $0 \leq \lambda_i \leq 1$. The eigenvalues of $\mathbf{I} - \mathbf{K}$ are $1 - \lambda_i \in [0,1]$, hence

$$\prod_i (1 - \lambda_i) = \det(\mathbf{I} - \mathbf{K}) = 1 \ . \tag{70}$$

The equality thus holds iff $\lambda_i = 0$ for all $i$, hence $\mathbf{K} = \mathbf{0}$. Since $\mathbf{C}^{-\frac{1}{2}}, \mathbf{A}^{-1} \succ \mathbf{0}$, we get $\mathbf{S} = \mathbf{0}$. Conversely, $\mathbf{S} = \mathbf{0}$ clearly forces equality. Therefore, the equality occurs exactly when

$$\mathbf{S} = \mathbf{L}\widetilde{\mathbf{\Sigma}}_e\mathbf{B} - \mathbf{B}\mathbf{\Sigma}_x\mathbf{L} = \mathbf{0} \ , \tag{71}$$

*i.e.*

$$\forall i,j: \qquad b_i l_j (\mathbf{\Sigma}_x)_{ij} = l_i b_j (\widetilde{\mathbf{\Sigma}}_e)_{ij} \ . \tag{72}$$

This completes the proof.

### B.6 Conditions for strict positivity of the likelihood-based criterion

From now on, we assume that the model is not degenerate in the sense that disturbances have a positive variance, that is $l_i = \mathrm{var}(e_i) > 0$ for all $i$.

The criterion LR derived in Appendix B.4 is used to determine the causal direction from its sign, with LR $= 0$ indicating that the two directions cannot be distinguished. Lemma 6 shows that LR $= 0$ if and only if

$$\mathbf{L}\widetilde{\mathbf{\Sigma}}_e\mathbf{B} = \mathbf{B}\mathbf{\Sigma}_x\mathbf{L} \ . \tag{73}$$

Entrywise, this is equivalent to

$$\forall i,j: \qquad l_i b_j (\widetilde{\mathbf{\Sigma}}_e)_{ij} = b_i l_j (\mathbf{\Sigma}_x)_{ij} \ . \tag{74}$$

Let us fix $i,j \in [\![1,m]\!]$.

**Case 1: $b_i = 0$ and $b_j = 0$ —** The equality immediately holds.

**Case 2: $b_i = 0$ and $b_j \neq 0$ —** The right-hand side in Eq. 74 is equal to 0. Since $l_i = \text{var}(e^i) > 0$ and $b_j \neq 0$, we must have $(\widetilde{\boldsymbol{\Sigma}}_e)_{ij} = 0$.

**Case 3: $b_i \neq 0$ and $b_j = 0$ —** The left-hand side in Eq. 74 is equal to 0. Since $l_j = \text{var}(e^j) > 0$ and $b_i \neq 0$, we must have $(\boldsymbol{\Sigma}_x)_{ij} = 0$.

**Case 4: $b_i \neq 0$ and $b_j \neq 0$ —** Multiplying Eq. 74 by $\frac{l_j}{b_i b_j^2}$, and using $l_j^2 = 1 - b_j^2$, we obtain

$$\frac{l_i l_j}{b_i b_j}(\widetilde{\boldsymbol{\Sigma}}_e)_{ij} = \left(\frac{1}{b_j^2} - 1\right)(\boldsymbol{\Sigma}_x)_{ij} \ . \tag{75}$$

Switching the roles of $i$ and $j$, and using the symmetry of $(\widetilde{\boldsymbol{\Sigma}}_e)_{ij}$ and $(\boldsymbol{\Sigma}_x)_{ij}$, yield

$$\frac{l_i l_j}{b_i b_j}(\widetilde{\boldsymbol{\Sigma}}_e)_{ij} = \left(\frac{1}{b_i^2} - 1\right)(\boldsymbol{\Sigma}_x)_{ij} \ . \tag{76}$$

hence

$$b_i^2(\boldsymbol{\Sigma}_x)_{ij} = b_j^2 \ (\boldsymbol{\Sigma}_x)_{ij} \ . \tag{77}$$

Similarly, exchanging the roles of $(b_i, b_j, (\boldsymbol{\Sigma}_x)_{ij})$ and $(l_i, l_j, (\widetilde{\boldsymbol{\Sigma}}_e)_{ij})$ yields

$$l_i^2(\widetilde{\boldsymbol{\Sigma}}_e)_{ij} = l_j^2 \ (\widetilde{\boldsymbol{\Sigma}}_e)_{ij} \ . \tag{78}$$

Now suppose $(\boldsymbol{\Sigma}_x)_{ij} \neq 0$ or $(\widetilde{\boldsymbol{\Sigma}}_e)_{ij} \neq 0$. From Eq. 77 and Eq. 78 it follows that $b_i^2 = b_j^2$ or $l_i^2 = l_j^2$. Since $b_i^2 + l_i^2 = 1$, both equalities hold simultaneously:

$$|b_i| = |b_j| \qquad \text{and} \qquad l_i = l_j \tag{79}$$

where the asbolute value can be dropped for $l_i$ because $l_i = \text{var}(e_i) > 0$. Plugging these equalities into Eq. 75 yields

$$l_j^2(\widetilde{\boldsymbol{\Sigma}}_e)_{ij} = \pm(1 - b_j^2)(\boldsymbol{\Sigma}_x)_{ij} \tag{80}$$

hence

$$|(\widetilde{\boldsymbol{\Sigma}}_e)_{ij}| = |(\boldsymbol{\Sigma}_x)_{ij}| \ . \tag{81}$$

Moreover, for Eq. 74 to hold we must also have the sign equality

$$\text{sign}(b_i) \, \text{sign}(b_j) = \text{sign}((\boldsymbol{\Sigma}_x)_{ij}) \, \text{sign}((\widetilde{\boldsymbol{\Sigma}}_e)_{ij}) \ . \tag{82}$$

Altogether, either both $(\boldsymbol{\Sigma}_x)_{ij}$ and $(\widetilde{\boldsymbol{\Sigma}}_e)_{ij}$ vanish, or the following four conditions hold:

$$|(\widetilde{\boldsymbol{\Sigma}}_e)_{ij}| = |(\boldsymbol{\Sigma}_x)_{ij}| \neq 0 \tag{83}$$
$$|b_i| = |b_j| \tag{84}$$
$$l_i = l_j \tag{85}$$
$$\text{sign}(b_i) \, \text{sign}(b_j) = \text{sign}((\boldsymbol{\Sigma}_x)_{ij}) \, \text{sign}((\widetilde{\boldsymbol{\Sigma}}_e)_{ij}) \ . \tag{86}$$

Conversely, if $(\boldsymbol{\Sigma}_x)_{ij} = (\widetilde{\boldsymbol{\Sigma}}_e)_{ij} = 0$, or if conditions Eq. 83–Eq. 86 are satisfied, then Eq. 74 holds.

**Conclusion —** By contraposition, Eq. 73 is violated whenever there exists $(i, j)$ such that one of the following cases holds:

- $b_i = 0$, $b_j \neq 0$, and $(\widetilde{\boldsymbol{\Sigma}}_e)_{ij} \neq 0$
- $b_i \neq 0$, $b_j = 0$, and $(\boldsymbol{\Sigma}_x)_{ij} \neq 0$
- $b_i \neq 0$, $b_j \neq 0$, $(\boldsymbol{\Sigma}_x)_{ij} \neq 0$ or $(\widetilde{\boldsymbol{\Sigma}}_e)_{ij} \neq 0$, and at least one of the conditions Eq. 83–Eq. 86 is not met.

In particular, a simple sufficient condition is

$$\exists i, j: \qquad (b_i \neq 0 \qquad \text{or} \qquad b_j \neq 0) \qquad \text{and} \qquad |(\widetilde{\boldsymbol{\Sigma}}_e)_{ij}| \neq |(\boldsymbol{\Sigma}_x)_{ij}| \ . \tag{87}$$

## C  Cross-covariance-based criterion

Consider the setting described in Appendix B.1.

### C.1  Derivation of the cross-covariance-based criterion

A simple approach to infer the causal direction is to exploit the assumption that root variables and residuals are independent. Specifically, if the true direction is $x \to y$, then $\boldsymbol{x}$ and $\boldsymbol{e}$ are independent; if the true direction is $y \to x$, then $\boldsymbol{y}$ and $\boldsymbol{d}$ are independent. This implies that the cross-moment matrix $\mathbb{E}[\boldsymbol{e}\boldsymbol{x}^\top]$ should vanish when $x \to y$, while $\mathbb{E}[\boldsymbol{d}\boldsymbol{y}^\top]$ should vanish when $y \to x$. Comparing the Frobenius norms of these matrices therefore provides a natural criterion, based solely on cross-covariances:

$$\text{FC} := \left\| \mathbb{E}[\boldsymbol{d}\boldsymbol{y}^\top] \right\|_F - \left\| \mathbb{E}[\boldsymbol{e}\boldsymbol{x}^\top] \right\|_F \ . \tag{88}$$

Intuitively, if this criterion is positive we deduce that $x \to y$, and if negative we deduce that $y \to x$.

### C.2  Conditions for strict positivity of the cross-covariance criterion

Assume without loss of generality that the true causal direction is $x \to y$, and let us analyze the sign of the criterion FC.

Given that $x \to y$, the true model is

$$\boldsymbol{y} = \boldsymbol{B}\boldsymbol{x} + \boldsymbol{e} \tag{89}$$

where $\boldsymbol{B} = \text{diag}(\text{cov}(x^1, y^1), \dots, \text{cov}(x^m, y^m))$, and $\boldsymbol{e}$ is independent from $\boldsymbol{x}$. So, we have

$$\mathbb{E}[\boldsymbol{x}\boldsymbol{e}^\top] = \boldsymbol{0} \ . \tag{90}$$

In the wrong direction, the residuals take the form

$$\boldsymbol{d} = \boldsymbol{x} - \boldsymbol{C}\boldsymbol{y}, \qquad \boldsymbol{C} = \boldsymbol{B} \ . \tag{91}$$

The cross-covariance between observations $\boldsymbol{y}$ and residuals $\boldsymbol{d}$ is then

$$\mathbb{E}\big[\boldsymbol{y}\boldsymbol{d}^\top\big] = \mathbb{E}\big[\boldsymbol{y}(\boldsymbol{x} - \boldsymbol{B}\boldsymbol{y})^\top\big] \tag{92}$$

$$= \mathbb{E}\big[(\boldsymbol{B}\boldsymbol{x} + \boldsymbol{e})(\boldsymbol{x} - \boldsymbol{B}(\boldsymbol{B}\boldsymbol{x} + \boldsymbol{e}))^\top\big] \tag{93}$$

$$= \mathbb{E}\big[\boldsymbol{B}\boldsymbol{x}\boldsymbol{x}^\top(\boldsymbol{I} - \boldsymbol{B}^2)\big] + \mathbb{E}\big[\boldsymbol{e}\boldsymbol{x}^\top(\boldsymbol{I} - \boldsymbol{B}^2))\big] - \mathbb{E}\big[\boldsymbol{B}\boldsymbol{x}\boldsymbol{e}^\top\boldsymbol{B}\big] - \mathbb{E}\big[\boldsymbol{e}\boldsymbol{e}^\top\boldsymbol{B}\big] \tag{94}$$

$$= \boldsymbol{B}\,\boldsymbol{\Sigma}_x\,(\boldsymbol{I} - \boldsymbol{B}^2) - \boldsymbol{\Sigma}_e\,\boldsymbol{B} \tag{95}$$

where $\boldsymbol{\Sigma}_x = \mathbb{E}\big[\boldsymbol{x}\boldsymbol{x}^\top\big]$ and $\boldsymbol{\Sigma}_e = \mathbb{E}\big[\boldsymbol{e}\boldsymbol{e}^\top\big]$.

From Eq. 90 and the non-negativity of the Frobenius norm, we see that FC is always non-negative. Moreover, using Eq. 95, we deduce that FC $= 0$ if and only if

$$\boldsymbol{B}\,\boldsymbol{\Sigma}_x\,(\boldsymbol{I} - \boldsymbol{B}^2) - \boldsymbol{\Sigma}_e\,\boldsymbol{B} = \boldsymbol{0} \ . \tag{96}$$

Note that the fact that $l_i > 0$, for all $i$, makes $\boldsymbol{L}$ invertible. So, using the identity $\boldsymbol{I} - \boldsymbol{B}^2 = \boldsymbol{L}^2$, and multiplying on the right by $\boldsymbol{L}^{-1}$, we obtain

$$\boldsymbol{B}\,\boldsymbol{\Sigma}_x\,\boldsymbol{L} = \boldsymbol{\Sigma}_e\,\boldsymbol{B}\,\boldsymbol{L}^{-1} \tag{97}$$

hence

$$\boldsymbol{B}\,\boldsymbol{\Sigma}_x\,\boldsymbol{L} = \boldsymbol{L}\,\widetilde{\boldsymbol{\Sigma}}_e\,\boldsymbol{B} \tag{98}$$

since $\boldsymbol{\Sigma}_e = \boldsymbol{L}\,\widetilde{\boldsymbol{\Sigma}}_e\,\boldsymbol{L}$, and $\boldsymbol{B}$ and $\boldsymbol{L}$ commute.

This condition coincides exactly with Eq. 73. In particular, this shows that the likelihood-based criterion and the cross-covariance-based criterion share the same consistency assumptions.

# D IDENTIFIABILITY OF LIMVAM

## D.1 MULTIVARIATE CONDITIONS FOR STRICT POSITIVITY OF A CRITERION

The following assumption is a multivariate generalization of the assumption detailed in B.6 and C.2.

**Assumption 6** (Diversity of the views) *Let $j \neq j'$ denote any two component indices such that $\boldsymbol{x}_j \rightarrow \boldsymbol{x}_{j'}$. Then, there exist two views $i$ and $i'$ such that the three following conditions hold:*

    *1. $\mathrm{corr}(\boldsymbol{x}_j^i, \boldsymbol{x}_{j'}^i) \neq 0$ or $\mathrm{corr}(\boldsymbol{x}_j^{i'}, \boldsymbol{x}_{j'}^{i'}) \neq 0$*    *2. $\mathrm{corr}(\boldsymbol{x}_j^i, \boldsymbol{x}_j^{i'}) \neq 0$ or $\mathrm{corr}(\boldsymbol{e}_{j'}^i, \boldsymbol{e}_{j'}^{i'}) \neq 0$*

    *3. At least one of the four following inequalities is met:*

$$|\mathrm{corr}(\boldsymbol{x}_j^i, \boldsymbol{x}_j^{i'})| \neq |\mathrm{corr}(\boldsymbol{e}_{j'}^i, \boldsymbol{e}_{j'}^{i'})|, \ |\mathrm{corr}(\boldsymbol{x}_j^i, \boldsymbol{x}_{j'}^i)| \neq |\mathrm{corr}(\boldsymbol{x}_j^{i'}, \boldsymbol{x}_{j'}^{i'})|, \ \mathrm{var}(\boldsymbol{e}_{j'}^i) \neq \mathrm{var}(\boldsymbol{e}_{j'}^{i'})$$

$$\mathrm{sign}(\mathrm{corr}(\boldsymbol{x}_j^i, \boldsymbol{x}_{j'}^i)) \, \mathrm{sign}(\mathrm{corr}(\boldsymbol{x}_j^{i'}, \boldsymbol{x}_{j'}^{i'})) \neq \mathrm{sign}(\mathrm{corr}(\boldsymbol{x}_j^i, \boldsymbol{x}_j^{i'})) \, \mathrm{sign}(\mathrm{corr}(\boldsymbol{e}_{j'}^i, \boldsymbol{e}_{j'}^{i'}))$$

*where* corr *denotes the correlation coefficient and* sign *denotes the sign function.*

These conditions can only be met for indices $i \neq i'$, so they can be interpreted as a need for correlation and diversity across views. Because they may be hard to interpret at first glance, we next visualize in Figure 4 what these conditions mean for the causal graph.

In particular, Assumption 6 immediately implies its simpler version used in the main text, Assumption 1.

## D.2 AN EXAMPLE FOR INTERPRETING THE SOS-BASED IDENTIFIABILITY CONDITION IN ASSUMPTION 1

Consider one view and two variables denoted for notational simplicity $x$ and $y$. We have

$$y^1 = b^1 x^1 + e^1 \tag{99}$$

and assume for simplicity that $x^1$ and $y^1$ have been standardized.

Now, suppose we observe two datasets from this one view, denote them by $(x^1, y^1)$ and $(x^{1'}, y^{1'})$, which have identical distributions and are independent of each other. Crucially, all this data is from one view. However, we could artificially generate a new "view" by adding those two datasets together:

$$x^2 = x^1 + x^{1'} \tag{100}$$

and likewise for $y^2$ and $e^2$. Clearly the new data follows the SEM with the same $b$:

$$y^2 = b^1 x^2 + e^2 \tag{101}$$

since we are simply adding each of the terms separately in the two datasets.

But this "multi-view" data is degenerate. In fact, the correlation coefficients are

$$\mathrm{corr}(x^1, x^2) = \frac{\mathrm{cov}(x^1, x^1) + \mathrm{cov}(x^1, x^{1'})}{\mathrm{std}(x^1)\mathrm{std}(x^2)} = \frac{\mathrm{var}(x^1)}{\sqrt{2}\mathrm{std}(x^1)\mathrm{std}(x^1)} = \frac{1}{\sqrt{2}} \ . \tag{102}$$

An exactly identical calculation applies for $\mathrm{corr}(e^1, e^2)$ which is also equal to $1/\sqrt{2}$.

Thus, we see that this case violates the simple identifiability condition in Assumption 1. This is understandable since no new information is created when computing the variable in the second view, and the two views are thus degenerate.

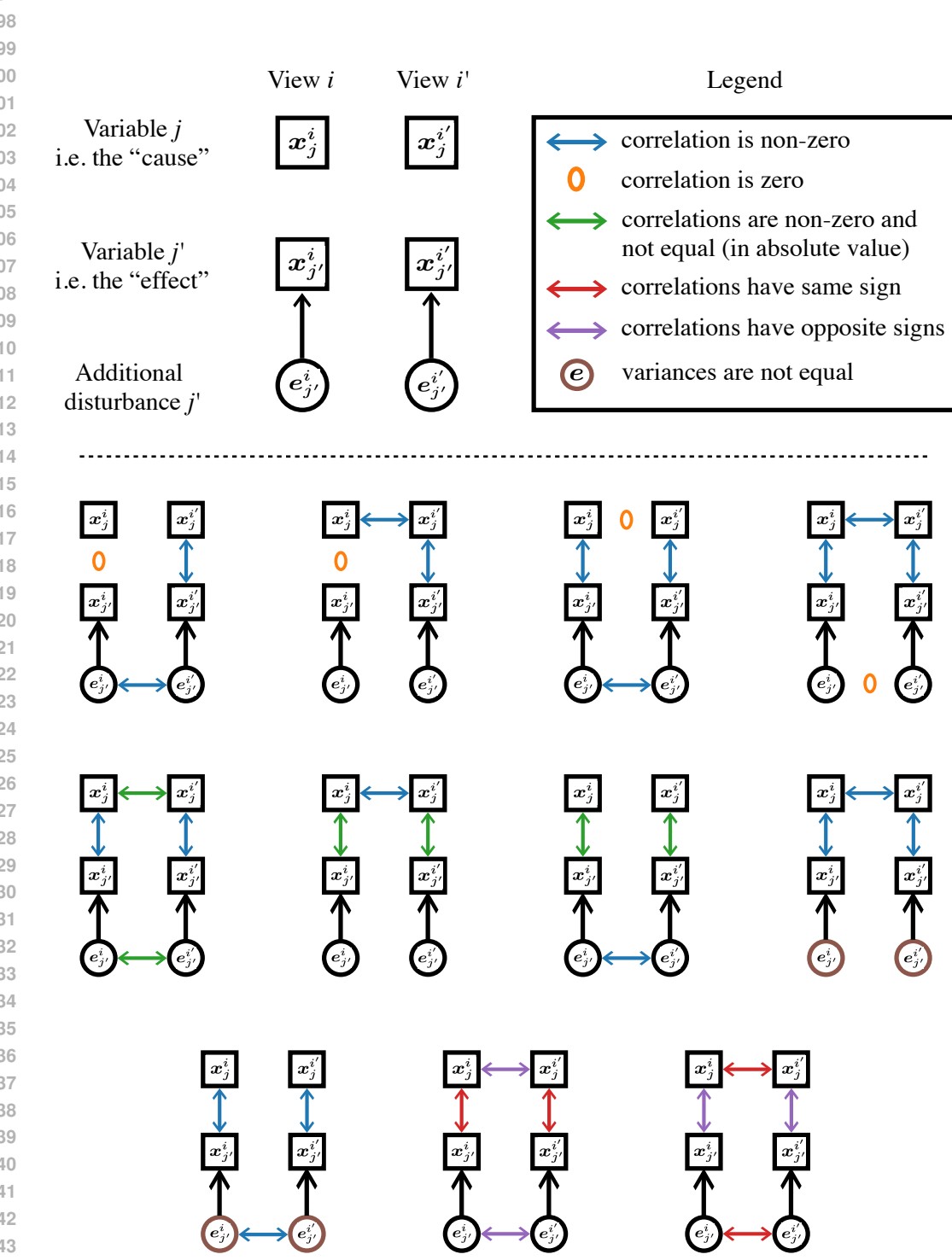

Figure 4: Assume the direction is $x_j \rightarrow x_{j'}$. Assumption 6 corresponds to being in one of the 11 situations showed in the figure. Note that no arrow means that no information is required.

## D.3 PROOF OF THEOREM 1

We proceed in three steps. (i) We show that the LiMVAM model always admits at least one root variable and that our LR and FC criteria recover one consistently. (ii) We prove a *recursive residuals* lemma: removing a recovered root and regressing on it preserves the model form and the causal ordering, which implies that the full (partial) ordering is consistently recoverable. (iii) Conditional on the recovered ordering, each view-specific strictly lower-triangular system can be estimated consistently, hence the adjacency matrices are identifiable.

**Preliminaries**   For clarity, we unpack the "shared ordering" assumption. This assumption is equivalent to the existence of a *main* DAG $\mathcal{G}$ that is the union of view-specific DAGs $\{\mathcal{G}^i\}_{i=1}^m$: an edge appears in $\mathcal{G}$ if it appears in at least one $\mathcal{G}^i$. Root variables are therefore common candidates across views. Furthermore, in terms of vocabulary, we distinguish (a) *directed paths* (a sequence of edges with consistent direction) from (b) *direct edges* (one edge between adjacent variables). An *indirect path* is a directed path with at least two edges.

### D.3.1 RECOVERING A ROOT VARIABLE

Let observations follow LiMVAM in Eq. 3. Since all views share an ordering, there exists a (possibly non-unique) permutation $\boldsymbol{P}$ such that

$$\boldsymbol{T}^i = \boldsymbol{P}^\top \boldsymbol{B}^i \boldsymbol{P} \quad \text{is strictly lower triangular for all } i \in [\![1, m]\!] \ . \tag{103}$$

Reordering the observations and disturbances with $\boldsymbol{x}'^i = \boldsymbol{P}\boldsymbol{x}^i$ and $\boldsymbol{e}'^i = \boldsymbol{P}\boldsymbol{e}^i$, the model is

$$\boldsymbol{x}'^i = \boldsymbol{T}^i \boldsymbol{x}'^i + \boldsymbol{e}'^i, \qquad i \in [\![1, m]\!] \ . \tag{104}$$

The first row of each $\boldsymbol{T}^i$ is zero, so $x_1'^i$ is exogenous; hence a root variable always exists.

To *find* a root, consider distinct indices $(j, k)$ and define $M_{jk}$ to be the criterion from Eq. 5 or Eq. 6 (with $M_{kj} = -M_{jk}$). Stack these into $\boldsymbol{M} \in \mathbb{R}^{p \times p}$ with zero diagonal, and analyze signs under Assumption 6 (which concerns only direct relations):

1. **Direct edge:** $\boldsymbol{x}_j \to \boldsymbol{x}_k$ or $\boldsymbol{x}_k \to \boldsymbol{x}_j$. Under the bivariate conditions derived from Assumption 6, $M_{jk} > 0 > M_{kj}$ when $\boldsymbol{x}_j \to \boldsymbol{x}_k$, and $M_{jk} < 0 < M_{kj}$ when $\boldsymbol{x}_k \to \boldsymbol{x}_j$.

2. **Indirect path:** $\boldsymbol{x}_j \to \cdots \to \boldsymbol{x}_k$ or $\boldsymbol{x}_k \to \cdots \to \boldsymbol{x}_j$. Consider the reduced form of the model in Eq. 4, $\boldsymbol{x}^i = \boldsymbol{A}^i \boldsymbol{e}^i$ with $\boldsymbol{A}^i = (\boldsymbol{I} - \boldsymbol{B}^i)^{-1}$. The entry $A_{kj}^i$ is the *total causal effect* of $x_j^i$ on $x_k^i$, *i.e.* the sum over all directed paths from $j$ to $k$ of the products of edge coefficients. Hence for each view $i$,

$$x_k^i = A_{kj}^i x_j^i + \epsilon^i \tag{105}$$

where $\epsilon^i$ collects all terms not caused by $x_j^i$ and is independent of $x_j^i$. Thus, the pair $(x_j^i, x_k^i)$ behaves as if there were a direct edge with coefficient $A_{kj}^i$, and the sign analysis of Item 1 applies but with non-strict inequalities: $M_{jk} \geq 0 \geq M_{kj}$ when $\boldsymbol{x}_j \to \cdots \to \boldsymbol{x}_k$ (and conversely). If the bivariate conditions hold for $(j, k)$, the inequalities are strict; however, we do not require this here.

3. **No relation, no common ancestor:** then $\boldsymbol{x}_j$ and $\boldsymbol{x}_k$ depend on disjoint disturbance sets and are independent, hence $M_{jk} = M_{kj} = 0$.

4. **No relation but with a common ancestor.** If $\boldsymbol{x}_j$ and $\boldsymbol{x}_k$ are not connected by any directed path but share at least one ancestor, then their pairwise criteria $M_{jk}$ and $M_{kj}$ may take mixed signs. However, since each of $\boldsymbol{x}_j$ and $\boldsymbol{x}_k$ has at least one parent, there exist variables $\boldsymbol{x}_{j'}$ and $\boldsymbol{x}_{k'}$ such that $M_{jj'} < 0$ and $M_{kk'} < 0$. Thus, even if the signs of $M_{jk}$ and $M_{kj}$ are not determined, the $j$-th and $k$-th rows of $\boldsymbol{M}$ necessarily contain negative entries. For instance, with three variables, $\boldsymbol{x}_1 \to \boldsymbol{x}_2$ and $\boldsymbol{x}_1 \to \boldsymbol{x}_3$ makes $\boldsymbol{x}_2$ and $\boldsymbol{x}_3$ siblings: there is no directed path between them, but both rows 2 and 3 of $\boldsymbol{M}$ contain negative entries due to their direct parent $\boldsymbol{x}_1$.

Consequently, $\boldsymbol{x}_j$ is a root (no incoming edges) *if and only if* the $j$-th row of $\boldsymbol{M}$ has only nonnegative entries. Hence a root exists and is consistently detectable. If multiple roots exist, we pick one at random.

### D.3.2 Recovering the causal ordering

We now show the recursive step.

**Lemma 7** (Residuals preserve LiMVAM and the ordering) *Let $j$ be a root variable. Regress every other variable on $\boldsymbol{x}_j$ (within each view) and remove $\boldsymbol{x}_j$. The residual vector still follows a LiMVAM model with the* same *ordering over the remaining variables.*

*Proof.* The proof is a direct adaptation of Shimizu et al. (2011, Lemma 2 and Corollary 1) to the LiMVAM model: the original result is in the single-view setting with non-Gaussian disturbances.

Choose a permutation $\boldsymbol{P}$ that places $\boldsymbol{x}_j$ first for simplicity (since $\boldsymbol{x}_j$ is a root variable, this permutation exists). In the permuted system we write

$$\boldsymbol{x}'^i = \boldsymbol{T}^i \boldsymbol{x}'^i + \boldsymbol{e}'^i, \qquad \boldsymbol{x}'^i = \boldsymbol{P}\boldsymbol{x}^i, \ \ \boldsymbol{e}'^i = \boldsymbol{P}\boldsymbol{e}^i \tag{106}$$

where each $\boldsymbol{T}^i$ is strictly lower triangular and $\boldsymbol{x}'_1 = \boldsymbol{x}_j$. Equivalently, the model can be expressed in reduced form

$$\boldsymbol{x}'^i = \boldsymbol{A}^i \boldsymbol{e}'^i, \qquad \boldsymbol{A}^i := (\boldsymbol{I} - \boldsymbol{T}^i)^{-1} \tag{107}$$

where the matrix $\boldsymbol{A}^i$ is lower triangular with unit diagonal.

The entry $A^i_{k1}$ ($k \geq 2$) quantifies the dependence of $x'^i_k$ on $e'^i_1$. Since $x'_1 = e'^i_1$ is exogenous, we have

$$\mathbb{E}\left[x'^i_k x'^i_1\right] = A^i_{k1}\, \mathbb{E}\left[(x'^i_1)^2\right] \tag{108}$$

so the least-squares regression coefficient of $x'^i_k$ on $x'^i_1$ equals $A^i_{k1}$. Therefore, regressing each $x'^i_k$ ($k \geq 2$) on $x'^i_1$ yields residuals $\boldsymbol{r}^i_{(1)} \in \mathbb{R}^{p-1}$ such that

$$\boldsymbol{r}^i_{(1)} = \boldsymbol{x}'^i_{2:p} - \boldsymbol{A}^i_{2:p,1}\, x'^i_1 = \boldsymbol{A}^i_{2:p,:}\, \boldsymbol{e}'^i - \boldsymbol{A}^i_{2:p,1} e'^i_1 = \boldsymbol{A}^i_{2:p,2:p}\, \boldsymbol{e}'^i_{2:p} \ . \tag{109}$$

Here $\boldsymbol{A}^i_{2:p,2:p}$ is again lower triangular with unit diagonal, and $\boldsymbol{e}'^i_{2:p}$ has mutually independent components. Hence $\boldsymbol{r}^i_{(1)}$ follows a LiMVAM model on $p - 1$ variables. Moreover, since $\boldsymbol{A}^i_{2:p,2:p}$ is precisely the submatrix of $\boldsymbol{A}^i$ obtained by deleting the first row and column, the relative structure among the remaining variables is unchanged. In other words, the causal ordering of variables $2, \ldots, p$ is preserved. $\qquad\square$

Applying Lemma 7 recursively — find a root via $\boldsymbol{M}$ (under Assumption 6), regress out its effect, remove it, and repeat — recovers the full (partial) causal ordering consistently.

### D.3.3 Recovering the adjacency matrices

Sections D.3.1 and D.3.2 showed that, under Assumption 6, the algorithms PairwiseLiMVAM and DirectLiMVAM consistently recover the causal ordering. Equivalently, they recover a permutation matrix $\boldsymbol{P}$ that makes the $\boldsymbol{T}^i = \boldsymbol{P}^\top \boldsymbol{B}^i \boldsymbol{P}$ strictly lower triangular. So, we can reorder the variables as follows:

$$\boldsymbol{x}'^i = \boldsymbol{T}^i \boldsymbol{x}'^i + \boldsymbol{e}'^i \ , \qquad i \in [\![1, m]\!] \ . \tag{110}$$

Fix $j \in [\![1, p]\!]$. We have

$$x'^i_j = \sum_{k=1}^{j-1} T^i_{jk} x'^i_k + e'^i_j \ , \qquad i \in [\![1, m]\!] \ . \tag{111}$$

Because $e'^i_j$ is independent of $(x'^i_1, \ldots, x'^i_{j-1})$, the regressors are exogenous. This is a system of linear regressions across views with potentially cross-view–correlated errors. Feasible GLS is consistent in this setting: once a consistent estimate of the cross-view error covariance is obtained, the GLS estimator converges to the true coefficients (Greene, 2003). Therefore, $\{T^i_{jk}\}_{k<j}$ are consistently estimated for every $(i, j)$, hence each $\boldsymbol{T}^i$ is consistently recovered. Finally, $\boldsymbol{B}^i = \boldsymbol{P}\boldsymbol{T}^i\boldsymbol{P}^\top$, so the view-specific adjacency matrices are identifiable. This concludes the proof.

# E SOS-BASED ALGORITHMS FOR LIMVAM

## E.1 CONSISTENT ESTIMATOR OF THE LIKELIHOOD-BASED CRITERION

Assume we have observed $n$ independent samples, concatenated as $\boldsymbol{X} = [\boldsymbol{x}_1, \ldots, \boldsymbol{x}_n]$ and $\boldsymbol{Y} = [\boldsymbol{y}_1, \ldots, \boldsymbol{y}_n]$, with sample index $j$.

In practice, the four covariance matrices in Eq. 43 can be estimated by consistent estimators from the samples in $\boldsymbol{X}$ and $\boldsymbol{Y}$. Since $b_i = c_i = \mathrm{cov}(x_i, y_i)$ for all $i$, we define

$$\forall i: \qquad \hat{b}_i := \hat{c}_i := \frac{1}{n} \sum_{j=1}^{n} x_{ij}\, y_{ij} \xrightarrow{p} \mathrm{cov}(x_i, y_i) = b_i = c_i \tag{112}$$

so that $\hat{\boldsymbol{b}} := (\hat{b}_1, \ldots, \hat{b}_m)^\top \xrightarrow{p} \boldsymbol{b}$ and $\hat{\boldsymbol{c}} := (\hat{c}_1, \ldots, \hat{c}_m)^\top \xrightarrow{p} \boldsymbol{c}$. Using these vectors, we define the empirical covariances of the residuals

$$\widehat{\boldsymbol{\Sigma}}_e := \frac{1}{n} \sum_{j=1}^{n} (\boldsymbol{y}_j - \hat{\boldsymbol{b}} \odot \boldsymbol{x}_j)(\boldsymbol{y}_j - \hat{\boldsymbol{b}} \odot \boldsymbol{x}_j)^\top \xrightarrow{p} \boldsymbol{\Sigma}_e \tag{113}$$

$$\widehat{\boldsymbol{\Sigma}}_d := \frac{1}{n} \sum_{j=1}^{n} (\boldsymbol{x}_j - \hat{\boldsymbol{b}} \odot \boldsymbol{y}_j)(\boldsymbol{x}_j - \hat{\boldsymbol{b}} \odot \boldsymbol{y}_j)^\top \xrightarrow{p} \boldsymbol{\Sigma}_d \tag{114}$$

and we also define the empirical covariances of the observations

$$\widehat{\boldsymbol{\Sigma}}_x := \frac{1}{n} \sum_{j=1}^{n} \boldsymbol{x}_j \boldsymbol{x}_j^\top \xrightarrow{p} \boldsymbol{\Sigma}_x \tag{115}$$

$$\widehat{\boldsymbol{\Sigma}}_y := \frac{1}{n} \sum_{j=1}^{n} \boldsymbol{y}_j \boldsymbol{y}_j^\top \xrightarrow{p} \boldsymbol{\Sigma}_y \ . \tag{116}$$

We can then evaluate the criterion in these estimators.

$$\widehat{\mathrm{LR}} := \mathcal{L}(\widehat{\boldsymbol{\Sigma}}_x, \widehat{\boldsymbol{\Sigma}}_e; x \to y) - \mathcal{L}(\widehat{\boldsymbol{\Sigma}}_y, \widehat{\boldsymbol{\Sigma}}_d; y \to x) = -\log|\widehat{\boldsymbol{\Sigma}}_x| - \log|\widehat{\boldsymbol{\Sigma}}_e| + \log|\widehat{\boldsymbol{\Sigma}}_y| + \log|\widehat{\boldsymbol{\Sigma}}_d| \tag{117}$$

and it follows that this empirical criterion is a consistent estimator of the true criterion:

$$\mathcal{L}(\widehat{\boldsymbol{\Sigma}}_x, \widehat{\boldsymbol{\Sigma}}_e; x \to y) - \mathcal{L}(\widehat{\boldsymbol{\Sigma}}_y, \widehat{\boldsymbol{\Sigma}}_d; y \to x) \xrightarrow{p} \mathcal{L}(\boldsymbol{\Sigma}_x, \boldsymbol{\Sigma}_e; x \to y) - \mathcal{L}(\boldsymbol{\Sigma}_y, \boldsymbol{\Sigma}_d; y \to x) =: \mathrm{LR} \ . \tag{118}$$

## E.2 CONSISTENT ESTIMATOR OF THE CROSS-COVARIANCE-BASED CRITERION

Assume we have observed $n$ independent samples, concatenated as $\boldsymbol{X} = [\boldsymbol{x}_1, \ldots, \boldsymbol{x}_n]$ and $\boldsymbol{Y} = [\boldsymbol{y}_1, \ldots, \boldsymbol{y}_n]$, with sample index $j$.

In practice, the matrices $\mathbb{E}[\boldsymbol{ex}^\top]$ and $\mathbb{E}[\boldsymbol{dy}^\top]$ are estimated from the data. Performing univarite least-squares regression of $\boldsymbol{y}$ on $\boldsymbol{x}$ (resp. $\boldsymbol{x}$ on $\boldsymbol{y}$) yields the residual matrices $\widehat{\boldsymbol{E}} = [\hat{\boldsymbol{e}}_1, \ldots, \hat{\boldsymbol{e}}_n]$ (resp. $\widehat{\boldsymbol{D}} = [\hat{\boldsymbol{d}}_1, \ldots, \hat{\boldsymbol{d}}_n]$). The corresponding cross-covariance estimators are

$$\frac{1}{n} \sum_{j=1}^{n} \hat{\boldsymbol{e}}_j \boldsymbol{x}_j^\top \xrightarrow{p} \mathbb{E}[\boldsymbol{ex}^\top] \qquad \text{and} \qquad \frac{1}{n} \sum_{j=1}^{n} \hat{\boldsymbol{d}}_j \boldsymbol{y}_j^\top \xrightarrow{p} \mathbb{E}[\boldsymbol{dy}^\top] \ . \tag{119}$$

Consequently, the empirical criterion

$$\widehat{\mathrm{FC}} := \left\| \frac{1}{n} \sum_{j=1}^{n} \hat{\boldsymbol{d}}_j \boldsymbol{y}_j^\top \right\|_F - \left\| \frac{1}{n} \sum_{j=1}^{n} \hat{\boldsymbol{e}}_j \boldsymbol{x}_j^\top \right\|_F \tag{120}$$

is a consistent estimator of the population criterion:

$$\widehat{\mathrm{FC}} \xrightarrow{p} \left\| \mathbb{E}[\boldsymbol{dy}^\top] \right\|_F - \left\| \mathbb{E}[\boldsymbol{ex}^\top] \right\|_F =: \mathrm{FC} \ . \tag{121}$$

E.3  Algorithms for estimating the two criteria

In this section, the observations $\boldsymbol{X}, \boldsymbol{Y} \in \mathbb{R}^{m \times n}$ lie in two dimensions: the $m$ views, and the $n$ independent samples. We use the index $i$ for the views, and $k$ for the samples.

---

**Algorithm 1** Likelihood-based criterion

**Input:** Observations $\boldsymbol{X}, \boldsymbol{Y} \in \mathbb{R}^{m \times n}$ corresponding to two variables, residuals $\boldsymbol{E}, \boldsymbol{D} \in \mathbb{R}^{m \times n}$ corresponding to the directions $x \to y$ and $y \to x$, respectively.

1. Compute the covariance matrices for the direction $x \to y$:

$$\widehat{\boldsymbol{\Sigma}}_x = \frac{1}{n} \sum_{j=1}^n \boldsymbol{X}_j \boldsymbol{X}_j^\top \qquad \text{and} \qquad \widehat{\boldsymbol{\Sigma}}_e = \frac{1}{n} \sum_{j=1}^n \boldsymbol{E}_j \boldsymbol{E}_j^\top \tag{122}$$

and for the direction $y \to x$:

$$\widehat{\boldsymbol{\Sigma}}_y = \frac{1}{n} \sum_{j=1}^n \boldsymbol{Y}_j \boldsymbol{Y}_j^\top \qquad \text{and} \qquad \widehat{\boldsymbol{\Sigma}}_d = \frac{1}{n} \sum_{j=1}^n \boldsymbol{D}_j \boldsymbol{D}_j^\top . \tag{123}$$

2. Compute the likelihood based-criterion:

$$\widehat{\text{LR}} = -\log |\widehat{\boldsymbol{\Sigma}}_x| - \log |\widehat{\boldsymbol{\Sigma}}_e| + \log |\widehat{\boldsymbol{\Sigma}}_y| + \log |\widehat{\boldsymbol{\Sigma}}_d| . \tag{124}$$

**Output:** Likelihood-based criterion $\widehat{\text{LR}}$.

---

**Algorithm 2** Cross-covariance-based criterion

**Input:** Observations $\boldsymbol{X}, \boldsymbol{Y} \in \mathbb{R}^{m \times n}$ corresponding to two variables, residuals $\boldsymbol{E}, \boldsymbol{D} \in \mathbb{R}^{m \times n}$ corresponding to the directions $x \to y$ and $y \to x$, respectively.

1. Compute the second-moment matrices

$$\boldsymbol{M}_1 = \frac{1}{n} \sum_{j=1}^n \boldsymbol{E}_j \boldsymbol{X}_j^\top \qquad \text{and} \qquad \boldsymbol{M}_2 = \frac{1}{n} \sum_{j=1}^n \boldsymbol{D}_j \boldsymbol{Y}_j^\top . \tag{125}$$

2. Compute the cross-covariance-based criterion:

$$\widehat{\text{FC}} = \|\boldsymbol{M}_2\|_F - \|\boldsymbol{M}_1\|_F . \tag{126}$$

**Output:** Cross-covariance-based criterion $\widehat{\text{FC}}$.

---

E.4  How to find the root variable

Assume we have a scalar criterion of the causal direction between two variables $\boldsymbol{x}_j = (x_j^1, \ldots, x_j^m)^\top$ and $\boldsymbol{x}_k = (x_k^1, \ldots, x_k^m)^\top$, such that the criterion is positive if $\boldsymbol{x}_j \to \boldsymbol{x}_k$ and negative in the opposite direction. Let $M_{jk}$ denote the criterion obtained for the pair $(j, k)$, and collect all pairwise criteria in a matrix $\boldsymbol{M}$ with a zero diagonal. We now have a matrix $\boldsymbol{M}$ whose entries determine the causal direction by their sign: $M_{ij} > 0$ indicates $\boldsymbol{x}_i$ causes $\boldsymbol{x}_j$, and $M_{ij} < 0$ indicates that $\boldsymbol{x}_j$ causes $\boldsymbol{x}_i$. In this setting, $\boldsymbol{x}_j$ is a root variable if and only if the $j$-th row of $\boldsymbol{M}$ contains only non-negative entries. We follow Hyvärinen & Smith (2013, Section 3.2.2) to derive a principled criterion for finding the root variable given $\boldsymbol{M}$. We next recall their argument.

Suppose that the entries in $\boldsymbol{M}$ follow a normal distribution, so that $M_{ij} \sim \mathcal{N}(\mu_{ij}, \sigma^2)$ where $\sigma^2$ models the estimation error due to finite samples (and is supposed to be constant for simplicity).

Then, the log-likelihood of $\boldsymbol{x}_i$ being the root variable is

$$\log \prod_{j=1}^{p} P(\mu_{ij} > 0|M_{ij}) = \sum_{j=1}^{p} \log P\Big(\frac{\mu_{ij} - M_{ij}}{\sigma} > \frac{-M_{ij}}{\sigma}\Big|M_{ij}\Big) \tag{127}$$

$$= \sum_{j=1}^{p} \log \Phi\Big(\frac{M_{ij}}{\sigma}\Big) \approx \frac{-1}{2\sigma^2} \sum_{j=1}^{p} \min(0, M_{ij})^2 \tag{128}$$

where $\Phi$ is the cumulative distribution function of a standardized Gaussian, whose log we then approximated. We therefore obtain the index of the root variable using

$$\arg\min_i \sum_{j=1}^{p} \min(0, M_{ij})^2 \ . \tag{129}$$

### E.5 ONE-STEP FEASIBLE GENERALIZED LEAST SQUARES (FGLS)

The FGLS procedure begins with an Ordinary Least Squares (OLS) regression in each view to obtain residuals $(e_j'^1, \ldots, e_j'^m)$, which are then used to estimate the cross-view covariance of the disturbances. In a second step, the OLS coefficients are adjusted via Generalized Least Squares using this estimated covariance. In the population limit, this estimator coincides with OLS but is asymptotically more efficient whenever disturbances are correlated across views, as is the case here.

The estimation proceeds as follows. Assume we have observed $n$ independent samples. Here, we consider the model

$$\boldsymbol{x}'^i = \boldsymbol{T}^i \boldsymbol{x}'^i + \boldsymbol{e}'^i \ , \qquad i \in [\![1, m]\!] \tag{130}$$

where $\boldsymbol{x}'^i$ and $\boldsymbol{e}'^i$ are the reordered observations and disturbances, respectively, and $\boldsymbol{T}^i$ are strictly lower triangular matrices. By construction, each variable $x_j'^i$ only depends on its predecessors $\boldsymbol{x}_{<j}'^i := (x_1'^i, \ldots, x_{j-1}'^i)$.

Fix a variable index $j \in [\![1, p]\!]$. In view $i$, the structural equation reads

$$x_j'^i = \sum_{k=1}^{j-1} T_{jk}^i\, x_k'^i + e_j'^i \ . \tag{131}$$

Stacking all views, we obtain the block regression model

$$\boldsymbol{x}_j' = \boldsymbol{X}_{<j}\, \boldsymbol{T}_{<j} + \boldsymbol{e}_j' \tag{132}$$

where

- $\boldsymbol{x}_j' = \big(x_j'^1, \ldots, x_j'^m\big)^\top \in \mathbb{R}^{mn}$ stacks all samples across the $m$ views,
- $\boldsymbol{X}_{<j} = \mathrm{diag}\big(\boldsymbol{x}_{<j}'^1, \ldots, \boldsymbol{x}_{<j}'^m\big) \in \mathbb{R}^{mn \times m(j-1)}$ is block-diagonal and contains the predecessors,
- $\boldsymbol{T}_{<j} = \big(\boldsymbol{T}_{j,<j}^1, \ldots, \boldsymbol{T}_{j,<j}^m\big)^\top \in \mathbb{R}^{m(j-1)}$ collects the coefficients,
- $\boldsymbol{e}_j' = \big(e_j'^1, \ldots, e_j'^m\big)^\top \in \mathbb{R}^{mn}$ are disturbances.

The disturbance vector $\boldsymbol{e}_j'$ has covariance

$$\Omega_j := \mathbb{E}\big[\boldsymbol{e}_j'(\boldsymbol{e}_j')^\top\big] = \boldsymbol{\Sigma}_{\boldsymbol{e}_j} \otimes \boldsymbol{I}_n \in \mathbb{R}^{mn \times mn} \tag{133}$$

where $\boldsymbol{\Sigma}_{\boldsymbol{e}_j} \in \mathbb{R}^{m \times m}$ captures cross-view covariances for component $j$.

The one-step FGLS procedure is:

1. *OLS step:*

$$\widehat{\boldsymbol{T}}_{<j}^{\mathrm{OLS}} = \big(\boldsymbol{X}_{<j}^\top \boldsymbol{X}_{<j}\big)^{-1} \boldsymbol{X}_{<j}^\top \boldsymbol{x}_j' \tag{134}$$

yielding residuals $\hat{\boldsymbol{e}}_j' = \boldsymbol{x}_j' - \boldsymbol{X}_{<j} \widehat{\boldsymbol{T}}_{<j}^{\mathrm{OLS}}$.

2. *Covariance estimation:*

$$\widehat{\boldsymbol{\Sigma}}_{\boldsymbol{e}_j} = \frac{1}{n} \sum_{k=1}^{n} \hat{e}'_{j,k} (\hat{e}'_{j,k})^\top \tag{135}$$

where $k$ denotes the sample index.

3. *FGLS update:*

$$\widehat{\boldsymbol{T}}^{\text{FGLS}}_{<j} = \left( \boldsymbol{X}_{<j}^\top \widehat{\boldsymbol{\Sigma}}_{\boldsymbol{e}_j}^{-1} \boldsymbol{X}_{<j} \right)^{-1} \boldsymbol{X}_{<j}^\top \widehat{\boldsymbol{\Sigma}}_{\boldsymbol{e}_j}^{-1} \boldsymbol{x}'_j \ . \tag{136}$$

This estimator reduces to OLS if $\boldsymbol{\Sigma}_{\boldsymbol{e}_j}$ is diagonal (no cross-view correlation). More generally, it achieves asymptotic efficiency by exploiting cross-view correlations.

### E.6 FULL ALGORITHM

---

**Algorithm 3** Pairwise-comparison algorithms for multi-view causal discovery

---

**Input:** Observations $\boldsymbol{X} \in \mathbb{R}^{m \times p \times n}$.

1. *Preprocess the data $\boldsymbol{X}$.* Subtract the mean over the samples axis.

2. *Estimate the causal ordering $\boldsymbol{P}$.*

   (a) Perform pairwise regressions between each pair of variables, $\boldsymbol{x}_j$ on $\boldsymbol{x}_i$, such that $i \neq j$. Compute the residuals $\boldsymbol{e}_{i \to j}$.

   (b) Compute a skew-symmetric matrix $\boldsymbol{M} \in \mathbb{R}^{p \times p}$ that determines the causal direction between pairs of variables.

   The entries $M_{ij}$ are computed with Eq. 5 or Eq. 6 for all $i < j$, using the regression results $(\boldsymbol{x}_i, \boldsymbol{x}_j, \boldsymbol{e}_{i \to j}, \boldsymbol{e}_{j \to i})$. Then set $M_{ji} = -M_{ij}$. $M_{ij}$ positive means $\boldsymbol{x}_i$ causes $\boldsymbol{x}_j$, negative means $\boldsymbol{x}_j$ causes $\boldsymbol{x}_i$.

   (c) Determine the root variable $k$: it causes all others, so $M_{kj}$ should be positive for all $j$.

   $$k = \operatorname*{argmin}_i \sum_j \min(0, M_{ij})^2 \ . \tag{137}$$

   (d) Remove the root variable $k$ and replace observations with residuals obtained when regressing on $k$. Repeat.

   (e) Store the estimated ordering in a permutation matrix $\boldsymbol{P}$, and reorder the variables in $\boldsymbol{X}$ according to this permutation. This yields a new model $\boldsymbol{x}'^i = \boldsymbol{T}^i \boldsymbol{x}'^i + \boldsymbol{e}'^i$, where $\boldsymbol{x}'^i$ and $\boldsymbol{e}'^i$ are reordered versions of $\boldsymbol{x}^i$ and $\boldsymbol{e}^i$, respectively, and the $\boldsymbol{T}^i$ are strictly lower triangular matrices.

3. *Estimate the adjacency matrices $\boldsymbol{B}^i$.* For each variable $j \in [\![2, p]\!]$, estimate the $j$-th row of all matrices $\boldsymbol{T}^i$ with one-step Feasible Generalized Least Squares Zellner (1962). Recover matrices $\boldsymbol{B}^i = \boldsymbol{P}^\top \boldsymbol{T}^i \boldsymbol{P}$.

**Output:** Adjacency matrices $(\boldsymbol{B}^1, \ldots, \boldsymbol{B}^m) \in \mathbb{R}^{m \times p \times p}$.

---

### E.7 COMPUTATIONAL COMPLEXITY

The computational complexity of Algorithm 3 can be broken down across the different steps.

In Step 1, the preprocessing is in $O(m \cdot p \cdot n)$.

In Step 2.a., we solve $O(p^2)$ LS regression problems, each of which has complexity $O(m \cdot n)$.

In Step 2.b., we compute $O(p^2)$ entries of a matrix. Each entry is computed either using the cross-covariance-based criterion in Algorithm 2 which is in $O(m^2 \cdot n)$, or using the likelihood-based criterion in Algorithm 1 which is in $O(m^2 \cdot n + m^3)$ due to computing log determinants. So the total complexity of this step is $O(m^2 \cdot p^2 \cdot n)$ or $O(m^2 \cdot p^2 \cdot n + m^3 \cdot p^2)$.

In Step 2.c., we make $O(p^2)$ operations.

In Step 2.d., we simplify redefine a matrix.

These steps 2.a-d., are repeated $O(p)$ times, until all variables are removed, so this yields a complexity in either $O(m^2 \cdot p^3 \cdot n)$ or $O(m^2 \cdot p^3 \cdot n + m^3 \cdot p^3)$.

In Step 2.e., we reorder a matrix, which costs $O(m \cdot p \cdot n)$ operations.

In Step 3, we run the one-step Feasible GLS procedure, which is in $O(m^3 \cdot p^3 \cdot n)$. It scales cubically in the $m$ views and $p$ dimensions as it involves matrix multiplications and inversions. It then does $m$ matrix multiplications. Each of them involves permutation matrices which has a quadratic cost $O(p^2)$. So the cost of this step is dominated by $O(m^3 \cdot p^3 \cdot n)$.

The total computational complexity of the algorithm is thus $O(m^3 \cdot p^3 \cdot n)$.

# F   IDENTIFIABILITY OF LiMVAM WITH SHARED DISTURBANCES

In this section, we analyze the identifiability of the LiMVAM model with shared disturbances

$$\boldsymbol{x}^i = \boldsymbol{B}^i \boldsymbol{x}^i + \boldsymbol{D}^i \boldsymbol{s} + \boldsymbol{n}^i \,, \qquad i \in [\![1, m]\!] \tag{138}$$

where the $\boldsymbol{D}^i$ are diagonal matrices with positive entries on the diagonal, $\boldsymbol{s}$ has mutually independent entries and second-order moment $\mathbb{E}[\boldsymbol{s}\boldsymbol{s}^\top] = \boldsymbol{I}_p$, the view-specific noises are Gaussian $\boldsymbol{n}^i \sim \mathcal{N}(\boldsymbol{0}, \boldsymbol{\Sigma}^i)$ with diagonal $\boldsymbol{\Sigma}^i$, and the vectors $\boldsymbol{s}$ and $\boldsymbol{n}^i, i = 1, \ldots, m$, are mutually independent.

## F.1   PROOF OF THEOREM 2 (FIRST CLAIM)

Consider two sets $\Theta = (\boldsymbol{D}^1, \ldots, \boldsymbol{D}^m, \boldsymbol{\Sigma}^1, \ldots, \boldsymbol{\Sigma}^m, \boldsymbol{B}^1, \ldots, \boldsymbol{B}^m)$ and $\Theta' = (\boldsymbol{D}'^1, \ldots, \boldsymbol{\Sigma}'^m, \boldsymbol{B}'^1, \ldots, \boldsymbol{B}'^m)$ that parameterize the same statistical model in Eq. 138, with $\boldsymbol{D}^i$ and $\boldsymbol{D}'^i$ being diagonal matrices with positive entries on the diagonal, $\boldsymbol{\Sigma}^i$ and $\boldsymbol{\Sigma}'^i$ being diagonal matrices with non-zero entries on the diagonal, and $\boldsymbol{B}^i$ and $\boldsymbol{B}'^i$ being DAG matrices. The model in Eq. 138 can be reformulated as

$$\boldsymbol{x}^i = (\boldsymbol{I} - \boldsymbol{B}^i)^{-1} \boldsymbol{D}^i (\boldsymbol{s} + (\boldsymbol{D}^i)^{-1} \boldsymbol{n}^i) \tag{139}$$

which corresponds to the Shared ICA model

$$\boldsymbol{x}^i = \boldsymbol{A}^i (\boldsymbol{s} + \tilde{\boldsymbol{n}}^i) \tag{140}$$

for $\boldsymbol{A}^i = (\boldsymbol{I} - \boldsymbol{B}^i)^{-1} \boldsymbol{D}^i$ and $\tilde{\boldsymbol{n}}^i = (\boldsymbol{D}^i)^{-1} \boldsymbol{n}^i$. We can observe that the unmixing matrices $\boldsymbol{W}^i = (\boldsymbol{A}^i)^{-1} = (\boldsymbol{D}^i)^{-1}(\boldsymbol{I} - \boldsymbol{B}^i)$ and $\boldsymbol{W}'^i = (\boldsymbol{A}'^i)^{-1} = (\boldsymbol{D}'^i)^{-1}(\boldsymbol{I} - \boldsymbol{B}'^i)$ belong to the domain $\mathcal{W}$, which is included in the space of invertible matrices. Thus, the two sets of $\boldsymbol{W}^i$ and $\boldsymbol{W}'^i$ matrices are valid sets of unmixing matrices for the same multi-view ICA model in Eq. 140. Moreover, Assumption 4 allows the rescaled noises $\tilde{\boldsymbol{n}}^i$ to meet the noise diversity condition of Shared ICA. So, from the identifiability theory of multi-view ICA (Richard et al., 2021, Theorem 1), we know that there exist a sign-permutation matrix $\boldsymbol{Q}$ such that for any view $i \in [\![1, m]\!]$, we have

$$\begin{aligned} \boldsymbol{W}'^i &= \boldsymbol{Q}^\top \boldsymbol{W}^i \\ \boldsymbol{\Sigma}'^i &= \boldsymbol{Q}^\top \boldsymbol{\Sigma}^i \boldsymbol{Q} \,. \end{aligned} \tag{141}$$

Note that, contrary to the single-view context, the fact that we obtain $\boldsymbol{W}$ up to *sign* and permutation rather than *scale* and permutation is because $\mathbb{E}[\boldsymbol{s}\boldsymbol{s}^\top] = \boldsymbol{I}$. Then, we apply Lemma 3, which shows that being in the domain $\mathcal{W}$ imposes $\boldsymbol{Q} = \boldsymbol{I}$, $\boldsymbol{D}'^i = \boldsymbol{D}^i$, and $\boldsymbol{B}'^i = \boldsymbol{B}^i$.

## F.2   PROOF OF THEOREM 2 (SECOND CLAIM)

In the context of shared causal ordering $\boldsymbol{P}$, we prove that, under Assumptions 4 and 5, the decomposition of matrices $\boldsymbol{B}^i$ into matrices $\boldsymbol{T}^i$ and $\boldsymbol{P}$ is unique.

Consider the two sets $\Theta = (\boldsymbol{\Sigma}^1, \ldots, \boldsymbol{\Sigma}^m, \boldsymbol{P}, \boldsymbol{T}^1, \ldots, \boldsymbol{T}^m)$ and $\Theta' = (\bar{\boldsymbol{\Sigma}}^1, \ldots, \bar{\boldsymbol{\Sigma}}^m, \bar{\boldsymbol{P}}, \bar{\boldsymbol{T}}^1, \ldots, \bar{\boldsymbol{T}}^m)$ that parameterize the same statistical model in Eq. 138, where $\boldsymbol{P}$ is some permutation matrix, $\boldsymbol{T}^i$ are strictly lower triangular matrices, and $\bar{\boldsymbol{P}}$ and $\bar{\boldsymbol{T}}^i$ are the particular matrices given by Assumption 5. Note that no assumption on the sparsity of the $\boldsymbol{T}^i$ is made. From Theorem 2, we know that $\boldsymbol{\Sigma}^i = \bar{\boldsymbol{\Sigma}}^i$.

Assumption 5 amounts to saying that the "reunion" of the $\bar{\boldsymbol{T}}^i$ represents a fully connected graph. In other words, if we define the reunion as $\bar{\boldsymbol{T}}^U = \sum_{i=1}^m \mathrm{abs}(\bar{\boldsymbol{T}}^i)$, where abs denotes the element-wise absolute value function, then we are assuming that the strictly lower triangular part of $\bar{\boldsymbol{T}}^U$ only has non-zero elements. In a similar way to the definition of $\bar{\boldsymbol{T}}^U$, let us define

$$\bar{\boldsymbol{W}}^U = \sum_{i=1}^m \mathrm{abs}(\bar{\boldsymbol{W}}^i) \tag{142}$$

where $\bar{\boldsymbol{W}}^i = \boldsymbol{I} - \bar{\boldsymbol{P}}^\top \bar{\boldsymbol{T}}^i \bar{\boldsymbol{P}}$. The non-zero elements of $\bar{\boldsymbol{P}}^\top \bar{\boldsymbol{T}}^i \bar{\boldsymbol{P}}$ are outside of the diagonal, so matrices $\boldsymbol{I}$ and $\bar{\boldsymbol{P}}^\top \bar{\boldsymbol{T}}^i \bar{\boldsymbol{P}}$ contain non-zero elements at different locations. Thus, we have

$$\mathrm{abs}(\boldsymbol{I} - \bar{\boldsymbol{P}}^\top \bar{\boldsymbol{T}}^i \bar{\boldsymbol{P}}) = \boldsymbol{I} + \mathrm{abs}(\bar{\boldsymbol{P}}^\top \bar{\boldsymbol{T}}^i \bar{\boldsymbol{P}}) \,. \tag{143}$$

Furthermore, applying $\bar{\boldsymbol{P}}$ to the rows and columns of $\bar{\boldsymbol{T}}^i$ only shuffles its entries, without modifying their values. So we have

$$\mathrm{abs}(\bar{\boldsymbol{P}}^\top \bar{\boldsymbol{T}}^i \bar{\boldsymbol{P}}) = \bar{\boldsymbol{P}}^\top \mathrm{abs}(\bar{\boldsymbol{T}}^i) \bar{\boldsymbol{P}} \ . \tag{144}$$

Consequently,

$$\bar{\boldsymbol{W}}^U = \sum_{i=1}^m \boldsymbol{I} + \bar{\boldsymbol{P}}^\top \mathrm{abs}(\bar{\boldsymbol{T}}^i) \bar{\boldsymbol{P}} = \sum_{i=1}^m \bar{\boldsymbol{P}}^\top (\boldsymbol{I} + \mathrm{abs}(\bar{\boldsymbol{T}}^i)) \bar{\boldsymbol{P}} = \bar{\boldsymbol{P}}^\top (m\boldsymbol{I} + \bar{\boldsymbol{T}}^U) \bar{\boldsymbol{P}} \ . \tag{145}$$

So, dividing both sides by $m$ gives

$$\frac{1}{m}\bar{\boldsymbol{W}}^U = \bar{\boldsymbol{P}}^\top \left(\boldsymbol{I} - \left(-\frac{1}{m}\bar{\boldsymbol{T}}^U\right)\right) \bar{\boldsymbol{P}} \tag{146}$$

where $-\frac{1}{m}\bar{\boldsymbol{T}}^U$ is strictly lower triangular, and its strictly lower triangular part only has non-zero elements.

Next, we apply the same reasoning to the alternative set of parameters, given by $\boldsymbol{W}^i = \boldsymbol{I} - \boldsymbol{P}^\top \boldsymbol{T}^i \boldsymbol{P}$, and we consider $\boldsymbol{W}^U = \sum_{i=1}^m \mathrm{abs}(\boldsymbol{W}^i)$. The proof of Theorem 2 already implied that $\bar{\boldsymbol{W}}^i = \boldsymbol{W}^i$ for all $i$. Thus, we have $\bar{\boldsymbol{W}}^U = \boldsymbol{W}^U$ and

$$\bar{\boldsymbol{P}}^\top \left(\boldsymbol{I} - \left(-\frac{1}{m}\bar{\boldsymbol{T}}^U\right)\right) \bar{\boldsymbol{P}} = \boldsymbol{P}^\top \left(\boldsymbol{I} - \left(-\frac{1}{m}\boldsymbol{T}^U\right)\right) \boldsymbol{P} \tag{147}$$

where $\boldsymbol{T}^U$ is defined in a similar way as $\bar{\boldsymbol{T}}^U$, except that its strictly lower triangular part can be sparse. Using Lemma 4 on Eq. 147, we obtain that $\boldsymbol{P} = \bar{\boldsymbol{P}}$, and thus $\boldsymbol{T}^i = \bar{\boldsymbol{T}}^i$ for all $i$. In conclusion, all the sets of DAG decompositions that parameterize the same model are equal, as soon as for one of these decompositions, the reunion of the $\boldsymbol{T}^i$ is dense. We conclude that matrices $\boldsymbol{P}$ and $\boldsymbol{T}^i$ are unique, and thus identifiable in our terminology. This proof also implies that there can be only one matrix $\bar{\boldsymbol{P}}$ that fulfills Assumption 5.

## F.3  RESULTS FOR VIEW-SPECIFIC CAUSAL ORDERINGS

Next, we consider a case which is outside of the theory of the main paper, although a simple extension: we allow the causal orderings to be different in different views. It turns out that in the view-specific $\boldsymbol{P}^i$ case, identifiability is obtained by assuming that the directed acyclic graph $\boldsymbol{B}^i$ is dense enough in each view, as formalized in the following assumption. However, since the $\boldsymbol{B}^i$ can be permuted to strictly lower triangular matrices, they cannot be denser than having $\frac{p(p-1)}{2}$ non-zero entries.

**Assumption 7** (Dense connectivity in each view) *For each view $i$, the matrix $\boldsymbol{B}^i$ has exactly $\frac{p(p-1)}{2}$ non-zero entries.*

Using Assumption 7, the following theorem states that, in addition to identifying $\boldsymbol{B}^i$, $\boldsymbol{D}^i$, and $\boldsymbol{\Sigma}^i$, one can also identify $\boldsymbol{T}^i$ and $\boldsymbol{P}^i$.

**Theorem 8** (Identifiability of multiple causal orderings) *Consider the LiMVAM model with multiple causal orderings. Consider the quantities $(\boldsymbol{D}^1, \ldots, \boldsymbol{D}^m, \boldsymbol{\Sigma}^1, \ldots, \boldsymbol{\Sigma}^m, \boldsymbol{P}^1, \ldots, \boldsymbol{P}^m, \boldsymbol{T}^1, \ldots, \boldsymbol{T}^m)$ as the set of parameters to be estimated. Under Assumptions 4 and 7, all these parameters are identifiable.*

In the context of view-specific causal orderings $\boldsymbol{P}^i$, we prove that, under Assumptions 4 and 7, the decomposition of matrices $\boldsymbol{B}^i$ into matrices $\boldsymbol{T}^i$ and $\boldsymbol{P}^i$ is unique.

Consider two sets of parameters $\Theta = (\boldsymbol{\Sigma}^1, \ldots, \boldsymbol{\Sigma}^m, \boldsymbol{P}^1, \ldots, \boldsymbol{P}^m, \boldsymbol{T}^1, \ldots, \boldsymbol{T}^m)$ and $\Theta' = (\boldsymbol{\Sigma}'^1, \ldots, \boldsymbol{\Sigma}'^m, \boldsymbol{P}'^1, \ldots, \boldsymbol{P}'^m, \boldsymbol{T}'^1, \ldots, \boldsymbol{T}'^m)$ that parameterize the same statistical model in Eq. 138, where $\boldsymbol{P}^i, \boldsymbol{P}'^i$ are permutation matrices, and $\boldsymbol{T}^i, \boldsymbol{T}'^i$ are strictly lower triangular matrices. Note that here we parameterize by $\boldsymbol{P}^i$ and $\boldsymbol{T}^i$ rather than $\boldsymbol{B}^i$ or $\boldsymbol{W}^i$. From Theorem 2, we know that $\boldsymbol{\Sigma}^i = \boldsymbol{\Sigma}'^i$ and that the resulting causal matrices must be equal:

$$(\boldsymbol{P}^i)^\top \boldsymbol{T}^i \boldsymbol{P}^i = (\boldsymbol{P}'^i)^\top \boldsymbol{T}'^i \boldsymbol{P}'^i \ . \tag{148}$$

Assumption 7 states that, for each view $i$, the matrix $\boldsymbol{B}^i = (\boldsymbol{P}^i)^\top \boldsymbol{T}^i \boldsymbol{P}^i = (\boldsymbol{P}'^i)^\top \boldsymbol{T}'^i \boldsymbol{P}'^i$ contains exactly $\frac{p(p-1)}{2}$ non-zero elements, so it is also the case for $\boldsymbol{T}^i$ and $\boldsymbol{T}'^i$ which thus represent fully connected graphs. So, from Lemma 4, we deduce that, for each view $i$, we have $\boldsymbol{P}^i = \boldsymbol{P}'^i$ and $\boldsymbol{T}^i = \boldsymbol{T}'^i$, which concludes the proof.

# G  ICA-BASED ALGORITHM FOR LIMVAM WITH SHARED DISTURBANCES

## G.1  ICA-BASED ALGORITHM

---

**Algorithm 4** ICA-LiMVAM

---

1. *Estimate the adjacency matrices $\boldsymbol{B}^i$.*

   (a) Estimate the unmixing matrices $\boldsymbol{W}^i$ and the noise variance matrices $\boldsymbol{\Sigma}^i$, by running the Shared ICA estimation algorithm on the data.
   The algorithm returns an estimate of the true unmixing matrix $\boldsymbol{W}^i = \boldsymbol{M}(\boldsymbol{I} - \boldsymbol{B}^i)$ but up to sign-permutation matrix $\boldsymbol{M}$ that is the same for all views (Richard et al., 2021).

   (b) Determine the sign-permutation indeterminacy $\boldsymbol{M}$, using the structure of the underlying $(\boldsymbol{A}^i)^{-1}$ that has a diagonal of ones. Thus, we can adapt the simple two-step heuristic by Shimizu et al. (2006).
   First, find a permutation matrix $\boldsymbol{M}$ such that $\boldsymbol{M}\boldsymbol{W}$ has a non-zero diagonal, by solving with the Hungarian algorithm

   $$\min_{\boldsymbol{M}} \sum_{j=1}^{p} \frac{1}{|(\boldsymbol{M}\boldsymbol{W})_{jj}|} \,, \quad \boldsymbol{W} = \sum_{i=1}^{m} \mathrm{abs}(\boldsymbol{W}^i) \,, \tag{149}$$

   and then rescale the rows of $\boldsymbol{M}$ to ensure that $\boldsymbol{M}\boldsymbol{W}$ has a diagonal of ones

   $$\boldsymbol{M}_{ij} \leftarrow \mathrm{sign}((\boldsymbol{M}\boldsymbol{W})_{ii})\boldsymbol{M}_{ij} \,, \tag{150}$$

   where the sign function outputs 1 or $-1$.

   (c) Determine the scale matrices $\boldsymbol{D}^i = \mathrm{diag}\left(\frac{1}{(\boldsymbol{M}\boldsymbol{W}^i)_{11}}, \ldots, \frac{1}{(\boldsymbol{M}\boldsymbol{W}^i)_{pp}}\right)$.

   (d) Determine the causal matrices $\boldsymbol{B}^i = \boldsymbol{I} - \boldsymbol{D}^i\boldsymbol{M}\boldsymbol{W}^i$ and update the noise variance matrices with $\boldsymbol{\Sigma}^i \leftarrow (\boldsymbol{D}^i)^2\boldsymbol{\Sigma}^i$.

2. *Determine the causal ordering $\boldsymbol{P}$.*

   The DAG decomposition states that $\boldsymbol{B}^i := \boldsymbol{P}^\top\boldsymbol{T}^i\boldsymbol{P}$ where $\boldsymbol{T}^i$ is lower triangular, so we penalize for that

   $$\min_{\boldsymbol{P}} \sum_{l \geq k} \left(\boldsymbol{P}\boldsymbol{B}\boldsymbol{P}^\top\right)_{kl}^2 \,, \quad \boldsymbol{B} = \sum_{i=1}^{m} \mathrm{abs}(\boldsymbol{B}^i) \,. \tag{151}$$

   We approximately minimize this objective using the heuristic algorithm presented in Shimizu et al. (2006, Algorithm C).

---

Note that the algorithm described in Chen et al. (2024) finds the same adjacency matrices as we do. This is because of the sign-permutation matrix $\boldsymbol{M}$ from ICA: to determine that matrix, we ensure $\boldsymbol{M}\boldsymbol{W}$ has a diagonal of ones by dividing by the scalings $\boldsymbol{D}^i$. For Chen et al. (2024), these scalings are not part of the model: they are simply a part of the algorithm which determines $\boldsymbol{M}$. For us, these scalings are part of our model.

## G.2  COMPUTATIONAL COMPLEXITY

We next detail the worst-case computational complexity of the ICA-LiMVAM algorithm which is in $O(tnmp^3 + mp^5)$, for $t$ iterations of the SharedICA-ML algorithm, $n$ samples, $m$ views and $p$ components. Note that the term in $p^5$ comes from Algorithm C of the original LiNGAM and is worst-case: in practice, it took a couple of iterations only in our experiments. Below are the detail of the derivations for each step of Algorithm 4.

**Computational complexity** The computational complexity of ICA-LiMVAM can be broken down across the four steps in Algorithm 4. Step 1 calls SharedICA, with its most costly variant (SharedICA-ML) running in $O(t \cdot n \cdot m \cdot p^3)$, where $t$ is the number of iterations, $n$ the number

of samples, $m$ the number of views, and $p$ the number of components. Step 2 solves the permutation indeterminacy via a linear sum assignment in $O(m \cdot p^3)$, instead of trying $p!$ permutations. Step 3 involves $m$ multiplications with permutation matrices and additions, costing $O(mp^2)$. Step 4 optimizes over permutations. Instead of trying $p!$ permutations, we use the heuristic Algorithm C from LiNGAM. It is $O(m \cdot p^5)$ in the worst case (and much lower in practice). Further details are provided next:

1. Run the ICA algorithm SharedICA-ML, $O(tnmp^3)$

   Each of the $t$ iterations of the optimization performs the so-called "E-step" and "M-step" of the E-M algorithm, detailed in Section 4 of Richard et al. (2021). Their complexity is in $mnp^3$: the intuition is that for each view $m$ and each sample $n$ require some matrix operations (*e.g.* addition, multiplication, computing gradients and Hessians) all of which are dominated by $p^3$. So the final complexity is in $O(tnmp^3)$.

2. Solve the permutation-sign indeterminacy, $O(mp^3)$

   Computing the objective can be done in $O(p^2)$. The detail of this is: we begin by computing the matrix $W$ which is in $O(mp^2)$, then multiplying it with a permutation matrix which is in $O(p^2)$, and finally summing the diagonal elements which is in $O(p)$.

   Finding the correct permutation matrix can be done by solving a linear-sum assignment problem; this can be done using the Hungarian algorithm which is in $O(p^3)$.

3. Update the entries of a matrix which is in $O(p^2)$.

4. Update the entries of a matrix which is in $O(p^2)$.

   Adding the complexities thus far leads to a complexity of $O(mp^2 + p^3)$, which is dominated by $O(mp^3)$, for Step 2.

5. Compute the causal matrices, $O(mp^2)$

   Each of the $m$ causal matrices requires multiplying a dense matrix with a permutation which is in $O(p^2)$ and then adding a matrix which is in $O(p^2)$. The final cost is $O(mp^2)$.

6. Find the causal ordering(s), $O(mp^5)$

   Algorithm B in the original LiNGAM paper finds a row that has all zeros $O(p^2)$, removes it, does this $p$ times. Its complexity is in $O(p^3)$.

   Algorithm C in the original LiNGAM paper calls Algorithm B at most $p(p-1)/2 = O(p^2)$ times, so its final complexity is $O(p^2 p^3) = O(p^5)$.

   When the causal ordering is shared across views: first we compute sum $m$ matrices with $p^2$ entries which is in $O(mp^2)$, and then we use Algorithm C once which is in $O(p^5)$. The final complexity is in $O(mp^2 + p^5)$ which is dominated by $O(mp^5)$.

   When the causal ordering is view-dependent: for each of the $m$ views, we use Algorithm C which is in $O(p^5)$, so the final complexity is in $O(mp^5)$.

## H  Experiments

### H.1  Synthetic experiments

#### H.1.1  Data generation in Figure 1

The causal effect matrices $\boldsymbol{B}^i = \boldsymbol{P}^\top \boldsymbol{T}^i \boldsymbol{P}$ are generated from a random permutation $\boldsymbol{P}$ and strictly lower triangular $\boldsymbol{T}^i$ obtained from a standard Gaussian; the diagonal matrices $\boldsymbol{D}^i$ are drawn from a uniform density between 0.1 and 3; the common disturbances in $\boldsymbol{s}$ can be either Gaussian or non-Gaussian: Gaussian disturbances $s_j$ are generated from a standardized Gaussian and their corresponding noises $n_j^i$ have standard deviation $\Sigma_{jj}^i$ obtained by sampling from a uniform density between 0 and 1, while non-Gaussian disturbances are generated from a Laplace distribution (with scale parameter equal to $\frac{1}{2}$) and their corresponding noises all have a std of $\frac{1}{2}$. We use $m = 5$ views, $p = 4$ variables, and vary the number of samples $n$ between $10^2$ and $10^4$.

#### H.1.2  Simulation study in higher dimension

We evaluate the methods by measuring the estimation error on the matrices $\boldsymbol{B}^i$ across a range of settings involving more views and components than those considered in the main text. Specifically, we vary the number of views among $\{3, 5, 8, 12, 16, 20\}$ and the number of components among $\{3, 6, 9, 12\}$, using 1000 samples and 50 random seeds for each configuration.

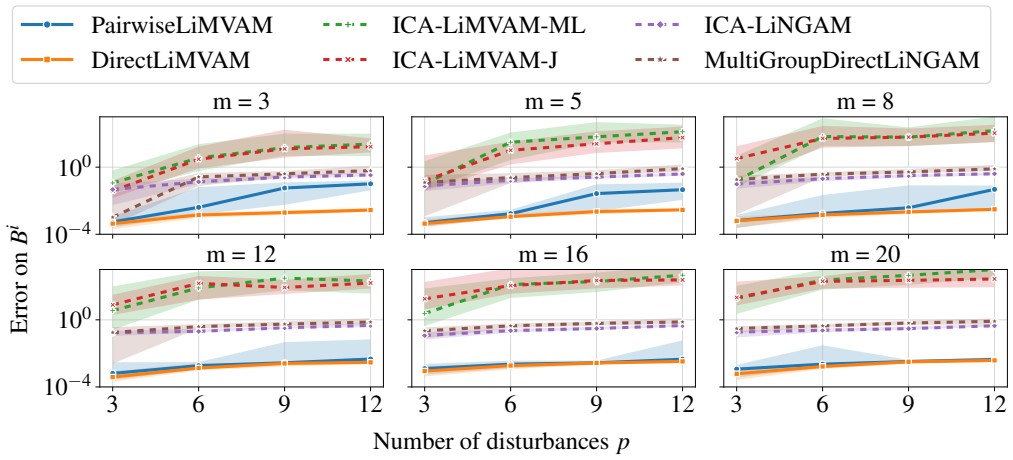

Figure 5: Separation performance of three pairwise-comparisons-based algorithms — Multi-GroupDirectLiNGAM from Shimizu (2012), and DirectLiMVAM and PairwiseLiMVAM which use the criteria in Eq. 5 and Eq. 6, respectively — and three ICA-based algorithms — ICA-LiNGAM, which naively uses Shimizu et al. (2006) separately in each view, and our implementation of the two variants of ICA-LiMVAM Chen et al. (2024). We varied the number of views and components. Disturbances are generated using the generalized normal distribution with shape parameter $\beta \in \{1.5, 2, 2.5\}$. The metric used is the $\ell_2$-distance between true and estimated causal effect matrices $\boldsymbol{B}^i$ (lower is better).

Data are generated from the model in Eq. 3, where disturbances $\boldsymbol{e}_j$ are split into three types—sub-Gaussian, Gaussian, and super-Gaussian—which motivates the choice of disturbance counts as multiples of 3. Sub-Gaussian disturbances follow a generalized normal distribution with shape parameter $\beta = 2.5$; super-Gaussian disturbances use $\beta = 1.5$; Gaussian disturbances correspond to $\beta = 2$. Each $\boldsymbol{T}^i$ is sampled as a strictly lower-triangular matrix with Gaussian entries, and a common permutation matrix $\boldsymbol{P}$ is used to construct the $\boldsymbol{B}^i = \boldsymbol{P}^\top \boldsymbol{T}^i \boldsymbol{P}$. For each method, we report the median error on the matrices $\boldsymbol{B}^i$, with the 25th and 75th percentiles shown as error bars.

As shown in Fig. 5, DirectLiMVAM and PairwiseLiMVAM consistently achieve lower errors than the other approaches. DirectLiMVAM performs best when the number of views is limited (3, 5, or 8) while the number of components remains relatively high (9 or 12), thereby outperforming PairwiseLiMVAM in these settings. This slightly surprising observation that DirectLiNGAM is a

bit better than PairwiseLiNGAM is perhaps due to the fact that there the particular non-Gaussianities used here are ill-suited for the Gaussian likelihood underlying PairwiseLiNGAM.

In contrast, ICA-LiNGAM and MultiGroupDirectLiNGAM achieve slightly better-than-average performance when both the number of views and the number of components are small, but their performance quickly degrades to the average level as dimensionality increases. This behavior reflects their inability to properly handle Gaussian disturbances. Finally, both ICA-LiMVAM variants fail completely, as they are not designed to operate without shared disturbances. Overall, the estimation error tends to increase with the number of components.

### H.1.3    COMPARISON WITH A MULTI-DOMAIN METHOD

Here, we illustrate the advantage of using multi-view methods like ours when cross-view correlations are present in the data. Specifically, we compare our DirectLiMVAM to the method of Perry et al. (2022), MSS (see Appendix I). We used their implementation with the KCI estimator, as this variant performs best in their paper. Since MSS recovers only the common DAG structure (and not the causal weights), we measure performance in terms of recovering the causal ordering.

We used the following experimental setup. Data were generated according to Eq. 1 with 6 views, 4 Gaussian disturbances, 500 samples, and the experiment was repeated over 30 random seeds. The noise variances were all fixed to one, and some entries in the matrices $\mathbf{B}^i$ were allowed to vary across views. The number of such varying entries corresponds to the number of "interventions", *i.e.* changes in causal mechanisms in the sense of Perry et al. (2022).

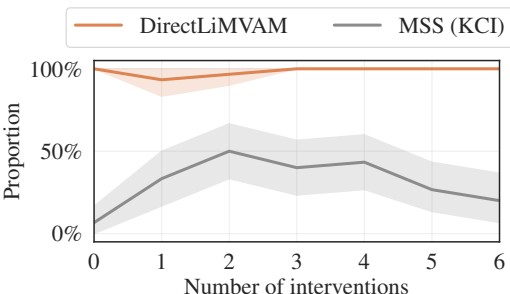

Figure 6: Proportion of runs in which the ordering is perfectly recovered (higher is better). The number of interventions corresponds to the number of entries in the $\mathbf{B}^i$ that are allowed to vary across views. We compare our multi-view algorithm DirectLiMVAM to the multi-domain algorithm MSS of Perry et al. (2022) (using the KCI estimator).

Figure 6 shows that DirectLiMVAM always recovers the correct ordering, regardless of how many entries in $\mathbf{B}^i$ are allowed to vary. By contrast, MSS fails to recover the DAG when the number of interventions is either too small or too large (in line with their Figures 4 and 5), and even in the regime of sparse interventions it perfectly recovers the ordering in only about 50% of the runs. Finally, their method is substantially slower than ours (not shown in the graph).

These results illustrate the distinction between multi-view and multi-domain methods: by explicitly exploiting cross-view correlations, DirectLiMVAM can achieve stable and accurate recovery in settings where a multi-domain method like MSS fails. Furthermore, this experiment confirms that our methods do not require changes in the $\mathbf{B}^i$ as long as there are sufficient correlations between views, thereby extending the identifiability theories in the work just discussed.

### H.1.4    TESTING ASSUMPTION 4

Assumption 4 provides a sufficient condition for the identifiability of the causal matrices $\mathbf{B}^i$ in the LiMVAM model with shared, Gaussian disturbances. To evaluate the practical impact of this assumption, we simulated data from the model in Eq. 3, with shared disturbances as in Eq. 9. We used 5 views, 4 common disturbances in $s$ (2 Gaussian and 2 Laplacian), and 1000 samples.

The scale matrices $D^i$ were drawn uniformly from the interval $[0.5, 2]$, and the strictly lower triangular matrices $T^i$ were sampled from a Gaussian distribution and then permuted to form the causal matrices $B^i$. The noise variances $\Sigma^i_{jj}$ corresponding to the Laplacian disturbances were fixed to $\frac{1}{2}$, while the noise variances corresponding to the two Gaussian disturbances, indexed by $j$ and $j'$, were initially sampled uniformly in $[0, 1]$. To test the assumption, we selected an increasing number of views $i$ such that the scaled noise variances $\frac{\Sigma^i_{jj}}{D^i_{jj}} = \frac{\Sigma^i_{j'j'}}{D^i_{j'j'}}$, ranging from 0 to 5. Note that Assumption 4 only fails when this condition holds across all 5 views. We repeated this experiment over 50 random seeds.

Figure 7 shows that ICA-LiMVAM maintains low estimation error on the $B^i$ matrices as long as the noise diversity assumption holds, but its performance deteriorates sharply once the assumption is violated. This confirms the practical relevance of the assumption. Interestingly, PairwiseLiMVAM appears unaffected by the violation of this condition.

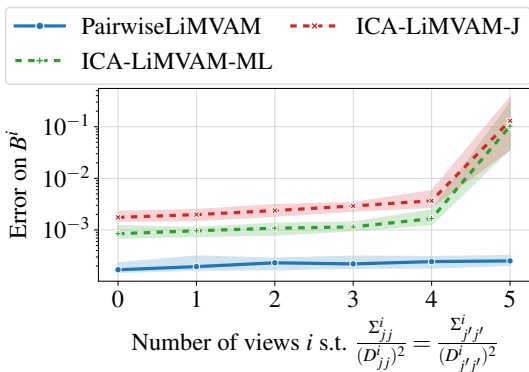

Figure 7: Effect of violating the noise diversity assumption (Assumption 4) on the estimation error of the causal matrices $B^i$. We report the average $\ell_2$ error over 50 repetitions for ICA-LiMVAM and PairwiseLiMVAM, as the number of views with identical scaled noise variances increases.

### H.1.5 TESTING ASSUMPTIONS 5 AND 7

Assumption 5 (resp. Assumption 7) gives a sufficient condition for the identifiability of the causal matrix $P$ (resp. causal orderings $P^i$) and matrices $T^i$, in the shared causal ordering (resp. multiple causal orderings) case. However, it is not straightforward how robust the algorithms are to these assumptions.

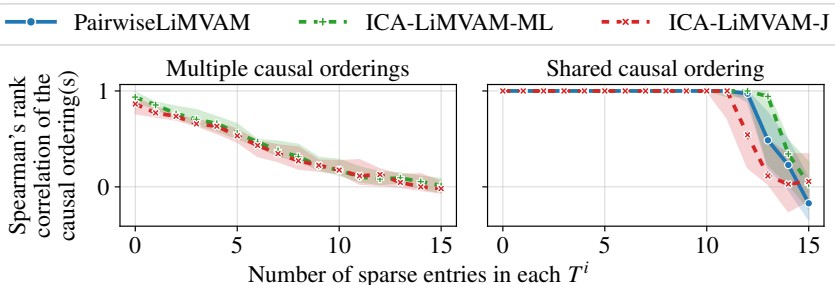

Figure 8: Spearman's rank correlation between true and estimated causal orderings as a function of the number of sparse entries in each $T^i$. Left: In the multiple-ordering setting, performance of ICA-LiMVAM-ML and ICA-LiMVAM-J degrades linearly with increasing sparsity, as the assumption of full support is violated. PairwiseLiMVAM is excluded since it assumes a shared ordering. Right: In the shared-ordering setting, all methods maintain near-perfect recovery up to 11 sparse entries, after which performance drops sharply.

To evaluate the practical impact of these two assumptions, we simulated data from the model in Eq. 3, with shared disturbances as in Eq. 9. We used 8 views, 6 common disturbances (two sub-Gaussians sampled form a generalized normal distribution with shape parameter $\beta = 2.5$; two super-Gaussians for $\beta = 1.5$; two Gaussians corresponding to $\beta = 2$), and 1000 samples. The scale matrices $D^i$ were drawn uniformly from the interval $[0.5, 2]$, the strictly lower triangular matrices $T^i$ were sampled from a Gaussian distribution and then permuted to form the causal matrices $B^i$, and the variances $\Sigma^i$ were sampled uniformly in $[0, 1]$. The experiment was repeated 30 times.

Fig. 8 reports the Spearman's rank correlation between the true and estimated causal orderings as a function of the number of sparse entries in each $\boldsymbol{T}^i$, randomly chosen among the $6 \times 5 = 30$ non-zero entries.

In the multiple causal orderings setting (left panel), both ICA-LiMVAM variants exhibit a roughly linear drop in performance as sparsity increases. This behavior is expected, as Assumption 7 no longer holds once any entry in the strictly lower triangular part of $\boldsymbol{T}^i$ is set to zero. Pairwise-LiMVAM is not included in this setting, as it is designed under the assumption of a shared causal ordering.

In the shared causal ordering setting (right panel), all methods benefit from the shared structure and recover the true ordering $\boldsymbol{P}$ reliably up to 11 sparse entries. Beyond this point, performance degrades sharply for all methods, reflecting the increasing violation of Assumption 5.

These results empirically support the practical relevance of Assumptions 5 and 7.

### H.2 REAL DATA EXPERIMENTS

#### H.2.1 DETAILS ON THE PREPROCESSING OF MEG DATA

The MEG data measures participants' responses to auditory and visual stimuli. The auditory stimuli were binaural pure tones. The visual stimuli were checkerboards presented both to the left and right of a central fixation for 34-ms duration. This task leads to strong signal power modulations during the motor preparation and motor execution.

The original MEG data were acquired with 306 sensors, recorded at 1000 Hz, and band-pass filtered between 0.03 and 330 Hz. All MEG processing was done using the MNE-Python library (Gramfort et al., 2013; 2014) and we largely followed the pre-processing steps used in Power et al. (2023). We applied a Maxwell filter Taulu & Simola (2006) to improve data quality and a band-pass filter between 8 and 27 Hz to focus on power effects spread over the alpha and beta bands of the brain. This range of waves is supposed to be particularly active in sensorimotor tasks, especially during movement preparation and execution, and it typically shows a characteristic suppression (event-related desynchronization) during movement, followed by a rebound (event-related synchronization) after movement cessation, which is thought to reflect sensorimotor processing and inhibitory mechanisms. In particular, the suppression and rebound are reflected in the energies of the signals, not raw signals.

The data were then parsed into trials synchronized to each button press, with a duration of 4.5 s, including a 1.5 s pre-movement interval. The 4.5 s window length was selected to ensure a sufficient post-movement interval to capture the entire beta rebound response. Trials were excluded if the button press occurred more than 1 s after the audiovisual cue (indicating poor task performance) or if another button press occurred within the time window. Then, a baseline correction was applied using the pre-movement interval $(-1.5\,\text{s}, -1\,\text{s})$. The procedure led to about 60 trials per participant on average.

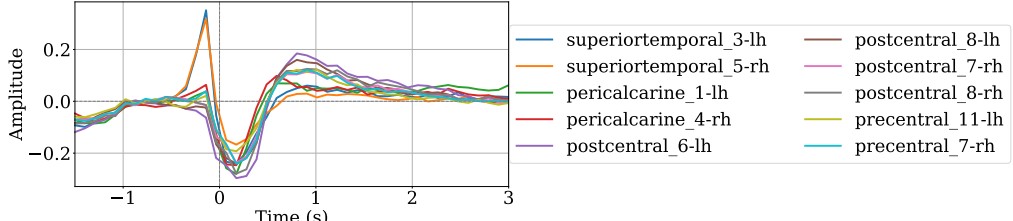

Figure 9: Example of time series obtained after the preprocessing. We averaged the data across subjects and trials, resulting in 10 time series, one for each brain region.

Next, we performed a cortical projection. First, each participant's MRI (additionally given in the Cam-CAN database) was segmented using FreeSurfer Dale et al. (1999). The segmentation provided a digitization of the cortical surface for source estimation, a transformation to the average brain (*i.e.* fsaverage) for spatial normalization and group statistics, and a boundary element model of the head to provide more accurate calculation of the forward solution. The inverse solution, based on the

MNE method Hämäläinen & Ilmoniemi (1994), allowed to consider cortical region activations for further analysis.

We used the cortical parcellation from Khan et al. (2018) to divide the cortical mantel (both hemispheres) into 448 distinct regions and summarized each region by an averaged time course. Then, we selected 10 of the 448 regions based on their known importance in sensorimotor tasks. Specifically, we picked for each hemisphere three regions in the motor cortex (two parcels in the "postcentral" and one in the "precentral"; visible in blue in Fig. 3a), one region in the auditory cortex ("superiortemporal" parcel; highlighted in pink), and one region in the visual cortex ("pericalcarine" parcel; not visible in the figure). Note again that the task for the participant is active as right index button presses are triggered by audiovisual stimulations.

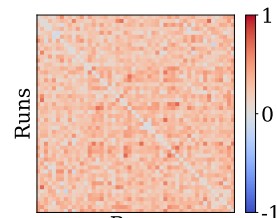

Figure 10: Pearson correlations between median causal effect matrices across 50 runs of ICA-LiMVAM-ML, each using a different random subset of 30 participants from the Cam-CAN cohort.

For each participant, we thus extracted one time series for each of the 10 regions and fixed the number of trials to 40. Participants with less than 40 trials were discarded and extra trials were averaged. Participants for whom some of the parcels did not have any vertex in the source space were also discarded, resulting in a total of 98 available participants. We performed a Z-score normalization of the time series to correct the depth bias and, importantly, computed the Hilbert *envelope* of the signals to study modulations in cortical source power. Finally, the signals were centered and downsampled from 1000 Hz to 10 Hz due to the slowly changing nature of the envelopes (energies). The final dataset consisted of 98 subjects, 10 brain regions, and 1760 time points.

Figure 9 shows the typical time series obtained after preprocessing. For visualization, we averaged the time courses across participants and trials. As expected, we observe a prominent peak in the "superiortemporal" auditory regions shortly before the button press ($t = 0$ s), reflecting the stimulus onset. In the motor regions, a characteristic rebound pattern emerges: the signals decrease around the time of the button press, consistent with event-related desynchronization of beta rhythms, and subsequently increase after approximately 0.3 seconds, indicating beta resynchronization. This figure also labels the ten cortical parcels selected for analysis.

### H.2.2 MEG EXPERIMENT USING ICA-LIMVAM-ML

In the following, we present additional analyses that extend the experiments described in Section 6.2.

Figure 11 displays median causal effect maps estimated using ICA-LiMVAM-ML on the Cam-CAN dataset. For each of the six panels, ICA-LiMVAM-ML was applied to a randomly chosen subset of 50% of the participants. These results are consistent with the patterns observed in Figure 3a, notably the frequent presence of directed connections between motor areas in opposite hemispheres. This further supports the hypothesis of consistent inter-hemispheric causal influences in sensorimotor processing.

Finally, we conducted the same robustness analysis for ICA-LiMVAM-ML as previously done for PairwiseLiMVAM (Figure 3b). Specifically, we ran ICA-LiMVAM-ML 50 times, each time using a randomly selected subset of 30 participants from the full cohort, and computed the Pearson correlations between the element-wise median of the estimated individual matrices $B^i$ across runs. The resulting correlations are shown in Figure 10. While the correlations remain predominantly positive (with an average of 0.27), they are noticeably lower than those obtained with PairwiseLiMVAM. This suggests that, in terms of stability across subsets, PairwiseLiMVAM exhibits greater consistency than ICA-LiMVAM-ML.

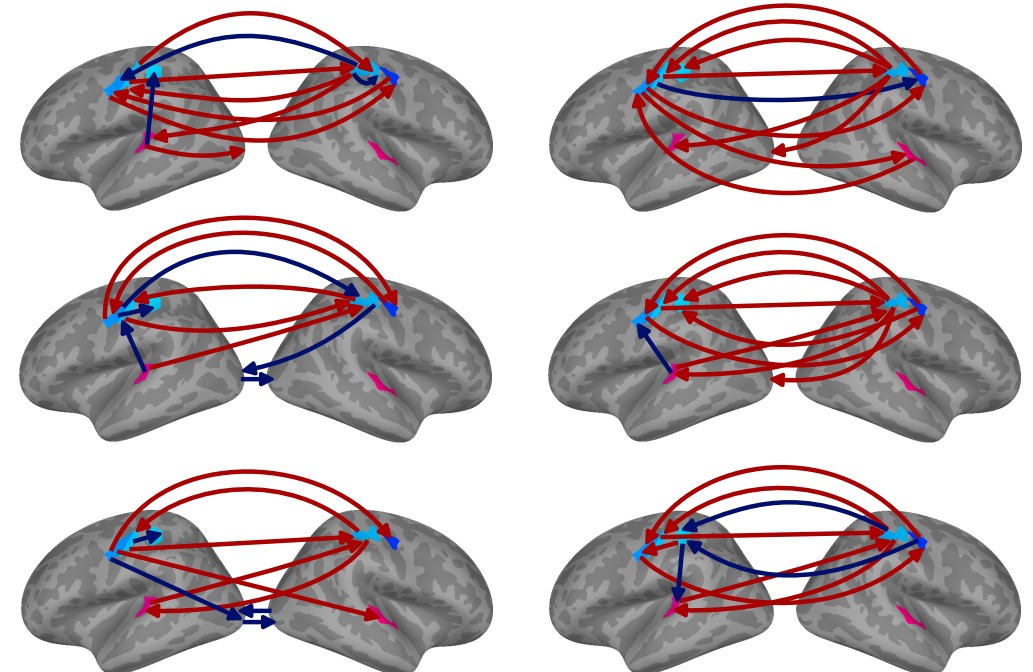

Figure 11: Top ten strongest median causal effects (estimated by ICA-LiMVAM-ML) of six different runs. Each run was performed on the data of 49 randomly chosen subjects. Red arrows represent positive effects and blue arrows negative effects.

# I    DISTINCTION BETWEEN MULTI-VIEW AND MULTI-DOMAIN FRAMEWORKS

Several frameworks study causal discovery from multiple related datasets that share the same causal structure. While *multi-view* methods (like ours) model *correlated* views arising from a joint (non-factorial) distribution, *multi-domain* methods treat datasets as *independent* domains drawn from related but distinct distributions. Consequently, multi-domain methods cannot leverage cross-view correlations and instead rely exclusively on distributional shifts across domains. *Multi-environment* methods are a special case of multi-domain methods, where distributional shifts arise from *interventions* (whether designed by the practitioner or arising from uncontrolled environment differences) on the causal mechanisms. In the following, we review some multi-domain/multi-environment methods that are related to our work.

Ghassami et al. (2018) propose a multi-domain method that, like ours, studies linear causal discovery from multiple datasets that share the same underlying DAG in the causally sufficient setting and without a non-Gaussianity assumption. However, they assume that causal mechanisms $\mathbb{P}(x_j^i | \mathbf{PA}_j^i)$ are *independent* (both within and across domains), where $\mathbf{PA}_j^i$ denotes the parents of $x_j^i$. Our approach does not require such an independence assumption. For identifiability of the true DAG, their method further assumes that noise variances or causal weights vary across domains, and one of their two criteria additionally requires these changes to be sparse. In our paper, we show that such changes in noise variances are sufficient for identifiability (see Assumption 6), but not necessary when the views exhibit diverse correlations.

Adams et al. (2021) study multi-domain identifiability in the more complex setting of *latent confounders*. Their main contributions are necessary and sufficient conditions (bottleneck and strong non-redundancies) that are specific to the confounded setting and vanish in our causally sufficient case. In particular, in this causally sufficient case, their conditions reduce to assuming heterogeneous variances (see their Theorem 1), which is precisely one of the sufficient conditions in our Assumption 6. Moreover, they require the causal weights $\mathbf{B}^i$ to *remain constant* across domains, which is a strong assumption we do not make. Thus, in the causally sufficient case, our method encompasses theirs and goes much further by leveraging correlations (across views and within a view), while allowing causal weights to differ. They suggested, in fact, that their assumptions were too strong in Section 3: "Note that this theorem gives sufficient conditions; our empirical results suggest that they are not necessary."

Peters et al. (2016) propose a multi-environment method that aims to identify the parents of *a single target variable* rather than recovering the full graph, while assuming purely *Gaussian* distributions. Their approach relies on *invariance* of the target's causal mechanism across environments, which implicitly requires knowing where interventions occurred (and that the target was not intervened upon), an assumption that is often unrealistic in practice.

Similarly, Perry et al. (2022) propose another multi-environment method that leverages invariance and can be more easily applied to causal discovery of the whole structure. However, their method considers the more complex case of *nonlinear* relationships. Like Ghassami et al. (2018), they rely on changes in $\mathbb{P}(x_j^i | \mathbf{PA}_j^i)$ across environments, but instead of measuring independence between mechanisms, they count how often these mechanisms change and select the DAG that minimizes this number. Their identifiability results require sparsity in the number of changes; in particular, not all causal mechanisms are allowed to differ across environments. In contrast, our theory accommodates arbitrarily many changes in $\mathbf{B}^i$ and in the distribution of $\mathbf{e}^i$; moreover, these changes are not required for identifiability if cross-view correlations are diverse.

