# OpenReview forum: "Multi-View Causal Discovery without Non-Gaussianity: Identifiability and Algorithms"
_ICLR.cc/2026/Conference — Submitted to ICLR 2026_

### Official Review · Reviewer_vvkn · 2025-10-25

**Soundness:** 4
**Presentation:** 4
**Contribution:** 4
**Rating:** 8
**Confidence:** 3

**Summary:**

This paper proposes a new framework for causal discovery that eliminates the need for the conventional assumption of non-Gaussianity and instead exploits a multi-view data structure to ensure identifiability of causal relations. Traditional approaches generally assume a single view, where it is theoretically impossible to determine causal direction from data without strong additional assumptions. One of the most prominent models, LiNGAM, achieves identifiability by assuming non-Gaussian, independent disturbances. The proposed model, LiMVAM (Linear Multi-View Acyclic Model), is a linear structural equation model (SEM) that handles multiple related data views jointly. The authors show that identifiability can be guaranteed by leveraging second-order statistics (SOS), the covariances across views, without relying on higher-order statistics. Specifically, they prove that if (i) each variable pair is correlated in at least some views, (ii) the correlation patterns are not identical across views, and (iii) the overall graph formed by all views is connected, then both the causal ordering and the coefficient matrices are identifiable using SOS alone. Based on this theory, the paper proposes three new algorithms, PairwiseLiMVAM, DirectLiMVAM, and ICA-LiMVAM, which can be interpreted as multi-view extensions of existing LiNGAM-based methods. In particular, ICA-LiMVAM generalizes ICA-LiNGAM as a special case. Through simulations under both Gaussian and non-Gaussian settings, PairwiseLiMVAM and DirectLiMVAM achieve higher accuracy and faster computation than prior approaches such as ICA-LiNGAM and Multi-Group DirectLiNGAM. Applications to brain-imaging (MEG/fMRI) data further demonstrate that the proposed methods recover physiologically plausible causal relationships.

**Strengths:**

This paper makes an important theoretical contribution by formalizing the general intuition that performing multiple instances of statistical causal discovery can enhance estimation robustness. Under a multi-view formulation, the authors provide a clear set of assumptions and a rigorous logical development showing that causal ordering can be uniquely identified using only second-order statistics (SOS)—without relying on the conventional non-Gaussianity assumption. To the best of my knowledge, this is the first work to demonstrate such identifiability purely from SOS. The paper is clearly written and well structured: the connection to the well-known LiNGAM model is carefully explained, and the mathematical exposition is detailed and persuasive. The demonstration that ICA-LiMVAM generalizes ICA-LiNGAM as a special case further clarifies the positioning of the proposed approach.

The simulation experiments are well designed and convincingly show that the proposed methods perform as predicted under the stated assumptions. Moreover, experiments on real MEG and fMRI datasets demonstrate that the approach is not only theoretically sound but also practically useful.

By leveraging multiple views rather than imposing unrealistic distributional assumptions, the proposed framework provides a promising and potentially high-impact direction for improving the robustness of statistical causal discovery in realistic settings.

**Weaknesses:**

In practical data analysis, it may be difficult to ensure that the assumptions of cross-view correlation and diversity are satisfied. In particular, Assumption 1 requires that for each pair of variables, (i) the correlation is non-zero in at least some views, and (ii) the correlation structures across views are not perfectly proportional. However, in high-dimensional real-world datasets, many variable pairs often exhibit near-zero correlations (i.e., near independence), providing little informative variation, while other groups of variables may be dominated by common latent factors, resulting in nearly identical correlation structures across views. A deeper discussion of how frequently such situations may arise in practice, and how they might be detected or mitigated, would substantially strengthen the paper and improve the practical applicability of the proposed method.

Although this point could also be framed as an opportunity rather than a limitation, one intriguing extension would be to actively generate multiple views, for example, via bootstrap resampling, and apply the proposed framework to enhance the robustness of statistical causal discovery. Such an idea could even lead to new approaches for evaluating the reliability or confidence of inferred causal relations, a problem that has long been challenging. Discussing these possible directions would make the paper’s impact and future potential even greater.

**Questions:**

- In practical datasets, how likely is it that the assumptions of cross-view correlation and diversity are violated? Could the authors provide a deeper discussion on how frequently such violations might occur in real-world settings, and what strategies could be used to detect or avoid such situations?

- Would it be possible to actively generate multiple views, for example, through bootstrap resampling, and apply the proposed method to substantially enhance the robustness of statistical causal discovery? Could this idea potentially evolve into a new approach for assessing the reliability or confidence of estimated causal relationships?

---

> ### Author Response · Authors · 2025-11-21
>
> We thank the reviewer for their positive comments. The two questions they raised might indeed enable promising extensions of the method.
>
> **When are the assumptions challenged ---**
> The reviewer brings up a very interesting point: in which practical situations are our assumptions challenged? We answer in the following and will add the discussion to the main paper.
>
> Our identifiability guarantees rely on the correlations between variables (the reviewer's point (i)) and across views (the reviewer's point (ii)). We agree that both of these may be challenged in practical setups. Specifically, for point (i), it is possible that some pair of variables has near-zero correlation: our assumptions rest on the hope that in at least *one* view, the correlation is strong enough. A pathogological situation is when a problematic variable $x_j$ is uncorrelated with *any* other variables, across *all* views. We believe our framework could naturally deal with such a case: our SoS criteria for determining the causal direction would assign zero to all pairs involving the problematic variable $x_j$, suggesting there is no edge and that it can be completely removed from the dataset. We could, in fact, explicitly incorporate such a step in our algorithms to “filter out” variables that have no edges.
>
> Regarding point (ii), the pathological situation is when the correlation structures across views are nearly identical. This could happen when groups of variables are dominated by common latent factors. Hypothetically, in a neuroimaging setting, a latent common factor could be a stimulus whose amplitude completely determines the response of different subjects while erasing any variability between them. Theoretically, identifiability guarantees would still hold so long as the correlation structures are not identical across views, but this could still lead to numerical challenges and reduced statistical efficiency. We have not explicitly encountered such situations in practice. Nevertheless, one could, on a speculative note, imagine artificially adding some “diversity” to the correlation structures to mitigate the effect of the dominating latent factors; perhaps the reviewer's point about bootstrapping could help here. For example, one could add random low-amplitude noise to some of the views, in the hope that it creates enough diversity in the correlation structures between views, while not perturbing the final DAG.
>
> **Bootstrapping ---**
> It would be a very interesting idea to generate new views by some random process such as bootstrapping; we do not have a clear idea under which circumstances that would work. (In Appendix D.2, we do propose a kind of counterexample on generating new views artificially, but this is using a very different approach.) On the other hand, using bootstrapping to assess the reliability of the results sounds perfectly feasible with our algorithms. We will mention this as a promising extension and leave it for future work.

---

### Official Review · Reviewer_WBtZ · 2025-10-26

**Soundness:** 3
**Presentation:** 2
**Contribution:** 2
**Rating:** 4
**Confidence:** 3

**Summary:**

The paper introduces a multi-view linear SEM (LiMVAM) for causal discovery that exploits correlations across views to obtain identifiability and scalable algorithms without requiring non-Gaussian noise. Two SOS-only, residual-recursive procedures, PairwiseLiMVAM (likelihood-ratio with closed-form LR) and DirectLiMVAM (cross-covariance Frobenius criterion), recover a shared causal ordering and then estimate per-view coefficients with joint FGLS. A complementary “shared-disturbance” formulation (with per-view scalings) connects LiMVAM to Shared ICA, yielding identifiability even when some disturbances are Gaussian. Conceptually, LiMVAM generalizes multi-environment/non-stationary ideas by treating environments as “views” and using diversity of second-order structure to replace classical non-Gaussianity assumptions.

**Strengths:**

a. The multi-view setting is both common and important. This paper extends two single-view causal-discovery algorithms to the multi-view case and removes the need for non-Gaussian noise assumptions.

b. The estimators are fast and straightforward: closed-form likelihood-ratio and cross-covariance tests within a recursive-residual scheme, followed by joint FGLS for edge weights, essentially no tuning required.

c. The real-world experiments are good, including MEG/fMRI case studies with cross-subject stability checks that support the practical value of the approach.

**Weaknesses:**

a. The contribution is close to existing multi-environment/multi-domain lines. Several works also target identification of linear systems in multi-view or multi-domain settings but are not discussed. For example, [1] addresses causal structure learning for linear relations without relying on non-Gaussian noise or inter-view correlation; [2] studies identification under heterogeneous/noise-variance shifts across domains; and invariance-based approaches such as [3] provide a related lens on leveraging distributional stability across environments. A clearer positioning relative to these would sharpen the paper’s novelty claims.

b. In sections 4.2–4.4, many key conditions and results deferred to the appendix. It would improve readability and verifiability to promote the central statements to the main text as formal theorems/definitions (with precise assumptions), leaving proofs and extended discussion to the appendix.

c. In synthetic experiments, multi-environment baselines beyond LiNGAM-style variants are limited.


[1] Ghassami A E, et al. Multi-domain causal structure learning in linear systems[J]. Advances in neural information processing systems, 2018, 31.

[2] Adams J, et al. Identification of partially observed linear causal models: Graphical conditions for the non-gaussian and heterogeneous cases[J]. Advances in Neural Information Processing Systems, 2021, 34: 22822-22833.

[3] Peters J, et al. Causal inference by using invariant prediction: identification and confidence intervals[J]. Journal of the Royal Statistical Society Series B: Statistical Methodology, 2016, 78(5): 947-1012.

**Questions:**

a. Lines 96–100: the partial-ordering example is unclear. If the adjacency/structural matrix is the zero matrix, the DAG has no edges, so all variables are mutually independent; in that case there is no informative partial order beyond the trivial one?

b. On the assumption that views must be correlated, does this imply a shared latent factor across views?

c. Theorems 1–2: if each view’s coefficient matrix $B^i$ is identifiable, why is an additional assumption needed to recover the causal ordering $P$? Is $B^i$ identified only up to a common permutation of variables (leaving the topological order ambiguous), or is $B^i$ in Theorems 1–2 different from the structural matrix in Eq. (3)?

---

> ### Author Response · Authors · 2025-11-21
>
> We thank the reviewer for taking the time to provide these comments. In the following two messages, we give a detailed response. We hope this satisfactorily addresses the reviewer's concerns.
>
> **Multi-view vs. Multi-domain ---**
> We hope the reviewer agrees our *multi-view* model and the *multi-domain* models cited are quite different. In statistical terms, the fundamental difference is that in the multi-view setting, the views are statistically dependent (correlated), while in the multi-domain setting, the domains are independently generated and must have different distributions. We agree with the reviewer that a deeper discussion of multi-domain methods would better situate our contribution and highlight the novelty of the paper, and we welcome the opportunity to clarify these distinctions. In the following we go through the cited work in detail; this discussion will be added to the paper to the extent that space permits.
>
> **Comparison with the cited work ---**
> Ghassami et al. and Peters et al. both exploit distributional shifts across environments. Specifically, they rely on changes (or the absence of changes) in conditional distributions $\mathbb{P}(x^i_j | \mathbf{PA}^i_j)$, where $\mathbf{PA}^i_j$ denotes the parents of $x^i_j$. For their methods to be consistent, they require assumptions we do not make. While Ghassami et al. do not impose restrictions on the noise distributions, Peters et al. focus mainly on *Gaussian* distributions (which we do not require). In addition, Ghassami et al. assume *independent causal mechanisms* and require that noise variances or causal weights vary across environments, with one of their two criteria further imposing sparsity in these changes. In contrast, we show that such changes in noise variances are in themselves sufficient for identifiability (see Assumption 6), but not necessary when the views exhibit diverse correlations. Finally, Peters et al. aim to identify the parents of *a single target variable* rather than recovering the full graph. Their approach relies on *invariance* of the target’s causal mechanism across environments, which implicitly requires knowing where interventions occurred (and that the target was not intervened upon), an assumption that is often unrealistic in practice.
>
> Similarly, Perry et al. [1] propose another method that leverages invariance and can be more easily applied to causal discovery of the whole structure. However, their method studies the more complex case of *nonlinear* relationships, leading to stronger identifiability conditions.
>
> Adams et al. study identifiability in the more complex setting of *latent confounders*. Their main contributions are necessary and sufficient conditions (bottleneck and strong non-redundancies) that are specific to the confounded setting and vanish in our causally sufficient case. In particular, in this causally sufficient case, their conditions reduce to assuming heterogeneous variances (see their Theorem 1), which is precisely one of the sufficient conditions in our Assumption 6. Moreover, they require the causal weights $\mathbf{B}^i$ to *remain constant* across environments, which is a strong assumption we do not make. Thus, in the causally sufficient case, our method encompasses theirs and goes much further by leveraging correlations (across views and within a view), while allowing causal weights to differ. They suggested, in fact, that their assumptions were too strong in Section 3: “Note that this theorem gives sufficient conditions; our empirical results suggest that they are not necessary.”
>
> Our identifiability theory can thus be seen as an extension of these four papers, since leveraging correlations between views, in a multi-view setting, allows for milder conditions.
>
> In summary, our work is clearly related to multi-domain causal discovery, and we agree that spelling out these differences is important for positioning. The central distinction is conceptual: in the multi-view setting considered here, views are statistically *dependent* and may even share the same marginal distribution; in the multi-domain setting, domains are treated as *independent* (or dependence is not modeled) and must differ in distribution to be informative. Importantly, we show how cross-view correlations can be used to create new identifiability theory and new, efficient algorithms. We will make this positioning explicit in the revised version.
>
> [1] Perry R, et al. Causal discovery in heterogeneous environments under the sparse mechanism shift hypothesis. NeurIPS 2022.

---

> ### Author Response · Authors · 2025-11-21
>
> We continue here the previous response.
>
> **Technical details ---**
> We thank the reviewer for their valuable feedback.
> Weakness b: We will revise Sections 4.2-4.4 to include additional details and clarifications. The original 9-page limit constrained the level of detail we could provide, but the extra page in the camera-ready version will allow us to address this.
> Question a: You are correct that in the degenerate example the induced causal order is trivial, as it contains no precedence constraints. Our intention was to highlight that such a trivial order is compatible with any *permutation* of the variables (and not any order), but we will rephrase this in the final version.
> Question c: This question is related to the confusion with Weakness b. In Theorems 1-2, the $\mathbf{B}^i$ do correspond to the structural matrices in Eq. (3), and they are entirely identified (not up to a common permutation of the variables). However, as discussed in Weakness b, a DAG is associated with an order that corresponds to multiple permutations when the order is partial, and only one permutation when the order is total. Thus, the permutation $\mathbf{P}$ can only be identifiable when the order is total (which is not the case in the degenerate example, for instance), which explains why we need an extra assumption.
>
> **Baseline comparisons ---**
> We agree that benchmarking against additional methods is always worthwhile. In this work, we focused on methods from the LiNGAM family that are most directly comparable to our setting. We are currently working on incorporating the algorithm of Perry et al. into our simulations and, if successful, we will include the corresponding results in the revised version.
>
> **Does correlation between views imply latent factors? ---**
> Indeed, if the data is Gaussian, any correlation between views can always be modelled as being a result of a *linear* latent factors. In fact, *any* dependencies can be modelled by *nonlinear* latent factors as is well-known in the theory of nonlinear ICA (*e.g.* Hyvarinen \& Pajunen, Neural Networks, 1999). However, we don't need to use such an extra layer of latent variable modelling in our approach, and for simplicity, we choose to simply allow for arbitrary correlation structures over views (in the main model).

---

> ### Author Response · Authors · 2025-12-02
>
> **Baseline comparisons (follow-up) ---**
> The reviewer noted that multi-environment comparisons beyond LiNGAM-style variants are limited. The reason is that *multi-view* causal discovery methods remain scarce. Nevertheless, to better address this concern, we conducted simulations with an additional method from the *multi-domain* framework. Among the four methods mentioned by reviewers KaQA and WBtZ, only the method of Perry et al. (MSS) has publicly available code; we used their implementation with the KCI estimator, as this variant performs best in their paper. Since MSS recovers only the common DAG structure (and not the causal weights), we compare it to our DirectLiMVAM in terms of recovering the causal ordering.
>
> We used the following experimental setup. Data were generated according to our model with 6 views, 4 Gaussian disturbances, 500 samples, and the experiment was repeated over 30 random seeds. The noise variances were all fixed to one, and some entries in the matrices $\mathbf{B}^i$ were allowed to vary across views. The number of such varying entries corresponds to the number of “interventions”, *i.e.* changes in causal mechanisms in the sense of Perry et al. The results are reported (and highlighted in blue) in Appendix H.1.3 of the revised paper. In particular, DirectLiMVAM *always* recovers the correct ordering, regardless of how many entries in $\mathbf{B}^i$ are allowed to vary. By contrast, MSS fails to recover the DAG when the number of interventions is either too small or too large (in line with their Figures 4 and 5), and even in the regime of sparse interventions it perfectly recovers the ordering in only about 50\% of the runs. Finally, their method was far slower than ours.
>
> These results illustrate the distinction between multi-view and multi-domain methods: by explicitly exploiting cross-view correlations, DirectLiMVAM can achieve stable and accurate recovery in settings where a multi-domain method like MSS fails. Furthermore, this experiment confirms that our methods do not require changes in the $\mathbf{B}^i$ as long as there are sufficient correlations between views, thereby extending existing identifiability theories.

---

### Official Review · Reviewer_KaQA · 2025-10-30

**Soundness:** 3
**Presentation:** 3
**Contribution:** 3
**Rating:** 6
**Confidence:** 3

**Summary:**

This paper tackles the formidable challenge of uncovering causal structure from observational data by capitalizing on multi-view architectures, wherein multiple correlated datasets (or "views") share an underlying causal dependency system. Transcending the conventional prerequisite of non-Gaussianity for reliable identifiability (as is characteristic of LiNGAM), the authors propose an innovative multi-view linear SEM framework.This framework achieves identifiability under substantially less stringent assumptions by exclusively utilizing heterogeneous Second-Order Statistics (SOS) across the distinct views. The study furnishes rigorous theoretical guarantees of identifiability, devises streamlined multi-view extensions of established algorithms (DirectLiNGAM, PairwiseLiNGAM, and an ICA-based approach), and empirically validates and benchmarks these methodologies on both synthetic and large-scale neuroimaging datasets.

**Strengths:**

- The paper delivers a significant contribution by establishing the identifiability of linear SEMs using only second-order statistics, eliminating the reliance on non-Gaussianity
- The generalization of DirectLiNGAM and PairwiseLiNGAM to the multi-view setting is addressed in a thoughtful manner, yielding novel fast SOS-based algorithms. Additionally, the adaptation of ICA-LiNGAM methodology to accommodate multi-view shared disturbances is conceptually elegant.
- The paper is well-organized and clearly written.

**Weaknesses:**

- What is the difference between Ghassami[1] and Perry[2] in this paper?

- In Section 3, It would be better that the assumption, "All adjacency matrices ${B}_i$ share the same causal ordering" be explicitly highlighted (or formal definition).

- The definitions of some superscripts and subscripts in the text are easily confusing. For example, in Equation (8) of Section 4.5, which represents different views, the superscripts $i$ and $i'$ hinder readability.

- In Section C.2，Why is matrix $B$ in the form of $diag(cov(x^1,y^1),...,cov(x^m,y^m))$?

- The paper only conducted limited baseline comparisons, without benchmarking against recently emerged causal discovery methods.




## References
> 1、AmirEmad Ghassami, Negar Kiyavash, Biwei Huang, and Kun Zhang. 2018. Multi-domain causal structure learning in linear systems. In Proceedings of the 32nd International Conference on Neural Information Processing Systems (NIPS'18). Curran Associates Inc., Red Hook, NY, USA, 6269–6279.

> 2、Ronan Perry, Julius von Kügelgen, and Bernhard Schölkopf. 2022. Causal discovery in heterogeneous environments under the sparse mechanism shift hypothesis. In Proceedings of the 36th International Conference on Neural Information Processing Systems (NIPS '22). Curran Associates Inc., Red Hook, NY, USA, Article 792, 10904–10917.

**Questions:**

See Weaknesses.

---

> ### Author Response · Authors · 2025-11-21
>
> We thank the reviewer for taking the time to provide these comments. In the following two messages, we give a detailed response.
>
> **Multi-view vs. Multi-domain ---**
> We wish to emphasize that our model is *multi-view*, while the work cited by the reviewer is *multi-domain*. These are two quite different things, and therefore, there is little overlap between our work and theirs. As an intuitive example, multi-view would correspond to a person seeing the same object from different angles, while multi-domain would correspond to a person seeing different objects. In statistical terms, the fundamental difference is that in the multi-view setting, the views are statistically dependent (correlated), while in the multi-domain setting, the domains are independently generated. In the following, we provide a detailed discussion that we will add to the paper, space permitting.
>
> **Differences with Ghassami and Perry ---**
> Indeed, Ghassami et al. and Perry et al. study causal discovery from multiple domains/environments that share the same underlying DAG in the causally sufficient setting and without a non-Gaussianity assumption. However, their approach differs substantially from ours and is more naturally classified as “multi-domain” rather than “multi-view”. Both of the papers cited treat environments as *independent* domains drawn from related but distinct distributions, whereas we model *correlated* views arising from a joint (non-factorial) distribution. Consequently, they cannot leverage cross-environment correlations and instead rely exclusively on distributional shifts across environments.
>
> More specifically, Ghassami et al. consider linear relationships, as we do, and assume that the causal mechanisms $\mathbb{P}(x^i_j | \mathbf{PA}^i_j)$ are *independent* (both within and across environments), where $\mathbf{PA}^i_j$ denotes the parents of $x^i_j$. Our approach does not require such an independence assumption. For identifiability of the true DAG, their method further assumes that noise variances or edge weights vary across environments, and one of their two criteria additionally requires these changes to be sparse. In our paper, we show that such changes in noise variances are sufficient for identifiability (see Assumption 6), but not necessary when the views exhibit diverse correlations. Our algorithms explicitly exploit these cross-view correlations. Finally, our code is open-source, whereas we were unable to find an implementation of Ghassami et al. online.
>
> Perry et al. address a more general setting with *nonlinear* relationships. Like Ghassami et al., they rely on changes in $\mathbb{P}(x^i_j | \mathbf{PA}^i_j)$ across environments, but instead of measuring independence between mechanisms, they count how often these mechanisms change and select the DAG that minimizes this number. Their identifiability results require sparsity in the number of changes; in particular, not all causal mechanisms are allowed to differ across environments. In contrast, our theory accommodates arbitrarily many changes in $\mathbf{B}^i$ and in the distribution of $\mathbf{e}^i$; moreover, these changes are not required for identifiability if cross-view correlations are diverse. Computationally, the procedure of Perry et al. can be time-consuming, while our methods, by contrast, are simple and fast, yet perform well in practice.
>
> Overall, the key conceptual difference between these two works and ours follows the distinction between multi-domain and multi-view methods (despite some variation in terminology in the literature). In the multi-view setting, as in our work, views are statistically *dependent* --- they are generated from a non-factorized joint distribution --- and may even share the same marginal distribution. In the multi-domain setting, as in Ghassami et al. and Perry et al., domains are assumed *independent* and must differ in distribution to be informative.
>
> In neuroimaging, this distinction makes multi-view methods particularly well-suited to experiments centered around stimulus onsets (as in our MEG and fMRI experiments), while multi-domain methods are more naturally aligned with resting-state experiments (as in the fMRI experiment of Ghassami et al.). We agree that making this distinction explicit would improve the paper, and we will incorporate this discussion in the revised version.

---

> ### Author Response · Authors · 2025-11-21
>
> We continue here the previous response.
>
> **Technical details ---**
> We thank the reviewer for their valuable comments. We will highlight as a formal assumption the fact that all $\mathbf{B}^i$ share the same causal ordering. We also acknowledge that the notation $\mathbf{B}$ in Sections B and C is confusing. Throughout the paper, the $\mathbf{B}^i$ denote adjacency matrices, whereas in Sections B and C, we define $\mathbf{B} = \mathrm{diag}(\mathrm{cov}(x^1, y^1), \dots, \mathrm{cov}(x^m, y^m))$, which does not correspond to the aforementioned $\mathbf{B}^i$. We will adopt a different notation in the revised version to avoid this ambiguity. However, the reviewer asked if it was possible to change some subscripts and superscripts, in particular in Equation (8). Although we agree that these notations can be confusing, we have not found an alternative that is clearly less ambiguous. We therefore prefer to keep the current notation, but will clarify after Equation (8) that $i, i'$ are for views, and $j, j'$ for variables. We hope this will sufficiently alleviate the concern.
>
> **Baseline comparisons ---**
> We agree that benchmarking against additional methods is always worthwhile. In this work, we focused on methods from the LiNGAM family that are most directly comparable to our setting. We are currently working on incorporating the algorithm of Perry et al. into our simulations and, if successful, we will include the corresponding results in the revised version.

---

> ### Author Response · Authors · 2025-12-02
>
> **Baseline comparisons (follow-up) ---**
> The reviewer noted that comparisons against recently emerged causal discovery methods are limited. The reason is that *multi-view* causal discovery methods remain scarce. Nevertheless, to better address this concern, we conducted simulations with an additional method from the *multi-domain* framework. Among the four methods mentioned by reviewers KaQA and WBtZ, only the method of Perry et al. (MSS) has publicly available code; we used their implementation with the KCI estimator, as this variant performs best in their paper. Since MSS recovers only the common DAG structure (and not the causal weights), we compare it to our DirectLiMVAM in terms of recovering the causal ordering.
>
> We used the following experimental setup. Data were generated according to our model with 6 views, 4 Gaussian disturbances, 500 samples, and the experiment was repeated over 30 random seeds. The noise variances were all fixed to one, and some entries in the matrices $\mathbf{B}^i$ were allowed to vary across views. The number of such varying entries corresponds to the number of “interventions”, *i.e.* changes in causal mechanisms in the sense of Perry et al. The results are reported (and highlighted in blue) in Appendix H.1.3 of the revised paper. In particular, DirectLiMVAM *always* recovers the correct ordering, regardless of how many entries in $\mathbf{B}^i$ are allowed to vary. By contrast, MSS fails to recover the DAG when the number of interventions is either too small or too large (in line with their Figures 4 and 5), and even in the regime of sparse interventions it perfectly recovers the ordering in only about 50\% of the runs. Finally, their method was far slower than ours.
>
> These results illustrate the distinction between multi-view and multi-domain methods: by explicitly exploiting cross-view correlations, DirectLiMVAM can achieve stable and accurate recovery in settings where a multi-domain method like MSS fails. Furthermore, this experiment confirms that our methods do not require changes in the $\mathbf{B}^i$ as long as there are sufficient correlations between views, thereby extending existing identifiability theories.

---

### Official Review · Reviewer_ayyF · 2025-11-01

**Soundness:** 1
**Presentation:** 1
**Contribution:** 1
**Rating:** 2
**Confidence:** 4

**Summary:**

This article addresses the challenging problem of causal discovery, which traditionally relies on strong assumptions like non-Gaussianity. The authors introduce a novel approach that leverages multi-view data.

**Strengths:**

This article addresses the challenging problem of causal discovery, which traditionally relies on strong assumptions like non-Gaussianity. The authors introduce a novel approach that leverages multi-view data.

**Weaknesses:**

I have a concern regarding the step in the identifiability proof where independence is concluded from the vanishing covariances. As is well-known, vanishing covariance does not generally imply independence. Since the proof seems to leverage this implication, its validity depends critically on the underlying distributional assumptions. Please clarify how the non-Gaussianity of the disturbances, potentially via the framework of the Darmois–Skitovich theorem, guarantees that this implication holds in the proposed multi-view model. A more detailed explanation in the manuscript would be essential. The second order of statistics may not be sufficient for the estimation.

**Questions:**

See above.

---

> ### Author Response · Authors · 2025-11-21
>
> We are quite confused by the reviewer's comments which do not seem to apply to our paper.
> The reviewer claims that zero correlation is generally *not* equivalent to independence. We agree, but never do we use this in our proofs. Could the reviewer point out exactly which “step in the identifiability proof” (line or equation number) is of concern? We would be happy to clarify.
> Overall, we *do* show that second-order statistics are sufficient for estimating a multi-view Structural Squation Model (SEM); we do not use the Darmois-Skitovich Theorem which requires non-Gaussianity and is therefore not relevant to methods based on second-order statistics. Our contributions include novel identifiability guarantees and algorithms that do not need non-Gaussianity, as also agreed by the other reviewers.

---

### Meta-Review · Area_Chair_VQnk · 2025-12-28

**Summary:**

This submission proposes LiMVAM, a multi-view linear SEM framework for causal discovery that aims to replace the standard non-Gaussianity requirement with heterogeneity in second-order statistics across correlated views. The paper provides identifiability claims for acyclic SEMs under cross-view correlation/diversity conditions, introduces SOS-based multi-view analogues of DirectLiNGAM/PairwiseLiNGAM and an ICA-inspired variant, and supports the approach with simulations and neuroimaging (MEG/fMRI) experiments. One reviewer finds the work technically and conceptually strong with clear promise, while the remaining reviews are more cautious, emphasizing that key aspects of positioning, assumptions, and evaluation are not yet convincingly resolved within the current submission.

**Reviewer Concerns:**

The primary concern is that the novelty and scope relative to closely related multi-environment/multi-domain and invariance-based lines (e.g., heterogeneous environment structure learning and mechanism-shift/invariance approaches) are not sufficiently crisp in the paper as reviewed; while the authors’ rebuttal explains the intended “multi-view (dependent joint)” versus “multi-domain (independent environments)” distinction and adds additional discussion and experiments, the submission still reads as requiring substantial reframing to make the conceptual separation, assumptions, and main claims unambiguous to a broad ICLR audience. A second concern is presentation and verifiability: multiple reviewers noted that central conditions/results are deferred to appendices and that notation is confusing in places; the rebuttal commits to improvements, but this indicates that the current version is not yet in a state where readers can readily audit the assumptions-to-identifiability-to-algorithm pipeline. A third concern is empirical coverage: baseline comparisons beyond LiNGAM-style variants were originally limited; the authors added a multi-domain baseline (MSS) comparison in an appendix and report favorable outcomes, but the overall evaluation remains narrower than what is typically expected for a strong acceptance, particularly given the claim of competing with broader heterogeneous-environment methods. Finally, reviewer ayyF provides a strongly negative score with a brief technical objection about an “independence from zero covariance” step; the authors dispute that this step exists and flagged the review as anomalous. Even discounting the anomalous tone, the existence of such a mismatch underscores that the manuscript’s proof exposition should be tightened so that any potential points of confusion are eliminated; taken together with the remaining unresolved positioning and evaluation concerns, I do not find the rebuttal sufficient to reach a clear acceptance recommendation.

**Reviewer Scores:**

vvkn would plausibly remain strongly positive, KaQA would likely remain near the margin or move slightly upward, and WBtZ could move from slightly-below-threshold to slightly-above-threshold if the promised revisions and added baseline are credited, while ayyF would most likely remain unchanged or be treated with reduced weight given the anomalous characteristics. Even under these optimistic inferences, the overall picture remains mixed and dependent on substantial clarification and broader empirical positioning, which supports a Reject decision for this cycle.

---

### Decision · Program_Chairs · 2026-01-26

Reject